# BounDr.E: Predicting Drug-likeness through knowledge alignment and EM-like one-class boundary optimization

## Abstract

The advent of generative AI models is revolutionizing drug discovery, generating *de novo* molecules at unprecedented speed. However, accurately identifying and rescuing drug candidates among countless generated molecules remains an open problem. The essence of this drug-likeness prediction task lies in constructing a compact subspace that encompasses majority of approved drugs with only a small number of unknown compounds (drug candidates) inside. Computational challenges arises in constructing a decision boundary on an unbound chemical space that lacks definite negatives, i.e, non drug-likeness. Approved drugs exist highly dispersed across structural space, making it more harsh to effectively separate drugs from non-drugs through existing classifiers. Addressing such challenges, we introduce BounDr.E: a novel approach for learning a compact boundary of drug-likeness through an Expectation-Maximization (EM)-like iterative optimization process. Specifically, we refine both the boundary and the distribution of the embedding space via metric learning, allowing the model to iteratively tighten the drug-like boundary while pushing non-drug-like compounds outside. Augmented by integration of biomedical context within knowledge graphs via multi-modal alignment, our model demonstrates 10% increase in F1 score over the previous state-of-the-art, along with strongest robustness to cross-dataset validation. Zero-shot toxic compound filtering and comprehensive drug discovery pipeline case studies further showcases its utility in large-scale screening of AI-generated compounds. To facilitate *in silico* drug discovery, we provide the code and benchmark data under various splitting schemes at: anonymous.4open.science/r/boundr_e.

## 1 Introduction

The expansion of deep generative models for molecular design is transforming the drug discovery landscape, generating vast libraries of candidate molecules with unprecedented speed (Guan et al., 2023; Lee et al., 2023; Song et al., 2024). However, evaluating the drug-likeness of these molecules is still a major challenge. Conventional filters, such as Lipinski's Rule of Five (Ro5; Lipinski et al. (1997)) and Quantitative Estimate of Drug-likeness (QED; Bickerton et al. (2012)), offer helpful initial screens, but they fail to provide a definitive boundary for drug-like properties (Jin et al., 2018;

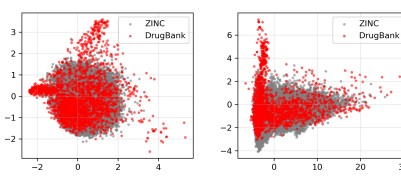

(a) Fingerprint  (b) GraphMVP

**Figure 1:** PCA visualization of embedding spaces of approved drugs (red) and 100k ZINC compounds (gray).

Lee et al., 2023; Li et al., 2024). A more precisely defined boundary of drug-likeness is required, capturing approved drugs while excluding non-drug-like compounds.

However, approved drugs are widely dispersed across chemical space, with an average of only 1.97 drugs sharing the same scaffold. This high dispersion makes it challenging to draw compact boundaries for drug-likeness. For example, a one-class hypersphere of drugs often includes *all* non-drugs within the drug boundary, regardless of whether Morgan fingerprint or deep learning representation spaces (Liu et al., 2022) are used (Figure 1, Appendix B). Furthermore, the task of drug-likeness prediction poses two computational challenges:

**Figure 2:** Overview of BOUNDR.E. **Step 1** performs multi-modal mixup of two drug spaces: knowledge graph $\mathcal{K}$ and molecular fingerprint $\mathcal{S}$ spaces into a unified space $\mathcal{U}$. **Step 2** performs EM-like boundary optimization, where in E-step boundary $\mathcal{B}$ is updated and in M-step the latent space $\mathcal{Z}$ is updated by pushing the out-boundary non-drugs further while contracts drugs to the center to yield an optimized drug-like boundary.

1. **The absence of a clear negative class.** In drug discovery, every compound could potentially be a drug candidate, and there is no definitive set of "non-drugs."
2. **The vastness of chemical space.** With an estimated number of $10^{23} \sim 10^{60}$ synthesizable compounds (Polishchuk et al., 2013), it is impractical to sample a representative set for training, and also there is no known ratio of drug-like to non-drug-like compounds.

Existing methods for drug-likeness prediction tend to fall short. Supervised models (Sun et al., 2022) often generate overly strict decision boundaries by treating unlabeled compounds as hard negatives, while PU learning approaches (Lee et al., 2022) assume the unlabeled set as a mixture of *tractable* positive and negative label distribution, which is unpractical in compound space (Appendix A.6) Both approaches root on risk minimization which enforces their reliance on the negative set. Unsupervised models (Li et al., 2024) produce overly broad boundaries that fail to generalize.

One-class classification models (Schölkopf et al., 2001; Tax & Duin, 2004; Ruff et al., 2018), while promising in mitigating the issue of reliance on ill-defined negatives, their static nature and reliance on fixed feature space lead to broad boundary and high false positive ratio in open chemical spaces.

To overcome these limitations, we introduce BOUNDR.E, a dynamic approach for predicting drug-likeness that refines the boundary and the embedding space iteratively using an EM-like optimization process, guided by biomedical knowledge alignment (Figure 2). Our method iteratively adapts the boundary to enclose as many drug-like compounds as possible while pushing non-drug-like compounds outward through metric learning. By continuously adapting the boundary, BOUNDR.E improves upon the overly rigid approaches of conventional one-class classification (OCC) models, ensuring a tighter, more precise boundary for drug-likeness prediction.

For the guidance of biomedical knowledge, we augment the initial embedding space using multi-modal knowledge alignment, integrating molecular structure with biological and pharmacological data through a novel knowledge-enhanced mixup technique. This fusion of information allows our model to capture more biologically meaningful features of drug-likeness, improving performance in time-based and scaffold-based splits and generalization to unseen chemical scaffolds. Our contributions can be summarized as following: 1) Novel formulation of drug-likeness prediction as a one-class classification without reliance on negatives. 2) Proposal of EM-like optimization of both the drug-likeness boundary and the embedding space for accurate drug-likeness prediction. 3) Knowledge-integrated multi-modal alignment of structure and biomedical knowledge embeddings for defining drug-likeness with machine learning.

Experimental results demonstrate superior performance in drug-likeness prediction, achieving high F1-scores and Matthews correlation coefficients (MCC), as well as favorable recommendation performance metrics including Average Precision (AP). Additionally, BOUNDR.E excels in zero-shot toxic compound filtering, showcasing its cross-dataset generalizability. Comprehensive case studies further showcase its utility in large-scale screening of AI-generated compounds, offering a highly efficient solution for initial screening of real-world *in silico* drug discovery applications.

## 2 RELATED WORKS

**Computational Prediction of Drug-likeness** Computational identification of drug-like compounds has long been a focus in drug discovery (Clark & Pickett, 2000), starting with molecular

descriptor-based metrics including the Rule of Five (Ro5) (Lipinski et al., 1997) and QED (Bickerton et al., 2012). However, these methods are limited to acting as "necessary conditions" for drug-likeness, not as definitive classifiers as pointed out by several studies (Lee et al., 2022; 2023; Li et al., 2024). Recent graph neural network-based approaches including D-GCAN (Sun et al., 2022) and DeepDL (Lee et al., 2022) employ binary classifiers and PU learning, but their reliance on explicit negative sets makes them less effective in open-world chemical spaces where negatives are undefined. Unsupervised methods, such as DrugMetric (Li et al., 2024), utilize VAE-generated latent spaces combined with Gaussian Mixture Models (GMMs) to assess drug-likeness.

**One-class boundary models**   One-class classification (OCC) methods aim to define a boundary around a positive class (e.g., drugs) without relying on negatives, making them more suited for open-world problems. Techniques like OC-SVM (Schölkopf et al., 2001), Support Vector Data Description (SVDD) (Tax & Duin, 2004), and DeepSVDD (Ruff et al., 2018) seek to minimize the volume around positive samples, assuming the feature space to be fixed and optimal. However, their static nature and reliance on fixed feature spaces often lead to overly broad boundaries and high false-positive rates in expansive chemical spaces. Our proposed BOUNDr.E model addresses these limitations by dynamically refining both the boundary and the feature embedding space through an iterative EM-like process.

**Deep Multi-modal alignment**   Multi-modal alignment or multi-modal learning refers to the process of mapping diverse data modalities, such as image, text, video, and audio, into a unified embedding space that enables effective joint learning and generalization across various downstream tasks (Girdhar et al., 2023). A prominent example of multi-modal alignment is CLIP (Radford et al., 2021), which learns representations by aligning text descriptions with images through contrastive learning. Several frameworks extend CLIP-based multi-modal learning through finetuning (Goyal et al., 2023) or training-free approaches (Zhang et al., 2022) for more robust optimization. One such method is the recently proposed Geodesic Mixup (Oh et al., 2024), which ensures that multi-modal mixed samples lie on a geodesic path, preserving the structure of L2-normalized embeddings well mixed on a hypersphere.

In the biochemical domain, CLOOME (Sanchez-Fernandez et al., 2023) has been proposed to modify the CLIP loss through leave-one-out boosting with continuous modern Hopfield networks for chemical and bioassay image alignment. Recently, contrastive learning has also been actively integrated into the fields of drug-target interaction prediction (Ye et al., 2021) and element knowledge integration (Fang et al., 2022; 2023). Despite the significant advancements in multi-modal learning, there has not been an attempt to extend such concepts to align the knowledge graph embedding space with the structural embedding space through multimodal-alignment to construct an approved drug chemical space for drug-likeness prediction.

## 3   DEEP DRUG-LIKE BOUNDARY OPTIMIZATION

### 3.1   PROBLEM DEFINITION

We propose a new perspective on the problem of drug-likeness prediction as constructing a compact and adaptive one-class boundary $\mathcal{B}$ around drug-like compounds in a theoretically unbounded chemical space. Let this space of all chemical compounds be denoted as $\mathcal{X}_{\text{comp}}$, with subset $\mathcal{X}_{\text{drug}} \subset \mathcal{X}_{\text{comp}}$ representing drug-like compounds. The approved drug set $\mathcal{D}_{\text{drug}}$ represents a subset of the $\mathcal{X}_{\text{drug}}$, while compound set $\mathcal{D}_{\text{comp}}$ is a biased subset of $\mathcal{X}_{\text{comp}}$, where its small yet unknown portion are potential drugs that are to be rescued. (Appendix A.6)

Given the highly dispersed nature of drugs in chemical space and their approval based on both structure and biomedical knowledge, our framework combines these two modalities into a unified space for more accurate boundary construction, followed by iterative refinement of a hyperspherical boundary to capture drug-like compounds (Figure 2). The alignment of the embedding spaces and the boundary optimization are key to addressing the challenges posed by an unbounded chemical space and the absence of explicit negatives. Notations throughout this paper are organized in Appendix D.

**Knowledge-integrated multi-modal alignment**   The first step in our framework involves the alignment of two complementary drug spaces: the structural space $\mathcal{S}$, and the biomedical knowledge space $\mathcal{K}$. The key objective is to learn a structural encoder that can also map *non-drugs*, which

**Figure 3:** Comparison of contrastive losses using structural encoder $\mathcal{E}_\sigma$ and knowledge encoder $\mathcal{E}_\kappa$. CLIP enforces pair-wise similarity between knowledge graph and structural embeddings from a single entity. Softened CLIP allows pair-wise similarity between knowledge graph and structural embeddings to match the prior similarity matrix ($W_{\text{ATC}}$). $S$-Mix (and $K$-Mix), $KS$-Mix performs intra-modality interpolation.

have no corresponding biomedical information, into a biomedical context-enriched space. Each drug is represented by these two embeddings, which encode different aspects of drug-likeness: molecular structure and biomedical context. The goal is to unify these embeddings into a common latent space $\mathcal{U}$, where both structural and knowledge representations of drugs are aligned and consistent.

To achieve this, we introduce a *knowledge-integrated multi-modal mixup* strategy. This involves softening the CLIP loss (Radford et al., 2021) to encourage alignment between the two embedding spaces based on semantic drug similarities as prior knowledge. The alignment is further augmented with geodesic mixup (Oh et al., 2024), which ensures that the interpolated samples lie on a geodesic path between the embeddings. By employing this strategy, we create a unified embedding space that leverages the contexts from both molecular structure and biomedical knowledge, capturing a richer representation of drug-like properties.

**Drug-Like Boundary Optimization**   Once the multi-modal embeddings are aligned into the unified space $\mathcal{U}$, we define a hyperspherical boundary $\mathcal{B}$ in a latent space $\mathcal{Z}$, which is generated by an encoder $f_\theta : \mathcal{U} \to \mathcal{Z}$. This boundary is characterized by its center $\boldsymbol{c}$ and radius $r$, and the goal is to optimize $\mathcal{B}$ such that it encapsulates as many drug-like compounds as possible while minimizing the inclusion of non-drug-like compounds, leading to decreased in-boundary compound ratio $\rho$.

The optimization of $\mathcal{B}$ is an EM-like iterative process, with each iteration improving the compactness of the boundary and reducing the false-positive rate. The iterative refinement not only adapts the boundary $\mathcal{B}$ but also dynamically adjusts the embedding space $\mathcal{Z}$ through the encoder, making the model more flexible in handling the complex and heterogeneous nature of drug-likeness.

### 3.2 KNOWLEDGE-INTEGRATED MULTI-MODAL ALIGNMENT

We propose a multi-modal alignment approach, using a knowledge-guided soft CLIP loss augmented with geodesic mixup, to blend structural and biomedical embeddings into a unified space. This process ensures smooth transitions between the two distant domains by interpolating embeddings on a hypersphere (Figure 3).

We begin by aligning two key embedding spaces of: the biomedical knowledge graph embeddings $\boldsymbol{k}_{\text{drug}} \in \mathcal{K}$ (Bang et al., 2023) and the molecular structural embeddings $\boldsymbol{s}_{\text{drug}} \in \mathcal{S}$ (Morgan Fingerprint). This integration is crucial as it enriches drug representations by combining molecular structures with their biomedical contexts. We train two encoders: a knowledge encoder $\mathcal{E}_\kappa : \mathcal{K} \to \mathcal{U}$ and a structural encoder $\mathcal{E}_\sigma : \mathcal{S} \to \mathcal{U}$, where both map their respective embeddings to a unified latent space $\mathcal{U} \subset \mathbb{R}^d$. The details of the aligned spaces are explained in Appendix C.1.

#### 3.2.1 SOFTENED CLIP LOSS WITH ATC SIMILARITY

In this section, we propose a novel knowledge-integration strategy for multi-modal contrastive learning. We soften the CLIP loss (Radford et al., 2021) by incorporating semantic similarity (Jiang & Conrath, 1997) between drugs using Anatomic Therapeutic Chemical (ATC) classification. For a batch of data $D = \{(\boldsymbol{s}_i, \boldsymbol{k}_i)\}_{i=1}^{M}$, the original CLIP loss is given by:

$$C(\boldsymbol{s}, \boldsymbol{k}) = \frac{1}{M} \sum_{i=1}^{M} -\log \frac{\exp(\boldsymbol{s}_i \odot \boldsymbol{k}_i/\tau)}{\sum_{j=1}^{M} \exp(\boldsymbol{s}_i \odot \boldsymbol{k}_j/\tau)} \qquad \mathcal{L}_{\text{CLIP}} = \frac{1}{2}(C(\boldsymbol{s}, \boldsymbol{k}) + C(\boldsymbol{k}, \boldsymbol{s})) \qquad (1)$$

where $C(\boldsymbol{s}, \boldsymbol{k})$ is the contrastive loss for structural and knowledge embeddings, $\boldsymbol{s}_i \odot \boldsymbol{k}_i = \mathcal{E}_\sigma(\boldsymbol{s}) \cdot \mathcal{E}_\kappa(\boldsymbol{k})^T$ represents their dot-product similarity, and $\tau$ is the scaling temperature factor.

To introduce prior knowledge of drug similarities, we incorporate an ATC code similarity matrix $W_{\text{ATC}} = [w_{i,j}]$, where $w_{i,j} \in [0, 1]$ measures the semantic similarity between drugs $i$ and $j$. The modified loss incorporating $W_{\text{ATC}}$ becomes a weighted sum over the soft labels (Eq. 2):

$$C_{\text{soft}}(\boldsymbol{s}, \boldsymbol{k}, W_{\text{ATC}}) = \frac{1}{M} \sum_{i=1}^{M} \sum_{j=1}^{M} w_{i,j} \left( -\log \frac{\exp(\boldsymbol{s}_i \odot \boldsymbol{k}_j / \tau)}{\sum_{l=1}^{M} \exp(\boldsymbol{s}_i \odot \boldsymbol{k}_l / \tau)} \right) \tag{2}$$

$$\mathcal{L}_{\text{softCLIP}} = \frac{1}{2}(C_{\text{soft}}(\boldsymbol{s}, \boldsymbol{k}, W_{\text{ATC}}) + C_{\text{soft}}(\boldsymbol{k}, \boldsymbol{s}, W_{\text{ATC}})) \tag{3}$$

Here, instead of assuming a hard one-hot target where $w_{i,i} = 1$ and $w_{i,j} = 0$ for $i \neq j$ (as of the original CLIP loss), the soft labels $w_{i,j}$ encourage similarity of drug pair embeddings to match their semantic similarity. Details of ATC similarity computation are provided in Appendix C.2.

### 3.2.2 GEODESIC MIXUP FOR EMBEDDING ALIGNMENT

Several studies have reported the problem of "modality gap" in contrastive learning frameworks including CLIP (Wang & Isola, 2020; Liang et al., 2022). To further improve alignment of the two domains, we apply geodesic mixup (Oh et al., 2024) to interpolate between embeddings on a hypersphere, ensuring the points are aligned uniformly in the latent space. Given two points $\vec{a}$ and $\vec{b}$, the mixup is performed along the geodesic path:

$$m_\lambda(\vec{a}, \vec{b}) = \vec{a} \frac{\sin(\lambda\vartheta)}{\sin(\vartheta)} + \vec{b} \frac{\sin((1-\lambda)\vartheta)}{\sin(\vartheta)}$$

where $\vartheta = \cos^{-1}(\vec{a} \cdot \vec{b})$, and $\lambda \sim \text{Beta}(\alpha, \alpha)$. Within the batch of length $M$, geodesic mixup interpolates information from data indices $i$ and $i' = M - i$ with $\lambda$ and $1 - \lambda$ fraction, respectively. This allows smooth interpolation between data pairs, improving consistency within the latent space.

With our formulation, we introduce three forms of mixup (Figure 3):

**Structural Mix (S-Mix)**  Interpolates within the structural embedding space:

$$C_S(\boldsymbol{s}, \boldsymbol{k}) = \frac{1}{M} \sum_{i=1}^{M} -\lambda \log \frac{\exp(m_\lambda(\boldsymbol{s}_i, \boldsymbol{s}_{i'}) \odot \boldsymbol{k}_i / \tau)}{\sum_{j=1}^{M} \exp(\boldsymbol{s}_i \odot \boldsymbol{k}_j / \tau)} - (1 - \lambda) \log \frac{\exp(m_\lambda(\boldsymbol{s}_i, \boldsymbol{s}_{i'}) \odot \boldsymbol{k}_{i'} / \tau)}{\sum_{j=1}^{M} \exp(\boldsymbol{s}_i \odot \boldsymbol{k}_j / \tau)}$$

$$\mathcal{L}_{S\text{-Mix}} = \frac{1}{2}(C_S(\boldsymbol{s}, \boldsymbol{k}) + C_S(\boldsymbol{k}, \boldsymbol{s})) \tag{4}$$

**Knowledge Mix (K-Mix)**  Interpolates within the knowledge graph embedding space and has the same formula with S-Mix, except that it is applied to knowledge embedding-side.

$$\mathcal{L}_{K\text{-Mix}} = \frac{1}{2}(C_K(\boldsymbol{s}, \boldsymbol{k}) + C_K(\boldsymbol{k}, \boldsymbol{s})) \tag{5}$$

**Knowledge-Structural Mix (KS-Mix)**  Interpolates the knowledge and structural embeddings simultaneously:

$$C_{KS}(\boldsymbol{s}, \boldsymbol{k}) = \frac{1}{M} \sum_{i=1}^{M} -\log \frac{\exp(m_\lambda(\boldsymbol{s}_i, \boldsymbol{s}_{i'}) \odot m_\lambda(\boldsymbol{k}_i, \boldsymbol{k}_{i'}) / \tau)}{\sum_{j=1}^{M} \exp(\boldsymbol{s}_i \odot \boldsymbol{k}_j / \tau)}$$

$$\mathcal{L}_{KS\text{-Mix}} = \frac{1}{2}(C_{KS}(\boldsymbol{s}, \boldsymbol{k}) + C_{KS}(\boldsymbol{k}, \boldsymbol{s})) \tag{6}$$

These interpolations ensure the robustness of embedding space by smoothing the transitions between similar drugs and ensuring embeddings respect the L2-norm constraint of the hypersphere.

The final multi-modal alignment loss is a weighted sum:

$$\mathcal{L}_{\text{multi-modal}} = \lambda_{\text{softCLIP}} \mathcal{L}_{\text{softCLIP}} + \mathcal{L}_{S\text{-Mix}} + \mathcal{L}_{K\text{-Mix}} + \mathcal{L}_{KS\text{-Mix}} \tag{7}$$

We optimize the parameters of encoders $\mathcal{E}_\sigma$ and $\mathcal{E}_\kappa$ using the Adam optimizer (Kingma, 2014). The trained structure encoder $\mathcal{E}_\sigma$ is further utilized to project the chemical structural features into the unified embedding space $\mathcal{U}$ for downstream tasks including the drug-likeness boundary generation.

### 3.3 EM-LIKE ITERATIVE OPTIMIZATION OF DRUG-LIKENESS BOUNDARY

We formulate the boundary construction as an iterative process inspired by the Expectation-Maximization (EM) algorithm. The model adjusts the boundary parameters (a hypersphere with center $c \in \mathbb{R}^d$ and radius $r$) in the Expectation (E)-step, while refining the embedding space $\mathcal{Z}$ and its encoder $f_\theta$ during the Maximization (M)-step. This allows the boundary to evolve throughout training. The full algorithm is provided in Appendix A.1.

#### 3.3.1 EXPECTATION STEP: BOUNDARY UPDATE

In the E-step, we update $c$ and $r$ to enclose $\alpha \approx 100\%$ of drug-like compounds, keeping the embedding function $f_\theta$ fixed. Given the set of embedded drug compounds $z_{\text{drug}} = \{f(x; \theta^{(t)}) : x \in \mathcal{X}_{\text{drug}}\}$ at iteration time step $t$, the boundary parameters are updated as follows:

$$c^{(t+1)} = \frac{1}{|z_{\text{drug}}|} \sum_{z \in z_{\text{drug}}} z, \quad r^{(t+1)} = Q^\alpha_{z \in z_{\text{drug}}} \left( \|z - c^{(t+1)}\|_2 \right), \quad r^{(t+1)}_{\text{comp}} = \max_{z \in z_{\text{comp}}} \left( \|z - c^{(t+1)}\|_2 \right),$$

Here, $c^{(t+1)}$ is the center of the drug-like compounds at iteration $t+1$, $r^{(t+1)}$ is the radius of the smallest hypersphere containing $\alpha \approx 100\%$ of drug-like compounds defined by the $\alpha$-th percentile ($Q^\alpha$) of the set of distances $\|z - c^{(t+1)}\|_2$, and $r^{(t+1)}_{\text{comp}}$ captures the boundary of all compounds. Compounds outside the drug-like boundary are treated as pseudo-negatives in the next M-step:

$$\mathcal{X}_{\text{out}} := \{x \in \mathcal{X}_{\text{comp}} \mid d^{(t)}(x; \theta, c) > r^{(t+1)}\},$$

where $d^{(t)}(x; \theta, c) = \|f(x; \theta^{(t)}) - c^{(t+1)}\|_2$ is the Euclidean distance from the boundary center.

#### 3.3.2 MAXIMIZATION STEP: EMBEDDING FUNCTION UPDATE

In the M-step, we optimize the embedding function $f_\theta : \mathcal{U} \to \mathcal{Z}$ with parameters $\theta$ to reduce the inclusion of non-drug-like compounds inside the boundary while keeping drug-like compounds near the center. The total loss function consists of two metric terms:

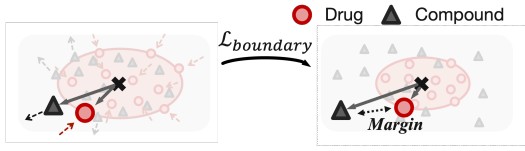

**Figure 4:** Latent space optimization during M-step. The margin between drug and compound are increased.

1. **Drug loss** $\mathcal{L}_{\text{drug}}$, which encourages drugs to be located closer the center of the boundary:
$$\mathcal{L}_{\text{drug}}(\theta) = \sum_{x \in \mathcal{X}_{\text{drug}}} d_t(x; \theta, c)$$

2. **Out-boundary loss** $\mathcal{L}_{\text{out}}$, which pushes non-drugs labeled as pseudo-negatives during the E-step to the compound space boundary:
$$\mathcal{L}_{\text{out}}(\theta) = \sum_{x \in \mathcal{X}_{\text{out}}} \max \left( r^{(t+1)}_{\text{comp}} - d_t(x; \theta, c), 0 \right)$$

The loss terms can be interpreted as reducing/increasing the samples' distances $d(x)$ to 0 and $r^{(t+1)}_{\text{comp}}$ for drugs and out-boundary compounds, respectively. We then combine the two loss terms to yield a total loss described as:

$$\mathcal{L}_{\text{boundary}}(\theta) = \mathcal{L}_{\text{drug}}(\theta) + \lambda_{\text{out}} \cdot \mathcal{L}_{\text{out}}(\theta) \tag{8}$$

where $\lambda_{\text{out}}$ controls the strength of the out-boundary penalty. This loss iteratively improves the separation between drug-like and non-drug-like compounds, increasing the margin $\sum_{x_{\text{drug}} \in \mathcal{X}_{\text{drug}}} \sum_{x_{\text{comp}} \in \mathcal{X}_{\text{comp}}} d(x_{\text{comp}}) - d(x_{\text{drug}})$ between drugs and compounds (Figure 4).

We show that minimizing the metric loss function (Eq. 8) leads to a boundary $\mathcal{B}$ that encapsulates drug-like compounds while excluding non-drug-like ones, improving drug-likeness prediction accuracy:

**Theorem 1** (Reduction of in-boundary non-drugs). *Optimizing a neural network encoder with the distance-based loss function reduces the number of non-drugs inside the boundary $|\mathcal{X}_{\text{in-boundary}}|$ between two successive steps $t_1 < t_2$, where $\mathcal{L}^{(t_1)}_{drug} > \mathcal{L}^{(t_2)}_{drug}$ and $\mathcal{L}^{(t_1)}_{out} > \mathcal{L}^{(t_2)}_{out}$.*

The proof is provided in Appendix A.2.

**Table 1:** Drug-like compound identification performance with time-split setting. Mean and standard deviation of 10 fold CV are provided. Best performance and its comparable results (paired t-test $p < 0.05$) are marked in bold, and second-best are underlined. (Avg: Average)

| | F1 ($\uparrow$) | IDR ($\uparrow$) | ICR ($\downarrow$) | AUROC ($\uparrow$) | Avg. Precision ($\uparrow$) |
|---|---|---|---|---|---|
| **FP-SVM** (Boser et al., 1992) | 0.665 (0.0126) | 0.823 (0.0111) | 0.067 (0.0052) | 0.963 (0.0021) | 0.724 (0.0174) |
| **FP-XGB** (Chen & Guestrin, 2016) | 0.692 (0.0141) | 0.815 (0.0205) | 0.055 (0.0048) | 0.966 (0.0026) | 0.775 (0.0213) |
| **FP-OCSVM** (Schölkopf et al., 2001) | 0.090 (0.0025) | 0.274 (0.0000) | 0.489 (0.0101) | 0.331 (0.0030) | 0.148 (0.0022) |
| **FP-DeepSVDD** (Ruff et al., 2018) | 0.166 (0.0087) | 0.834 (0.0350) | 0.840 (0.0381) | 0.494 (0.0532) | 0.097 (0.0157) |
| **FP-nnPU** (Kiryo et al., 2017) | 0.608 (0.0239) | 0.789 (0.0367) | 0.083 (0.0081) | 0.944 (0.0049) | 0.706 (0.0261) |
| **FP-PU with NN** (Li & Liu, 2003) | 0.634 (0.0224) | 0.791 (0.0296) | 0.072 (0.0079) | 0.949 (0.0045) | 0.720 (0.0214) |
| **DrugMetric** (Li et al., 2024)* | 0.170 (0.0319) | 0.767 (0.1271) | 0.760 (0.2028) | *N/A* | *N/A* |
| **D-GCAN** (Sun et al., 2022) | 0.669 (0.1770) | **0.942 (0.0337)** | 0.160 (0.2808) | 0.918 (0.1396) | 0.613 (0.1874) |
| **DeepDL** (Lee et al., 2022) | 0.740 (0.0584) | 0.888 (0.0546) | 0.054 (0.0225) | **0.979 (0.0114)** | 0.886 (0.0374) |
| **BOUNDR.E** | 0.826 (0.0486) | 0.781 (0.0326) | 0.012 (0.0086) | 0.973 (0.0075) | 0.877 (0.0419) |
| **BOUNDR.E$_{\text{MULT}}$** | **0.846 (0.0165)** | 0.799 (0.0184) | **0.009 (0.0031)** | 0.978 (0.0029) | **0.908 (0.0096)** |

*DrugMetric's GMM classifier fails to provide prediction probabilities for AUROC and Average Precision calculation

Finally, convergence is determined by the in-boundary compound ratio $\rho_t = |\mathcal{X}^{(t)}_{\text{in-boundary}}|/|\mathcal{X}_{\text{comp}}|$. The algorithm stops when the change in $\rho_t$ between iterations is smaller than a threshold $\epsilon$: $|\rho_{t+1} - \rho_t| < \epsilon$ for $n_{\text{patience}}$ consecutive iterations. In addition, we have applied a multi-initialization technique to avoid the sensitivity to initialization of the EM-like models, as an extension of our model as BOUNDR.E$_{\text{MULT}}$, further detailed and discussed in Appendix A.5.

Overall, our EM-like framework iteratively refines the boundary and embedding space, resulting in a compact boundary that effectively excludes non-drug-like compounds. The knowledge-aligned embeddings of $\mathcal{U}$ further enhances the model's drug-likeness prediction capabilities.

# 4 EXPERIMENTS

## 4.1 SETUP

**Dataset** Approved drug data is sourced from DrugBank v5.1.12 (Knox et al., 2024) and removed all withdrawn drugs. 100k non-drug compounds are sampled from ZINC20 (Irwin et al., 2020), limited to clean, annotated entries. We evaluate our model on drug-compound identification under two split scenarios: *scaffold*-based and *time*-based. The scaffold-based split ensures the molecular scaffolds in train, validation, and test sets are mutually exclusive, using the using the Bemis-Murcko scaffolds (Bemis & Murcko, 1996). This evaluation scheme is applied to measure the models' generalizablilty when an unseen scaffold compound is input, where approved drugs exist extremely sparse in the scaffold space (Appendix C.3.1). In the time-based split, drugs are partitioned based on their approval year (e.g., drugs approved post-2011 are in the test set), to reflect the temporal evolution of approved drug properties (Appendix C.3.2). Both split strategies aim to reflect real-world scenarios, where drug discovery must generalize to unseen chemical scaffolds. The data splitting strategy are detailed in Appendix C.3.

**Baselines** We compare our model to established drug-likeness prediction models: DeepDL (Lee et al., 2022), D-GCAN (Sun et al., 2022), and DrugMetric (Li et al., 2024), as well as several general machine learning classifiers: SVM (Boser et al., 1992), XGBoost (Chen & Guestrin, 2016), Naive PU algorithm by Li & Liu (2003) implemented with neural network, nnPU (Kiryo et al., 2017), OC-SVM (Schölkopf et al., 2001), and DeepSVDD (Ruff et al., 2018). Each general baseline is provided with molecular fingerprints as input features. Implementation details are provided in Appendix C.6.

## 4.2 DRUG-COMPOUND IDENTIFICATION PERFORMANCES

We evaluate performance of models in distinguishing approved drugs from ZINC compounds under both split strategies—time-based split and scaffold-based split. We report the results using F1-score, MCC, and two metrics: In-boundary Drug Ratio (IDR) and In-boundary Compound Ratio (ICR):

$$\text{IDR} = \frac{|\text{Drugs in boundary}|}{|\text{Total drugs in test set}|} = \text{TPR}, \quad \text{ICR} = \frac{|\text{Compounds in boundary}|}{|\text{Total compounds in test set}|} = \text{FPR}.$$

IDR, equivalent of True Positive Rate (TPR), reflects how well the boundary captures drug-like compounds, while ICR, representing False Positive Rate (FPR), measures how well non-drug compounds

**Table 2:** Cross-dataset evaluation of drug-like compound identification performance on scaffold-split setting, trained on PubChem/ChEMBL and evaluated with ZINC20 compounds. Mean and standard deviation of 10 fold CV are provided. Best and its comparable performances (paired t-test p < 0.05) are marked in bold.

| Train set | PubChem + DrugBank | | | ChEMBL + DrugBank | | |
|---|---|---|---|---|---|---|
| | F1 (↑) | Average Precision (↑) | AUROC (↑) | F1 (↑) | Average Precision (↑) | AUROC (↑) |
| FP-SVM | 0.268 (0.0194) | 0.334 (0.1912) | 0.795 (0.0759) | 0.371 (0.0519) | **0.494 (0.1982)** | 0.819 (0.0768) |
| FP-XGB | 0.254 (0.0209) | 0.320 (0.1181) | 0.773 (0.0741) | 0.358 (0.0589) | 0.469 (0.1839) | 0.814 (0.0784) |
| FP-OCSVM | 0.179 (0.0582) | 0.366 (0.2717) | 0.576 (0.1949) | 0.179 (0.0582) | 0.366 (0.2717) | 0.576 (0.1949) |
| FP-SVDD | 0.151 (0.0033) | 0.055 (0.0019) | 0.235 (0.0173) | 0.151 (0.0033) | 0.055 (0.0019) | 0.235 (0.0173) |
| FP-DeepSVDD | 0.147 (0.0294) | 0.080 (0.0146) | 0.415 (0.1224) | 0.147 (0.0294) | 0.080 (0.0146) | 0.415 (0.1224) |
| FP-nnPU | 0.244 (0.0182) | 0.240 (0.0816) | 0.749 (0.0556) | 0.327 (0.0525) | 0.380 (0.1999) | 0.778 (0.0812) |
| FP-PU with NN | 0.241 (0.0265) | 0.228 (0.0556) | 0.702 (0.0560) | 0.311 (0.0495) | 0.396 (0.1701) | 0.778 (0.0874) |
| DeepDL | 0.170 (0.0199) | 0.092 (0.0112) | 0.590 (0.0233) | 0.195 (0.0389) | 0.102 (0.0196) | 0.612 (0.0686) |
| D-GCAN | 0.213 (0.0232) | 0.135 (0.0153) | 0.685 (0.0436) | 0.314 (0.0620) | 0.211 (0.0601) | 0.737 (0.1076) |
| BOUNDR.E | 0.496 (0.0287) | 0.444 (0.0303) | 0.873 (0.0167) | 0.513 (0.0451) | 0.435 (0.0889) | 0.869 (0.0258) |
| BOUNDR.E$_{MULT}$ | **0.501 (0.0232)** | **0.460 (0.0380)** | **0.875 (0.0157)** | **0.546 (0.0406)** | 0.484 (0.0729) | **0.876 (0.0267)** |

are excluded. We also report the AUROC metric to report the models' capabilities in balancing the trade-off between TPR and FPR. In addition, we also report Average Precision (AP), Recall@k and Precision@k to evaluate the quality of recommended compounds (Appendix E.1).

As a result, our model consistently outperforms binary classifiers, PU learners, and one-class classification models across both split settings. For the time-based split (Table 1), our model achieves the highest F1, AUROC, and AP, demonstrating its ability to adapt to unseen drug-like compounds. Results for the scaffold-based split (Appendix E.2) further confirm the robustness of our approach, highlighting its capacity to generalize across diverse molecular structures.

**Cross-dataset evaluation** We further tested generalizability through cross-compound dataset evaluation. Models are first trained on PubChem or ChEMBL compound sets then tested with the ZINC compounds, with the drug set (DrugBank) and its split setting fixed. As a result, binary classifiers and PU-learning frameworks show heavy decline in performances whereas one-class classsifers show no effect. BounDr.E demonstrate only moderate decline in both scaffold-based (Table 2) and time-based (Appendix E.3) splits. This result shows the generalizability of our one-class boundary approach by not rely on the non-drug set. Experimental details are available in Appendix C.4.

### 4.3 ZERO-SHOT TOXIC COMPOUND IDENTIFICATION

To test our model's capacity to filter out potentially toxic compounds, we performed a zero-shot evaluation on toxic compounds from Drug-Bank's withdrawn drug list and toxic compound sets (hepatotoxic, cardiotoxic, and carcinogenic compounds) (Wu et al., 2023).

As shown in Table 3, our model demonstrates lower false-positive rate compared to baseline models,

**Table 3:** False-positive rate of toxic compound groups. Lowest and its comparable results (paired t-test p < 0.05) are marked in bold.

| | Withdrawn | Hepatotoxic | Cardiotoxic | Carcinogenic |
|---|---|---|---|---|
| FP-XGB | 0.96 (0.003) | 0.96 (0.003) | 0.85 (0.010) | 0.93 (0.010) |
| FP-OCSVM | 0.69 (0.002) | **0.53 (0.003)** | 0.25 (0.006) | 0.86 (0.001) |
| FP-nnPU | 0.95 (0.009) | 0.94 (0.007) | 0.87 (0.028) | 0.86 (0.017) |
| DrugMetric* | *N/A* | 0.77 (0.073) | 0.76 (0.118) | 0.82 (0.087) |
| DGCAN | 0.91 (0.020) | 0.85 (0.023) | 0.88 (0.045) | 0.95 (0.017) |
| DeepDL | 0.91 (0.016) | 0.92 (0.018) | 0.85 (0.042) | 0.84 (0.025) |
| BOUNDR.E | 0.52 (0.041) | 0.54 (0.028) | 0.20 (0.019) | 0.20 (0.043) |
| BOUNDR.E$_{MULT}$ | **0.51 (0.014)** | 0.54 (0.009) | **0.20 (0.009)** | **0.19 (0.014)** |

*DrugMetric fails to infer scaffolds not present in approved drug and ZINC datasets

consistently identifying toxic compounds from diverse categories as out of drug boundary. Furthermore, error analysis on the withdrawn drugs reveal that among the 52% false-positive, most of them are withdrawn from some regions yet approved in others. These results indicates that our boundary, along with its integrated biomedical contexts, can effectively generalize to compounds with toxic properties, offering a promising tool for early-stage toxicity filtering. Full table of baseline model performances are provided in Appendix E.4.

### 4.4 EMBEDDING SPACE VISUALIZATION

Figure 5 displays the evolution of our embedding space as the EM-like boundary optimization proceeds. It is easy to spot that the compounds from ZINC database are being pushed out of the boundary as FDA-approved drugs form more compact space as training epochs increase. The zoomed-in boxes of each epoch further visualizes how the density of ZINC-compounds decreases as the embedding space is optimized. This visualization effectively demonstrates our model's ability to iteratively refine the embedding space, making it increasingly more drug-focused over time.

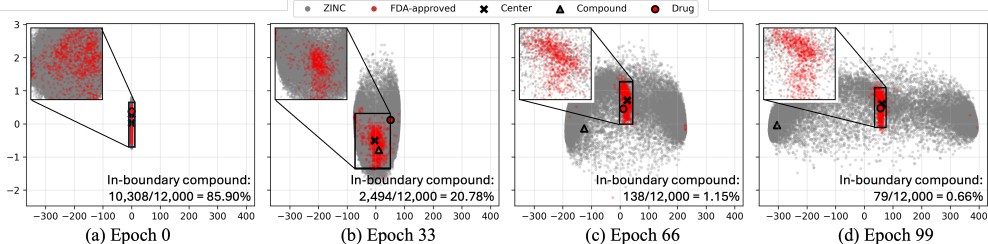

**Figure 5:** PCA visualization of knowledge-aligned embedding space and latent space at each epoch of boundary optimization. Box on the upper-left corner displays the space within the drug-like boundary based on PC1 and PC2. Red circle and gray triangle display the movement of drug and zinc compound samples respectively, as training proceeds.

**Table 4:** Drug-like compound identification with EM-like boundary optimization on embedding space aligned with different alignment methods. Best and its comparable results (paired t-test p < 0.05) are marked in bold.

| Alignment method | F1 (↑) | ICR (↓) |
|---|---|---|
| **No Alignment** (only FP) | 0.54 (0.032) | 0.057 (0.0161) |
| **Manifold Alignment** | 0.40 (0.045) | **0.009 (0.0055)** |
| **CLIP** | 0.59 (0.022) | 0.025 (0.0133) |
| **Geodesic Mixup** | 0.69 (0.045) | 0.025 (0.0133) |
| **Ours - softCLIP** | 0.73 (0.037) | **0.018 (0.0066)** |
| **Ours** | **0.83 (0.049)** | **0.012 (0.0086)** |

**Table 5:** Drug-like compound identification with different classifiers on knowledge-aligned space. Best and its comparable results (paired t-test p < 0.05) are marked in bold.

| Aligned space | F1 (↑) | ICR (↓) |
|---|---|---|
| + MLP | 0.77 (0.020) | 0.046 (0.0053) |
| + SVM | **0.86 (0.012)** | 0.050 (0.0050) |
| + XGB | 0.75 (0.012) | 0.019 (0.0023) |
| + naive PU | 0.82 (0.011) | 0.031 (0.0029) |
| + DeepSVDD | 0.32 (0.079) | 0.351 (0.1148) |
| + Ours − EM | 0.44 (0.162) | 0.259 (0.1931) |
| + Ours | 0.83 (0.049) | **0.012 (0.0086)** |

## 4.5 ABLATION STUDIES

**Effect of multi-modal alignment with softened CLIP loss** We compared our softened CLIP loss with alternative alignment strategies, including CLIP (Radford et al., 2021), Geodesic Mixup, naive manifold alignment (Ham et al., 2005), and unaligned space (i.e., molecular fingerprints) (Table 4). Our proposed method significantly improves boundary quality due to the enriched representation that aligns molecular structure with biomedical knowledge. The resulting embedding space produces a tighter drug boundary, leading to improved drug-like compound identification performances. The full ablation study results including each component of S-Mix, K-Mix and KS-Mix are provided in Appendix E.5, which also support the utility of integrating all the components.

**Effect of EM-like optimization** We evaluated the advantage of our EM-like boundary optimization against traditional binary classifiers, PU learners, and one-class models (Table 5). Our model achieves the lowest ICR (or FPR), showcasing the strength of iterative boundary refinement, which iteratively increases the out-boundary compounds (Appendix E.5.1). Figure 6 shows the robustness of our method under increasing compound-to-drug ratios (from 1:1 to 1:100), where our model maintains performance compared to baselines, as the non-drug compounds vastly outnumber drugs.

These ablations confirm the complementary nature of multi-modal alignment and boundary optimization in improving drug-likeness prediction.

## 4.6 DISTANCE DISTRIBUTION OF COMPOUNDS IN DIVERSE STAGES

To validate the effectiveness of our distance metric, we analyze the drug-likeness scores for six compound sets spanning different stages of drug discovery: AI-generated compounds (TargetDiff (Guan et al., 2023) and MOOD (Lee et al., 2023)), investigational compounds and world-approved drugs (ZINC20 (Irwin et al., 2020)), withdrawn drugs, and FDA-approved drugs (DrugBank (Knox et al., 2024)).

Figure 7 shows a clear progression, with compounds moving closer to the center of the drug boundary as they advance through the drug development pipeline. The result reflects the increasing likelihood of drug-like properties as a compound matures from early AI-generated candidates to approved drugs. Our model effectively differentiates AI-generated molecules from investigational and approved drugs. This ability to rank candidates based on drug-likeness provides a valuable tool for *in silico* screening, accelerating early-stage compound prioritization.

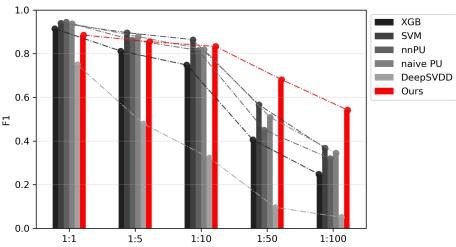

**Figure 6:** Change of F1 score with the decrease in drug-compound ratio of the test set.

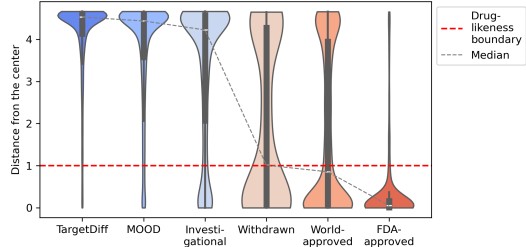

**Figure 7:** Distribution of drug-like scores of compound sets in different drug discovery stages.

### 4.7 APPLICATION TO RATIONAL TARGETED DRUG DISCOVERY PIPELINES

In this section, we demonstrate the utility of our model for initial screening and its potential real-world impact in target-based drug discovery pipeline. Utilizing three well-known anti-cancer targets, BCR-ABL, EGFR and CDK6, we first generated 10k anti-cancer compounds with pocket-aware generative model (Guan et al. (2023)). Then, we

**Table 6:** Number of filtered compounds by different filters.

| Filtering Method | BCR-ABL | EGFR | CDK6 |
|---|---|---|---|
| **Total Generated** | 10,543 (100%) | 12,550 (100%) | 11,496 (100%) |
| **PAINS filter** | 10,078 (95.7%) | 11,878 (94.6%) | 10,996 (95.6%) |
| **Rule of Five** | 4,997 (47.5%) | 6,520 (52.0%) | 5,782 (50.3%) |
| **Predicted IC50** | 2,786 (26.5%) | 1,018 (8.10%) | 4,734 (41.2%) |
| **BounDr.E** | 300 (2.8%) | 374 (3.00%) | 264 (2.3%) |
| **All filters** | **38 (0.36%)** | **17 (0.15%)** | **47 (0.40%)** |

compared the filtering capability of our approach with property-based filters, detailed in Appendix E.6.1. The results demonstrate the outstanding filtering ratio of our approach compared to others (Table 6). Additionally, by initially applying BounDr.E followed by all other filters yielded approximately 0.3% of screened compounds, a very practical number for downstream wet lab validations. This outcome illustrates how BounDr.E optimizes the workflow by minimizing the initial candidate pool for downstream experimental validation and simultaneously saving computational resources.

Furthermore, the filtered compound list yield a more distant distribution of compounds from the initially generated molecules, showing more desirable traditional measures in QED, Rule-of-five and Synthetic Accessibility Scores (SAS), along with higher probability of identifying existing approved drugs; imatinib (BCR), erlotinib (EGFR) and ribociclib (CDK6) (Appendix E.6.2).

Lastly, to test our model's capabilities to be adapted for cancer drug discovery, we trained our model on a narrower training set containing only cancer drugs (Appendix E.7). This anti-cancer variant, while showing strictness for toxic compounds, provided a broader boundary for generated anti-cancer compounds, showcasing our model's potentials to be tailored for specific therapeutic area.

## 5 CONCLUSION AND FUTURE WORKS

In this work, we introduced BOUNDR.E, a framework for drug-likeness prediction that combines knowledge-aligned embeddings with EM-like one-class boundary optimization. By leveraging structural and biomedical knowledge through a softened CLIP loss, BOUNDR.E creates a robust multi-modal embedding space. Our experiments show that BOUNDR.E consistently outperforms state-of-the-art models, excelling at identifying drug-like compounds while effectively filtering out toxic molecules, with case studies demonstrating its utility as initial screen of drug candidates.

Several opportunities for improvement remain in our framework. The EM-like strategy still requires solid approaches for reaching global optima, and lower reliance to initialization points. Further experimental validation of the screened compounds, including efficacy, toxicity and PK/PD profiles, may provide more convincing results on the utility data-driven drug filters in drug discovery endeavours. In particular, the applicability of our model to specific therapeutic area can be further elaborated. Nonetheless, we believe our model is a promising complementary solution for prioritizing drug-like compounds in early-stage development for efficiency in drug discovery.

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

# A    DETAILS IN EM-LIKE BOUNDARY OPTIMIZATION

## A.1    ALGORITHM OF EM-LIKE BOUNDARY OPTIMIZATION

---

**Algorithm 1** EM-like Training for Drug Boundary Optimization

---

**Require:** Dataset $\mathcal{X} = \{\boldsymbol{x}_i\}_{i=1}^N = \mathcal{X}_{\text{drug}} + \mathcal{X}_{\text{comp}}$,    Learning rate $\eta_\theta$ ,    Convergence tolerance $\epsilon$
**Ensure:** Optimized embedding space parameters $\theta^*$ and boundary parameters $\boldsymbol{c}^*, r^*$

1: Initialize neural network parameters $\theta^{(0)}$, boundary parameters $c^{(0)}, r^{(0)}$
2: $\mathcal{X} \leftarrow \mathcal{E}_\sigma(\mathcal{X})$ where $\mathcal{E}_\sigma$ is pretrained multi-modal structure encoder
3: $\rho^{(0)} = \frac{|\mathcal{X}_{\text{in-boundary}}^{(0)}|}{|\mathcal{X}_{\text{comp}}|}$    where    $\mathcal{X}_{\text{in-boundary}}^{(0)} := \left\{ \boldsymbol{x} \mid \|f(\boldsymbol{x}_{\text{comp}}; \theta^{(0)}) - \boldsymbol{c}^{(0)}\|_2 \leq r^{(0)} \right\}$
4: Set $t = 0$
5: **while** $|\rho_{t+1} - \rho_t| \geq \epsilon$ **do**
6:     **E-step** (Boundary update):
7:         $\boldsymbol{z}_{\text{drug}} \leftarrow f(\boldsymbol{x}_{\text{drug}}; \theta^{(t)})$
8:         $\boldsymbol{c}^{(t+1)} \leftarrow \frac{1}{|\boldsymbol{z}_{\text{drug}}|} \sum \boldsymbol{z}_{\text{drug}}$
9:         $r^{(t+1)} \leftarrow \max\left(\|\boldsymbol{z}_{\text{drug}} - \boldsymbol{c}^{(t+1)}\|_2\right),    r_{\text{comp}}^{(t+1)} \leftarrow \max\left(\|z_{\text{comp}} - \boldsymbol{c}^{(t+1)}\|_2\right)$
10:        Identify $\mathcal{X}_{\text{out}}$
11:    **M-step** (Embedding function update):
12:        $\mathcal{L}_{\text{boundary}}(\theta^{(t)}) \leftarrow \mathcal{L}_{\text{drug}}(\theta^{(t)}, \boldsymbol{c}^{(t+1)}, r^{(t+1)}) + \lambda_{\text{out}} \cdot \mathcal{L}_{\text{out}}(\theta^{(t)}, \boldsymbol{c}^{(t+1)}, r^{(t+1)})$
13:        $\theta^{(t+1)} \leftarrow \theta^{(t)} - \eta_\theta \cdot \text{Adam}\left(\nabla_\theta \mathcal{L}(\theta^{(t)}, \boldsymbol{c}^{(t+1)}, r^{(t+1)})\right)$
14:        $\rho^{(t+1)} \leftarrow \frac{|\mathcal{X}_{\text{in-boundary}}^{(t)}|}{|\mathcal{X}_{\text{comp}}|}$ where $\mathcal{X}_{\text{in-boundary}}^{(t)} := \left\{ \boldsymbol{x} \mid \|f(\boldsymbol{x}_{\text{comp}}; \theta^{(t+1)}) - \boldsymbol{c}^{(t+1)}\|_2 \leq r^{(t+1)} \right\}$
15:    Increment $t \leftarrow t + 1$
16: **end while**
17: **Return** Optimized parameters $\theta^*, \boldsymbol{c}^*, r^*$

---

## A.2    PROOF OF THEOREM 1

To recap, the M-step of the EM-like iterative optimizes the latent space with the following loss terms:

$$\mathcal{L}_{\text{drug}}(\theta) = \sum_{\boldsymbol{x} \in \mathcal{X}_{\text{drug}}} d_t(\boldsymbol{x}; \theta, \boldsymbol{c}) \tag{9}$$

$$\mathcal{L}_{\text{out}}(\theta) = \sum_{\boldsymbol{x} \in \mathcal{X}_{\text{out}}} \max\left(r_{\text{comp}}^{(t+1)} - d_t(\boldsymbol{x}; \theta, \boldsymbol{c}), 0\right) \tag{10}$$

$$\mathcal{L}_{\text{boundary}}(\theta) = \mathcal{L}_{\text{drug}}(\theta) + \lambda_{\text{out}} \cdot \mathcal{L}_{\text{out}}(\theta) \tag{11}$$

where $d_t(\boldsymbol{x}; \theta, \boldsymbol{c}) = \|f(\boldsymbol{x}; \theta^{(t)}) - \boldsymbol{c}^{(t+1)}\|_2$ is the Euclidean distance of samples from the drug center, and $\lambda_{\text{out}}$ is a hyperparameter controlling the strength of the out-boundary penalty. The loss terms can be interpreted as reducing/increasing the samples' distances $d(x)$ to 0 and $r_{\text{comp}}^{(t+1)}$ for drugs and out-boundary compounds, respectively.

**Theorem 1** (Reduction of In-boundary Non-drugs). *Optimizing a neural network encoder with Euclidean distance loss to regress distance of non-drugs toward a radius of $r_{comp}$ and drugs toward 0 leads to a decrease in the number of non-drugs in boundary $|\mathcal{X}_{in\text{-}boundary}|$ between two successive time steps $t_1 < t_2$ where $\mathcal{L}_{drug}^{(t_1)} > \mathcal{L}_{drug}^{(t_2)}$ and $\mathcal{L}_{out}^{(t_1)} > \mathcal{L}_{out}^{(t_2)}$.*

To prove this, we will break down the proof to show that the decreasing nature of $r$ and the inconsistency that arises if the number of points inside an arbitrary threshold $\nu$ increases during the optimization of the Euclidean distance-based loss.

**Proposition 1** (Shrinkage of $r$): *As the optimization of the Euclidean distance loss proceeds over time, the drug boundary radius $r$, defined as the maximum distance of drug-like points from the center $\boldsymbol{c}$, decreases.*

*Proof:* Let $\mathcal{X}_{\text{drug}} \subset \mathbb{R}^d$ denote the set of drug-like compounds and $\boldsymbol{c} \in \mathbb{R}^d$ be a center point. The drug loss function $\mathcal{L}_{\text{drug}}$ (Eq. 9) is given by:

$$\mathcal{L}_{\text{drug}} = \sum_{\boldsymbol{x} \in \mathcal{X}_{\text{drug}}} d(\boldsymbol{x}) = \sum_{\boldsymbol{x} \in \mathcal{X}_{\text{drug}}} \|f(\boldsymbol{x};\theta) - \boldsymbol{c}\|_2,$$

where $d(\boldsymbol{x})$ represents the Euclidean distance between point $f(\boldsymbol{x};\theta)$ and $\boldsymbol{c}$.

The objective of the optimization process is to minimize $\mathcal{X}_{\text{drug}}$ by penalizing larger distances more severely with the square operation, while attracting points further away from $\boldsymbol{c}$ more strongly, and since the Euclidean distance norm is a strictly convex function, any reductions in the loss $\mathcal{X}_{\text{drug}}$ implies a reduction in the distance $\|f(\boldsymbol{x};\theta) - \boldsymbol{c}\|_2$ for each $\boldsymbol{x} \in \mathcal{X}_{\text{drug}}$.

Thus the furthest point $\boldsymbol{x}* \in \mathcal{X}_{\text{drug}}$, which determines $r$, experiences a decrease in distance from $\boldsymbol{c}$ as the loss function decreases, and therefore, as $\mathcal{L}_{\text{drug}}$ is minimized, $r$ decreases as the optimization progresses. $\qquad\square$

**Lemma 1** (Impact of Compounds Inside $\nu$ to $\mathcal{L}_{\text{out}}$): *The contribution to the out-boundary loss $\mathcal{L}_{\text{out}}$ from points $\boldsymbol{x}$ with $d(\boldsymbol{x}) < \nu$ is greater than the contribution from points with $d(\boldsymbol{x}) \geq \nu$.*

*Proof:* The out-boundary loss $\mathcal{L}_{\text{out}}$ (Eq. 10) is given by:

$$\mathcal{L}_{\text{out}}(\theta) = \sum_{\boldsymbol{x} \in \mathcal{X}_{\text{out}}} \max\left(r_{\text{comp}} - d(\boldsymbol{x}), 0\right),$$

where $d(\boldsymbol{x})$ represents the Euclidean distance between the compound $\boldsymbol{x}$ and the center $\boldsymbol{c}$. Considering the loss contribution of a point $\boldsymbol{x} \in \mathcal{X}_{\text{out}}$ with distance $d(\boldsymbol{x})$, the individual contribution to the loss for this point is

$$\mathcal{L}_{\text{out},\boldsymbol{x}} = \max\left(r_{\text{comp}} - d(\boldsymbol{x}), 0\right).$$

So, for points $\boldsymbol{x}$ such that $\boldsymbol{x}$ with $d(\boldsymbol{x}) < \nu$ with given an arbitrary threshold radius, we have

$$r_{\text{comp}} - d(\boldsymbol{x}) > r_{\text{comp}} - \nu.$$

On the other hand, for points where $d(\boldsymbol{x}) \leq \nu$, we have

$$r_{\text{comp}} - d(\boldsymbol{x}) \leq r_{\text{comp}} - \nu.$$

Since the out-boundary loss $\mathcal{L}_{\text{out}}$ is the sum of the individual contributions for each point in $\mathcal{X}_{\text{comp}}$, increasing the number of points for which $d(\boldsymbol{x}) < \nu$ will increase the overall loss $\mathcal{L}_{\text{out}}$ more than increasing the number of points with $d(\boldsymbol{x}) \geq \nu$. Therefore, the points with the threshold radius $\nu$ contribute more to the loss than those outside. Thus, the contributions of points with $d(\boldsymbol{x}) < \nu$ is greater than that of points with $d(\boldsymbol{x}) \geq \nu$. $\qquad\square$

**Proposition 2** (Decrease in Points Inside $\nu$): If the out-boundary loss $\mathcal{L}_{\text{out}}$ decreases with each iteration step, that is, $\mathcal{L}_{\text{out}}^{(t_2)} < \mathcal{L}_{\text{out}}^{(t_1)}$, then the number of points $\boldsymbol{x}$ such that $d(\boldsymbol{x}) < \nu$ decrease between steps $t_1$ and $t_2$.

*Proof:* For given iterative steps $t_1$ and $t_2$, assume that the number of points $\boldsymbol{x}$ such that $d(\boldsymbol{x}) < \nu$ increases between iterative steps, meaning that more points fall within the threshold $\nu$ at step $t_2$ than at step $t_1$. From **Lemma 1**, we know that the contribution to the out-boundary loss $\mathcal{L}_{\text{out}}$ from points within the threshold $\nu$ is greater than the contribution from points outside $\nu$. Specifically, for any point $\boldsymbol{x}$ where $d(\boldsymbol{x}) < \nu$, the contribution to the loss satisfies

$$r_{\text{comp}} - d(\boldsymbol{x}) > r_{\text{comp}} - \nu.$$

Thus, if the number of points $\boldsymbol{x}$ such that $d(\boldsymbol{x}) < \nu$ increase at step $t_2$, the out-boundary loss $\mathcal{L}_{\text{out}}$ at step $t_2$ should increase relative to its value at step $t_1$, since the points inside $\nu$ contribution more to the loss. This would imply that the loss at step $t_2$, $\mathcal{L}_{\text{out}}^{(t_2)}$, is greater than or equal to the loss at step $t_1$, $\mathcal{L}_{\text{out}}^{(t_1)}$.

However, this contradicts the assumption that $\mathcal{L}_{\text{out}}^{(t_2)} < \mathcal{L}_{\text{out}}^{(t_1)}$, i.e., the loss decreases over steps. Therefore, our assumption that the number of points with $d(\boldsymbol{x}) < \nu$ increases between iterations is false.

Thus, for the optimization process of the out-boundary loss over steps, the number of points $\boldsymbol{x}$ such that $d(\boldsymbol{x}) < \nu$ is decreases between steps $t_1$ and $t_2$. $\qquad\square$

**Corollary 1** (Upper Bound of $r$): *The radius $r^{(t_1)}$ serves as an upper bound on the maximum distance of drug-like points from the center at $t_2$ where $t_1 < t_2$. As $r^{(t_1)} > r^{(t_2)}$, fewer compounds lie inside this radius at $t_2$, implying that the boundary of the drug-like space shrinks and becomes more compact.*

*Proof:* By **Proposition 1**, the drug boundary radius $r$, defined as the maximum distance of drug-like points from the center, decreases over steps. In other words, $r^{(t_2)} < r^{(t_1)}$ for $t_2 > t_1$, meaning the boundary becomes tighter as the optimization progresses.

And then, by **Proposition 2**, the number of points $x$ such that $d(x) < \nu$ is decreases over steps for any fixed threshold radius $\nu$. This implies that between steps $t_1$ and $t_2$. the number of compounds within the radius $r^{(t_2)}$ decreases more than the number of compounds within the radius $r^{(t_1)}$.

Since $r^{(t_1)}$ encompasses all drug-like points at time $t_1$ and $r^{(t_2)} < r^{(t_1)}$, we conclude that $r^{(t_1)}$ remains an upper bound on the maximum distance of drug-like points from the center at time $t_2$ even as the boundary shrinks. Therefore, as $r$ decreases with step, the drug boundary become increasingly compact, with fewer compounds lying within the shrinking boundary. $\qquad\square$

Based on the above proofs, we now move on to the proof of **Theorem 1**.

*Proof of Theorem 1:* By **Proposition 1**, we know that the radius $r$, which represents the boundary of drug-like points, decreases over steps as the Euclidean distance loss is minimized. This shrinking boundary implies that the space enclosing the drug-like compounds becomes more compact as the optimization proceeds from $t_1$ to $t_2$.

From **Proposition 2**, we concluded that if the out-compound loss $\mathcal{L}_{\text{out}}$ decreases over steps, the number of points inside an arbitrary radius $\nu$ decreases. Thus, the number of non-drug points within the boundary shrinks as $t$ progresses.

By **Lemma 1**, the contribution to the out-compound loss $\mathcal{L}_{\text{out}}$ from non-drug points inside a given radius $\nu$ is larger than from points outside. Hence, as the number of in-boundary points decreases, the out-compound loss decreases, consistent with the assumption that $\mathcal{L}_{\text{out}}^{(t_1)} > \mathcal{L}_{\text{out}}^{(t_2)}$.

According to the **Corollary 1**, the drug boundary radius $r^{(t_1)}$ serves as an upper bound on the maximum distance of drug-like points from the center, and this boundary becomes more compact over steps. As $r^{(t_2)} < r^{(t_1)}$, fewer non-drug points will lie inside the boundary at step $t_2$. $\qquad\square$

Combining these results, we see that as the optimization proceeds, both the drug boundary shrinks and the number of non-drug points within this boundary decreases. Given that $\mathcal{L}_{\text{drug}}$ and $\mathcal{L}_{\text{out}}$ both decrease between steps $t_1$ and $t_2$, we conclude that the number of non-drug points inside the boundary $|\mathcal{X}_{\text{in-boundary}}|$ decreases as well.

### A.3 CONVERGENCE CRITERION OF EM-LIKE OPTIMIZATION

For our EM-like optimization algorithm, we applied a convergence criterion based on the in-boundary compound ratio (ICR) metric. We initially considered using a traditional loss-based convergence criterion, which would directly correspond to the model's objective of distance minimization. However, due to the nature of our distance metric, convergence using a loss-based criterion proved challenging; it occasionally led to expansions or contractions of the latent space that risked numerical instability (e.g., overflow/underflow issues). Consequently, we adopted the in-boundary compound ratio as the convergence criterion with following reasons.

**Theoretical Alignment** Following the proof of theorem 1 in the Appendix A.2, optimizing the distance metric inherently results in a decrease in the in-boundary compound ratio. This proof establishes a theoretical link between loss minimization and our chosen convergence criterion, indicating that both approaches are consistent with the model's objectives.

**Empirical Stability** We conducted experiments to empirically compare the performance of our model when using the loss-based criterion versus the in-boundary compound ratio (Table 7). The results show no significant difference in final model accuracy, with a p-value of 0.737 which is

greater than 0.05 based on a two-sided paired t-test, demonstrating that the two methods converge to similar solutions. Furthermore, the average number of training epochs needed for convergence was slightly reduced when using the in-boundary compound ratio, indicating faster stabilization.

**Table 7:** Performances of BOUNDR.E with two different convergence metrics. (ICR: In-boundary compound ratio, Avg.: Average)

| Convergence metric | F1 ($\uparrow$) | IDR ($\uparrow$) | ICR ($\downarrow$) | AUROC ($\uparrow$) | Avg. Precision ($\uparrow$) | Avg. Epochs ($\downarrow$) |
|---|---|---|---|---|---|---|
| **Time-based split** | | | | | | |
| ICR | 0.826 (0.0486) | 0.781 (0.0326) | 0.012 (0.0086) | 0.973 (0.0075) | 0.877 (0.0419) | **47.7 (4.20)** |
| $\mathcal{L}_{boundary}$ | 0.833 (0.0463) | 0.806 (0.0236) | 0.014 (0.0098) | 0.973 (0.0071) | 0.885 (0.0463) | 202.7 (99.20) |
| **Paired t-test p-value** | 0.737 | 0.055 | 0.615 | 0.956 | 0.723 | |
| **Scaffold-based split** | | | | | | |
| ICR | 0.655 (0.0209) | 0.796 (0.0258) | 0.063 (0.0079) | 0.938 (0.0049) | 0.590 (0.0369) | **68.5 (4.39)** |
| $\mathcal{L}_{boundary}$ | 0.653 (0.0297) | 0.793 (0.0348) | 0.063 (0.0059) | 0.941 (0.0084) | **0.639 (0.0431)** | 174.2 (21.76) |
| **Paired t-test p-value** | 0.892 | 0.594 | 0.937 | 0.158 | 0.040 | |

## A.4 COMPUTATIONAL COMPLEXITY ANALYSIS

In this section, we provide the detailed computational complexity analysis, further supporting our model's efficiency and scalability.

**E-step (Boundary Update):** The E-step in our model relies on computing the Euclidean distance from the center, with a time complexity linear in both the number of samples ($N$) and the dimensionality ($D$) of the data, resulting in $O(N \times D)$. This ensures that the boundary update is scalable even for high-dimensional datasets.

**M-step (Neural Network Optimization):** In the M-step, the primary computational effort involves neural network optimization. If we denote $H$ as the number of layers, $F_h$ as the number of operations in layer $h$, and $N$ as the dataset size, then the complexity for a forward pass is $O(N \cdot \sum_{h=1}^{H} F_h)$. Given that the backward pass is approximately twice as computationally expensive, the overall complexity for each EM iteration is $O(N \times D) + O(N \cdot \sum_{h=1}^{H} F_h)$.

These complexities illustrate the model's linear behavior with respect to data size and dimensionality, making it efficient for large-scale drug discovery tasks. To validate these claims empirically, we trained our model with approximately 200 drugs and 2,000 non-drug compounds around 100 epochs using single NVIDIA RTX 3090 GPU, and the total training time was consistently under 5 minutes, demonstrating the alignment between theoretical analysis and practical performance.

## A.5 MULTIPLE-EM APPROACH FOR AVOIDANCE OF LOCAL OPTIMA

Avoiding local optima and searching for globally optimal parameters is the core challenge of machine learning. However, classical EM algorithms, including K-means clustering and GMMs, are prone to local optima convergence due to their deterministic and hill-climbing nature of monotonic increase in likelihood, which leads to the model's sensitivity to initialization conditions.

While our model's stochasticity applied with mini-batch training through Adam optimizer allows flexibility to escape monotonic increase and knowledge-aligned embedding space further provides informative initialization point, we aimed to provide a more direct solution to tackle the initialization-sensitiveness of our framework.

Inspired by successful strategies in EM-based models, such as the Multiple Expectation maximizations for Motif Elicitation (MEME) gene motif search algorithm (Bailey & Elkan, 1995), we initialize our boundary optimization process multiple times from different random seeds (for our experiments, $0 \sim 9$) and retain the best-performing model based on the validation set performance without any reliance on the test set. This approach has proven effective in enhancing performance by mitigating the risk of poor local optima. This variant of our model is refered as BOUNDR.E$_{MULT}$ throughout the manuscript.

## A.6 Problem formulation details and comparison with PU learning

Our problem setting roots on the idea to rescue any non-drugs from the compound libraries by not treating any as 'negative drugs'. This motivation naturally led us to apply an one-class classification based approach.

On the other hand, PU learning typically assumes that the distribution of unlabeled data, $P_{\text{unlabeled}}$, can be expressed as a mixture model: $P_{\text{unlabeled}} \sim P_{\text{positive}} + P_{\text{negative}}$. This leads to training objectives rooted in empirical risk minimization that assume a tractable and bounded space of both positive and negative examples with the dataset as a representative subset of such space. In this context, PU methods often aim to minimize classification error with cross entropy-based loss functions by estimating the contribution of a negative distribution, frequently relying on class prior (ratio of positive/negative in the dataset) estimates.

Conventional methods in drug-likeness prediction mainly employ binary classification and sometimes Positive-Unlabeled (PU) learning frameworks, seeking to classify compounds by minimizing the risk of misclassification between positive (drug-like) and negative (non-drug-like) examples with cross entropy-based objectives. However, these approaches rely on defined negative sets or a representative dataset from $P_{\text{negative}}$ distribution, which may not be feasible in the vast and partially known chemical space.

In contrast, our formulation of the drug-likeness prediction task does not assume a well-defined $P_{\text{negative}}$. The chemical space is vast, partially explored, and inherently complex, with any sampled "negative" set non-representative of the true distribution of non-drug compounds. Therefore, instead of attempting to estimate a boundary between positive and potential negatives, we propose a one-class classification framework that constructs a drug-likeness boundary to capture the compact space of drug-like compounds directly, optimized based on distance-based metric learning terms. We summarize the key differences between binary classification, PU-learning and our proposed problem definition of drug-likeeness prediction in Table 8.

**Table 8:** Key differences between binary classification, PU-learning setting and proposed definition of drug-likeness prediction.

|  | Binary classification | PU-learning | One-class Drug-likeness prediction |
|---|---|---|---|
| **Goal** | Decision boundary between positive and negative | Decision boundary between positive and unseen negative | Boundary around positives (here, approved drugs) |
| **Train set composition** | Positive + Negative | Positive + Unlabeled | Drug + Compound |
| **Positive data distribution** | Positives ($P_{\text{positive}}$) | Positives ($P_{\text{positive}}$) | $X_{\text{drugs}}$ as subset of $X_{\text{compound}}$ |
| **Unlabeled data distribution** | - (Only negative data) | $P_{\text{positive}} + P_{\text{negative}}$ (unseen) | $X_{\text{compound}}$ |
| **Assumption of unlabeled dataset** | - | Representative of $P_{\text{positive}}$ and $P_{\text{negative}}$ | Biased subset of intractable $X_{\text{compound}}$ |
| **Characteristics** | Strong reliance to negative set, lower generalizability | Reliance to unlabeled set, lower generalizability | Low reliance to compound set, higher generalizability |
| **Objective** | Risk minimization with cross-entropy | Risk minimization with class prior and cross-entropy | Metric learning (one-class hypersphere) |

## B Initial Study Details

**Scaffold-based distribution of approved drugs**   We analyzed 2,610 approved drugs from Drug-Bank using the Bemis-Murcko scaffold split, which partitions molecules into rings and the linker atoms between them. This decomposition resulted in 1,324 unique scaffold sets, with an average of 1.97 molecules per scaffold. These findings indicate a well-dispersed distribution of approved drugs in the chemical space, with minimal structural overlap. Notably, 1,074 scaffold sets (81.1%) contained only a single compound, further emphasizing the low scaffold redundancy among approved drugs.

Evaluating how models generalize to unseen scaffolds is crucial given the extreme sparsity of the scaffold distribution and its potential impact on model generalization, which encouraged us to perform a scaffold-based splitting scheme, further detailed in Appendix C.3.

**Distribution of approved drugs in representation spaces**  To explore the spatial distribution of approved drugs and non-drug compounds, we represented the structural features of 2,610 approved drugs and 100k ZINC compounds in two distinct spaces: Morgan fingerprints and pretrained Graph-MVP embeddings (Liu et al., 2022). Morgan fingerprints, a type of circular fingerprint, capture molecular structure by encoding atom environments within a specified radius. Each substructure, or circular neighborhood of bonds, is hashed into a bitstring, where each bit indicates the presence or absence of specific substructures in the molecule. This approach creates a fixed-length binary vector, efficiently capturing the molecular topology. In contrast, GraphMVP uses a GNN-based encoder, pretrained to align 2D and 3D molecular structures, to generate embeddings that reflect both graph-level and spatial information about molecules.

For each representation space, we calculated the center point of the drug embeddings (centroid) and defined the drug boundary as the maximum distance from the centroid to any drug. We then computed the distance of all 100k ZINC compounds from this centroid to determine the in-boundary compound ratio (ICR).

Our results indicate that all 100k ZINC compounds were positioned within the drug hypersphere in both the Morgan Fingerprint and GraphMVP spaces. Specifically, the maximum distance of approved drugs from the centroid (i.e., the drug radius) was consistently smaller than the maximum distance of ZINC compounds, confirming that non-drug compounds are distributed further from the drug center in both embedding spaces (Table 9).

**Table 9:** Distribution of drugs and compounds in the two latent spaces. Max: Maximum; ICR: In-boundary compound ratio.

| Representation | Max. Drug distance | Max. Compound distance | ICR |
|---|---|---|---|
| GraphMVP | 29.33 | 25.78 | 1.0 |
| Morgan Fingerprint | 12.02 | 10.01 | 1.0 |

## C  EXPERIMENTAL DETAILS

### C.1  MULTI-MODAL ALIGNMENT SPACES

**Biomedical knowledge graph space**  To represent the biomedical context of drugs, we use embeddings from DREAMwalk (Bang et al., 2023), which has shown efficacy in tasks of drug-disease association prediction and drug repurposing. DREAMwalk employs a heterogeneous skip-gram model to encode entities from the Multi-scale Interactome (MSI) network (Ruiz et al., 2021) into a 300-dimensional vector space. The MSI network integrates information on drugs, genes, diseases, and Gene Ontology terms, enriching each drug representation with biomedical knowledge. We utilize the embeddings of 1,449 approved drugs from DREAMwalk for alignment with their structural representations.

**Molecular Fingerprint Space**  For the structural representation of drugs, we use Morgan Fingerprints, a widely adopted method that encodes molecular structures based on substructure patterns. In this study, we employ 1,024-dimensional Morgan Fingerprints for multi-modal alignment, capturing the structural diversity of the molecules.

### C.2  SEMANTIC DRUG SIMILARITY CALCULATION WITH ATC CODES

**Anatomical Therapeutic Chemical Classification of drugs**  The ATC classification system categorizes drugs based on their therapeutic, pharmacological, and chemical properties. Each drug is assigned a unique ATC code that reflects its primary mechanism of action and target area. The hierarchy is naturally a tree-structured acyclic graph, and on the highest level (Level 1) exists 14 foundational categories, including $A$ (Alimentary tract and metabolism), $B$ (Blood and blood forming organs), $C$ (Cardiovascular system), and more.

A direct modeling of such complex hierarchical structure as prior knowledge in model training is challenging. In order to retain the essence of the hierarchical ATC relationships without complex adjustments to the architecture that may significantly increase computational overhead and complicate

the model training process, we utilized the concept of semantic similarities between terms within the hierarchy and integrated them as prior knowledge to our softened CLIP loss.

**Information Content (IC)**   We adopt the semantic similarity measure introduced by Jiang & Conrath (1997). To quantify the semantic similarity of drugs within the ATC hierarchy, we first need to calculate the Information Content (IC) of each entity. IC measures how informative an entity is, based on its frequency or position within a hierarchical structure. For a term $c$, IC is inversely proportional to the number of child terms $N_{child}(c)$, meaning that terms with fewer descendants have higher IC, as they provide more specific information. The IC for a term in a tree-structured hierarchy is computed as:

$$IC(c) = 1 - \frac{\log(N_{\text{child}}(c) + 1)}{\log(N_{\text{child}}(\text{root}))}$$

This formulation ensures that IC values are normalized within the range $[0, 1]$, where the root entity has an IC of 0.

**Semantic Similarity**   Given two entities $c_1$ and $c_2$ and their Most Informative Common Ancestor (MICA), the semantic distance between them is calculated as:

$$\text{dist}(c_1, c_2) = IC(c_1) + IC(c_2) - 2 \times IC\big(\text{MICA}(c_1, c_2)\big)$$

Since the maximum possible distance is 2 (when IC is 1 for both entities), we normalize the distance into a similarity score in the range $[0, 1)$ using the following equation:

$$\text{sim}(c_1, c_2) = 1 - \left( \frac{\text{dist}(c_1, c_2)}{2} \right)$$

We compute pairwise similarities for all drugs based on their ATC codes, generating a similarity matrix $S \in \mathbb{R}^{n \times n}$, where $n$ is the number of approved drugs.

## C.3   DATA SPLITTING SCHEMES

Two data splitting schemes are employed to rigorously evaluate model generalizability to unseen compounds: a scaffold-based split, which ensures structurally novel compounds appear in the test set, and a time-based split, where drugs approved after a certain time point are assigned to the test set. Since the structural complexity of approved drugs tends to increase over time, with molecular properties diverging (Stegemann et al., 2023), the time-based split is considered a more challenging evaluation compared to scaffold-based splits.

To simulate real-world drug discovery conditions, where the chemical space is much larger than the number of approved drugs, we follow a multi-step procedure: first, split the approved drugs into train-valid-test sets in an 8:1:1 ratio, then sample 10 times the number of test drugs from the 100k ZINC compounds to account for the larger compound space.

### C.3.1   SCAFFOLD-BASED SPLIT

In drug discovery, scaffold diversity is a key concern, as new drugs often emerge from novel scaffolds that were previously untested. The scaffold-split evaluation aligns closely with these real-world scenarios, making it a more rigorous and realistic test of generalization than a random split, where similar scaffolds are likely to appear in both training and test sets.

Drugs are first grouped based on their scaffolds, defined using Bemis-Murcko scaffolds (Bemis & Murcko, 1996), which capture core molecular ring systems and linkers, ensuring that structurally similar drugs are grouped together. Then, the scaffold sets are split into 10 parts for 10-fold cross-validation (CV), with an 8:1:1 ratio for train, validation, and test sets. Each fold ensures that test sets contain unseen scaffolds. The 100k ZINC compounds are also grouped by Bemis-Murcko scaffolds, then split similarly to match the number of drug scaffolds in each fold. For the test set, ZINC scaffolds are sampled to include 10 times the number of drugs.

Our pilot study demonstrates how prediction performance significantly decreases when using scaffold-split compared to randomly splitted setting (Table 10), indicating that the model's ability to handle unseen scaffolds is inherently more challenging. This underscores the necessity of scaffold-split as a more appropriate evaluation scheme for understanding the impact of scaffold sparsity and further evaluate the models' generalizability.

**Table 10:** Prediction performances of BounDr.E when applied on different split schemes. Our model displays significant decrease in prediction performances when applied with scaffold split, a splitting scheme to evalutate the models' generalizability in the sparse distribution of approved drugs' scaffolds. The best performance and comparable values (p-value < 0.05) are marked in bold.

|  | F1 | IDR | ICR | AUROC | Average Precision |
|---|---|---|---|---|---|
| Scaffold-based split | 0.655 (0.0209) | **0.796 (0.0258)** | 0.063 (0.0079) | **0.938 (0.0049)** | 0.590 (0.0369) |
| Random split | **0.689 (0.0142)** | 0.742 (0.0291) | **0.041 (0.0060)** | **0.942 (0.0037)** | **0.663 (0.0379)** |
| Paired t-test p-value | 4.4E-4 | 4.6E-4 | 1.6E-04 | 0.082 | 0.008 |

### C.3.2 TIME-BASED SPLIT

The properties of approved drugs have evolved over the past decades, particularly with the emergence of new therapeutic modalities and technologies. For example, kinase-targeted drugs and biologics became prominent in the 2000s, leading to an increase in molecular complexity, larger molecular weights, and drugs that often fall outside traditional Rule-of-5 constraints (DeGoey et al., 2017). Additionally, the advancement of drug delivery systems has allowed for a higher range of LogP values (lower solubility) among approved drugs (Vargason et al., 2021).

Drugs are first split based on their approval date, with approximate splits of 8:1:1 for train, validation, and test sets. The cut-off years are 2000 and 2011. Drugs approved before 2000 are assigned to the training set, those approved between 2000 and 2010 to the validation set, and drugs approved after 2011 to the test set. Then, The ZINC compound scaffolds are sampled following the same procedure as the scaffold-based split, ensuring 10 times more compounds in the test set.

To validate that our time-based split reflects these temporal trends, we have conducted a detailed analysis of drug properties over the periods represented in our dataset (Table 11). Specifically, we tracked changes in key chemical characteristics (e.g., molecular weight, LogP, polar surface area) across different temporal splits, observing clear shifts that align with known trends in drug development.

**Table 11:** Molecular properties averaged over drugs in the train set (approved before 2011) and test set (approved since 2011). Drugs in the test set show significant difference from the train set drugs, according to the temporal evolution of approved drugs. (Ro5: Number of passed criterions with the Lipinski's Rule of Five)

|  | Ro5 | Molecular Weight | LogP | Polar Surface Area |
|---|---|---|---|---|
| Train (Before 2011) | 3.652739 | 398.120084 | 2.142421 | 100.041105 |
| Test (Since 2011) | 3.379032 | 540.368339 | 2.937724 | 137.452177 |
| Paired t-test p-value | 0.000396 | 0.000583 | 0.024349 | 0.033635 |

### C.4 CROSS-COMPOUND DATASET EVALUATION

We have further performed the performed cross-dataset validation using PubChem and ChEMBL. PubChem contains a vast array of bioassays covering numerous biological targets, while ChEMBL provides curated information on chemical compounds linked to bioactivity against biological targets. These external repositories are widely recognized for their breadth and diversity in assay-centric compound data. We have carefully examined how these datasets complement our original validation set, ZINC20, and their distributions compared with approved drug distribution.

Specifically, we first measured the distributions of three key molecular properties in drug discovery: molecular weight (Mw), LogP and polar surface area (PSA) (Figure 8). The distances between the distributions were computed using 1-Wasserstein distance metric, which display the similarity between ChEMBL compounds and DrugBank approved drugs, followed by PubChem then ZINC20 compounds.

However, the pairwise Tanimoto similarity distribution of molecular fingerprint between DrugBank and other three compound sets reveal that PubChem molecules display the highest average similarity (0.112) compared to ZINC20 (0.111) and ChEMBL (0.013) (Figure 9) Overall, the dissimilarity between datasets demonstrate the uniqueness of each database, and these discrepancies necessitate cross-dataset evaluation for testing the generalizability of drug-likeness prediction models.

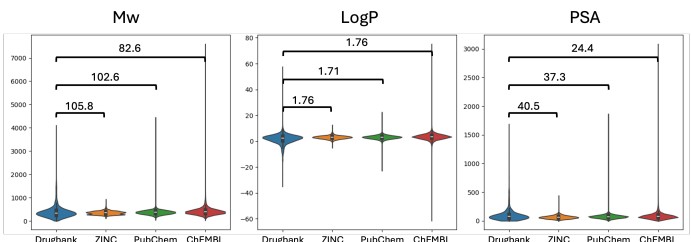 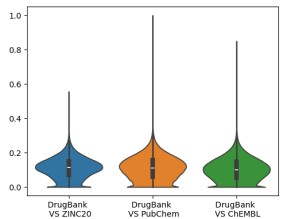

**Figure 8:** Distribution of molecular properties of DrugBank, ZINC20, PubChem and ChEMBL datasets. The numbers between the distributions represent the Wasserstein distance between the two distributions.

**Figure 9:** Distribution of pairwise similarities between DrugBank and compound datasets.

### C.5 MODEL PARAMETERIZATION AND TRAINING DETAILS

The chosen hyperparameter search space (Table 12) aligns with prior work in drug-likeness prediction and molecular property prediction, where 2-3 layers with 256-1024 dimensions are commonly used due to their balance between expressiveness and computational efficiency. The selected configuration was validated through a search on a validation set.

**Multi-modal alignment** Our multi-modal alignment encoders consists of 2-layer multi-layer perceptrons (MLPs) with LayerNorm and ReLU activation. The aligned space is set to `output_dimension=512`. The model is trained using the Adam optimizer (Kingma, 2014) with a `learning_rate=0.001` and `batch_size=32`.

**EM-like boundary optimization** For models requiring boundary optimization, we use a 2-layer MLP architecture with LayerNorm, ReLU activations, and a `hidden_dimension=512`. When generating latent spaces, the `output_dimension` is set to 2. The model is trained with the Adam optimizer (Kingma, 2014) using a `learning_rate=0.0005` and `batch_size=1024`.

**Table 12:** Hyperparameter search space and selected values.

| Parameter | Search space | Selected value |
|---|---|---|
| Alignment_hidden_dim | [512] | 512 |
| Alignment_num_layers | [2,3] | 2 |
| Alignment_drop_out | [None, 0.1] | 0.1 |
| $\lambda_{\text{soft}}$ (Soft CLIP loss weight) | [0.01, 0.1, 0.5, 1] | 0.1 |
| Boundary_hidden_dim | [128,512,1024] | 512 |
| Boundary_out_dim | [2,16,128,512] | 2 |
| Boundary_num_layers | [2,3,4] | 2 |
| Boundary_drop_out | [None, 0.1] | 0.1 |
| Boundary_learning_rate | [1e-4, 5e-4, 1e-3] | 5e-4 |
| Boundary_batch_size | [256, 512, 1024] | 1024 |
| $\alpha$ (drug boundary percentile) | [90, 95, 99, 99.9, 100] | 95 |
| $\lambda_{\text{out}}$ (out-boundary loss weight) | [0.1, 1, 1.5, 2] | 1 |

### C.6 BASELINES

#### C.6.1 DRUG-LIKENESS PREDICTION MODELS

**DrugMetric** DrugMetric[1] (Li et al., 2024) is an unsupervised drug-likeness prediction model based on JT-VAE (Jin et al., 2018) and Gaussian Mixture Models (GMMs). JT-VAE encodes molecules as tree-structured graphs of predefined substructures, with the VAE generating a latent space that follows a Gaussian distribution. Ensemble of GMMs are applied to model this latent space for predicting drug-likeness, and the drug-likeness score is computed using a Wasserstein distance-based metric.

---

[1]github.com/renly0313/DrugMetric

**DeepDL** DeepDL[2] (Lee et al., 2022) introduces two models: (1) an unsupervised LSTM-based model for drug-likeness scoring and (2) a PU learning-based Graph Convolutional Network (GCN) for binary drug-likeness classification. The LSTM model predicts the next token likelihood based on a molecule's string representation, aggregating these probabilities into a drug-likeness score. As this method does not perform strict classification, we focus on the PU learning GCN for comparison.

**D-GCAN** D-GCAN[3] (Sun et al., 2022) is a graph convolution attention network designed for binary drug-likeness classification. The model encodes molecular subgraphs into atom-level vector embeddings using graph convolutional layers, followed by graph attention layers, global sum pooling, and dense layers to learn representations from molecular structures. We reproduce results using the official repository.

### C.6.2 GENERAL CLASSIFIERS

To comprehensively evaluate our model's performance in drug-likeness prediction, we compare it against a range of classifiers for binary classification, PU-learning, and one-class classification tasks. Each model is trained on Morgan fingerprint vectors of dimension 1,024 as molecular input representations.

For comparisons with plain MLP-based architectures, we ensured that both our model and the baselines had identical numbers of layers and parameters. Specifically, each baseline was adjusted to match the total parameter count and architectural capacity of our model, ensuring comparable expressibility. For machine learning-based baseline models, we conducted limited search across a range of hyperparameters, including number of estimators. This search was performed using cross-validation to ensure that the most effective configurations were applied consistently across all models.

**Binary classifiers** For binary classification of drugs and non-drugs, we compare our model with traditional machine learning classifiers, including **Support Vector Machine (SVM)** (Boser et al., 1992) and **eXtreme Gradient Boosting (XGBoost)** (Chen & Guestrin, 2016). XGBoost is a gradient-boosting framework that excels in handling structured data and is widely used for molecular property prediction tasks due to its ability to capture complex patterns in sparse input spaces. SVM constructs a hyperplane (or multiple hyperplanes) to separate data points in high-dimensional space, often using a Radial Basis Function (RBF) kernel to model nonlinear decision boundaries. Both models have demonstrated strong performance in molecular property prediction, often surpassing neural network-based models for certain biological endpoints (Wu et al., 2023). For XGBoost model, we searched its `number of estimators` parameter among [50, 100, 200] and chose 100 as the best parameter.

**PU-learning baselines** Positive-Unlabeled (PU) learning algorithms are well-suited for scenarios where only positive examples (drug-like compounds) and a large set of unlabeled examples are available. We benchmark our model against two PU-learning methods:

- **Naive PU** (Li & Liu, 2003): This method uses the Rocchio classification algorithm, which computes centroids for the positive class and an unlabeled set to form a decision boundary. We adapt this approach with a neural network classifier identical to our model to capture more complex decision boundaries in molecular data.

- **nnPU** (Kiryo et al., 2017): nnPU is an advanced PU-learning algorithm that mitigates overfitting by introducing a non-negative correction term in the risk estimator. This method has shown strong empirical performance in cases where positive and unlabeled data exhibit significant overlap, providing a more robust solution for PU-learning tasks in drug discovery.

**One-Class Classification Baselines** One-class classification methods are designed to distinguish a single target class (e.g., drug-like compounds) from all other compounds without explicitly modeling the negative class. We evaluate the following one-class models:

---

[2]github.com/SeonghwanSeo/DeepDL
[3]github.com/JinYSun/D-GCAN

- **OCSVM** (Schölkopf et al., 2001): One-Class Support Vector Machines (OCSVM) estimate the support of a high-dimensional distribution, fitting a hyperplane that encompasses most of the positive (drug-like) examples. This is widely used in anomaly detection tasks, including outlier detection in chemical spaces.

- **SVDD** (Tax & Duin, 2004): Support Vector Data Description (SVDD) is an extension of SVMs for one-class classification, which minimizes the radius of a hypersphere that encloses the positive data points. The method is particularly effective in constructing compact decision boundaries around the positive class.

- **DeepSVDD** (Ruff et al., 2018): DeepSVDD extends SVDD by utilizing deep neural networks to learn a transformation of input data into a latent space, where the decision boundary is optimized. This method is well-suited for handling high-dimensional and non-linear representations of molecular structures, making it a strong baseline for drug-likeness prediction tasks in high-dimensional spaces.

## D  NOTATION

### Data Sets

| | | | |
|---|---|---|---|
| $\mathcal{X}_{\text{comp}}$ | the set of all chemical compounds | $\mathcal{X}_{\text{in-boundary}}$ | the set of non-drugs inside the boundary |
| $\mathcal{X}_{\text{drug}}$ | the set of drug-like compounds | $z_{\text{drug}}$ | the set of embedded drug compounds |
| $\mathcal{X}_{\text{out}}$ | the set of pseudo-negatives | $D$ | the set of batch data |

### Embedding Spaces and Arrays

| | | | |
|---|---|---|---|
| $\mathcal{S}$ | the structural embedding space | $s_{\text{comp}}$ | the structural embedding vector |
| $\mathcal{K}$ | the biomedical knowledge embedding space | $k_{\text{drug}}$ | the knowledge embedding vector |
| $\mathcal{U}$ | the unified latent space | $\mathcal{Z}$ | the latent space at EM-like training |

### Functions

| | | | |
|---|---|---|---|
| $\mathcal{E}_{\sigma}$ | a structural encoder from space $\mathcal{S}$ to $\mathcal{U}$ | $C(\cdot)$ | the contrastive loss function |
| $\mathcal{E}_{\kappa}$ | a knowledge encoder from space $\mathcal{K}$ to $\mathcal{U}$ | $\mathcal{L}(\cdot)$ | the loss function |
| $f_{\theta}$ | an encoder from space $\mathcal{U}$ to $\mathcal{Z}$ | $\odot$ | the dot-product similarity operator |
| $\mathcal{B}$ | a hyperspherical boundary | $d(\cdot)$ | the Euclidean distance from the boundary center |

### Parameters

| | | | |
|---|---|---|---|
| $c$ | the center of the drug-like compounds | $\rho$ | the in-boundary compound ratio |
| $r$ | the radius of the smallest hypersphere | $\tau$ | the scaling temperature factor |
| $r_{\text{comp}}$ | the radius for all compounds | $\eta$ | learning rate |
| $t$ | the number of iteration steps | $\epsilon$ | convergence tolerance |
| $\theta$ | neural network parameters | $\nu$ | an arbitrary threshold radius |

## E  ADDITIONAL EVALUATION RESULTS

### E.1  RANK-BASED EVALUATION

Since the core concept of our drug-likeness prediction problem lies in treating compound dataset as potential drugs, using classification-centric metrics including F1 score, is not perfectly fit for evaluation of drug-likeness prediction. Since our dataset does not have absolute negative samples,

we here provide further evaluation of models using average precision, precision@k and recall@k metrics in Table 13. These metrics further measure how well models identify drug-like compounds among the vast chemical space.

**Table 13:** Drug-like compound ranking performance with time-based split setting. Mean and standard deviation of 10 fold cross-validation are provided. Best performances marked in bold and second-best underlined.

| | Avg. Precision | Prec@50 | Prec@100 | Prec@200 | Rec@50 | Rec@100 | Rec@200 |
|---|---|---|---|---|---|---|---|
| FP-SVM | 0.724 (0.0174) | 0.852 (0.0160) | 0.777 (0.0090) | 0.540 (0.0067) | 0.344 (0.0065) | 0.627 (0.0072) | 0.871 (0.0108) |
| FP-XGB | 0.775 (0.0213) | 0.868 (0.0458) | 0.773 (0.0155) | 0.538 (0.0117) | 0.350 (0.0185) | 0.623 (0.0125) | 0.868 (0.0188) |
| FP-OCSVM | 0.148 (0.0022) | 0.280 (0.0000) | 0.180 (0.0100) | 0.132 (0.0023) | 0.113 (0.0000) | 0.145 (0.0081) | 0.212 (0.0037) |
| FP-SVDD | 0.143 (0.0022) | 0.240 (0.0000) | 0.144 (0.0049) | 0.108 (0.0040) | 0.097 (0.0000) | 0.116 (0.0040) | 0.174 (0.0064) |
| FP-DeepSVDD | 0.097 (0.0157) | 0.098 (0.0569) | 0.106 (0.0420) | 0.101 (0.0274) | 0.040 (0.0230) | 0.085 (0.0339) | 0.164 (0.0442) |
| FP-nnPU | 0.706 (0.0261) | 0.846 (0.0457) | 0.713 (0.0279) | 0.500 (0.0101) | 0.341 (0.0184) | 0.575 (0.0225) | 0.807 (0.0163) |
| FP-PU | 0.720 (0.0214) | 0.864 (0.0367) | 0.712 (0.0248) | 0.502 (0.0147) | 0.348 (0.0148) | 0.574 (0.0200) | 0.810 (0.0237) |
| DeepDL | 0.886 (0.0374) | 0.976 (0.0233) | 0.846 (0.0393) | 0.513 (0.0172) | **0.448 (0.0215)** | **0.777 (0.0390)** | **0.942 (0.0289)** |
| DGCAN | 0.613 (0.1874) | 0.512 (0.2461) | 0.464 (0.2520) | 0.499 (0.1687) | 0.217 (0.1047) | 0.393 (0.2126) | 0.884 (0.2857) |
| BoundR.E | 0.877 (0.0419) | 0.970 (0.0205) | **0.901 (0.0435)** | 0.562 (0.0108) | 0.391 (0.0083) | 0.727 (0.0351) | 0.907 (0.0205) |
| BoundR.E$_{MULT}$ | **0.908 (0.0096)** | **0.988 (0.0098)** | 0.923 (0.0135) | **0.569 (0.0070)** | 0.398 (0.0040) | 0.744 (0.0108) | 0.918 (0.0113) |

## E.2 Drug-Compound Identification with Scaffold Split

Drug-compound identification performances with scaffold split are provided in Table 14.

**Table 14:** Drug-like compound identification performance with scaffold-split setting. Mean and standard deviation of 10 fold cross-validation are provided. Best performances marked in bold and second-best underlined.

| | MCC (↑) | F1 (↑) | IDR (↑) | ICR (↓) | IDR/ICR (↑) |
|---|---|---|---|---|---|
| FP-SVM | 0.597 (0.0120) | 0.597 (0.0090) | **0.951 (0.0286)** | 0.122 (0.0061) | 7.798 (0.2746) |
| FP-XGB | 0.599 (0.0166) | 0.602 (0.0181) | 0.941 (0.0281) | 0.118 (0.0112) | 8.059 (0.6524) |
| FP-OCSVM | 0.060 (0.1159) | 0.179 (0.0582) | 0.551 (0.2165) | 0.446 (0.0172) | 1.223 (0.4332) |
| FP-SVDD | -0.132 (0.0287) | 0.151 (0.0033) | 0.881 (0.0203) | 0.970 (0.0022) | 0.909 (0.0211) |
| FP-DeepSVDD | -0.120 (0.1607) | 0.147 (0.0294) | 0.834 (0.1787) | 0.938 (0.0423) | 0.890 (0.1871) |
| FP-nnPU | 0.546 (0.0213) | 0.550 (0.0182) | 0.923 (0.0385) | 0.146 (0.0110) | 6.362 (0.4021) |
| FP-PU | 0.549 (0.0239) | 0.555 (0.0188) | 0.907 (0.0491) | 0.135 (0.0130) | 6.776 (0.5185) |
| DrugMetric | -0.028 (0.0794) | 0.160 (0.0238) | 0.692 (0.2932) | 0.690 (0.3452) | 1.115 (0.3095) |
| D-GCAN | 0.599 (0.0340) | 0.594 (0.0456) | 0.859 (0.0966) | 0.109 (0.2808) | 8.145 (1.9174) |
| DeepDL | 0.528 (0.0298) | 0.523 (0.0403) | 0.889 (0.0608) | 0.137 (0.0248) | 6.661 (0.8857) |
| BoundR.E | **0.626 (0.0211)** | **0.655 (0.0209)** | 0.796 (0.0258) | **0.063 (0.0079)** | **12.808 (1.4438)** |

## E.3 Cross-dataset evaluation results

We extended our experiments to cross-dataset evaluation two additional well-established datasets: PubChem and ChEMBL. Both datasets encompass a wide range of chemical scaffolds and molecular properties, making them suitable for testing our model's ability to generalize across varied chemical spaces. As shown in Table 15, our model maintains stable prediction performance across these diverse datasets, demonstrating its ability to generalize effectively beyond the training data.

## E.4 Zero-shot Toxic Compound Identification

### E.4.1 Full table of model performances

We provide the full table of zero-shot toxic compound identification performances on all baseline models in Table 16. DrugMetric in particular fails to yield predictions for withdrawn compound set

**Table 15:** Drug-like compound identification performance on time-split setting with cross-dataset evaluation setting. Mean and standard deviation of 10 fold cross-validation are provided. Best performances marked in bold and second-best underlined.

| | PubChem + DrugBank | | | | | ChEMBL + DrugBank | | | | |
|---|---|---|---|---|---|---|---|---|---|---|
| | F1 | IDR | ICR | Avg. Precision | AUROC | F1 | IDR | ICR | Avg. Precision | AUROC |
| FP-SVM | 0.268 (0.0194) | 0.835 (0.0734) | 0.434 (0.0174) | 0.334 (0.1912) | 0.795 (0.0759) | 0.371 (0.0519) | 0.681 (0.1427) | 0.195 (0.0200) | **0.494 (0.1982)** | 0.819 (0.0768) |
| FP-XGB | 0.254 (0.0209) | 0.810 (0.0804) | 0.451 (0.0197) | 0.320 (0.1181) | 0.773 (0.0741) | 0.358 (0.0589) | 0.675 (0.1411) | 0.206 (0.0213) | 0.469 (0.1839) | 0.814 (0.0784) |
| FP-OCSVM | 0.179 (0.0582) | 0.551 (0.2165) | 0.446 (0.0172) | 0.366 (0.2717) | 0.576 (0.1949) | 0.179 (0.0582) | 0.551 (0.2165) | 0.446 (0.0172) | 0.366 (0.2717) | 0.576 (0.1949) |
| FP-SVDD | 0.151 (0.0033) | **0.881 (0.0203)** | 0.970 (0.0022) | 0.055 (0.0019) | 0.235 (0.0173) | 0.151 (0.0033) | **0.881 (0.0203)** | 0.970 (0.0022) | 0.055 (0.0019) | 0.235 (0.0173) |
| FP-DeepSVDD | 0.147 (0.0294) | 0.834 (0.1787) | 0.938 (0.0423) | 0.080 (0.0146) | 0.415 (0.1224) | 0.147 (0.0294) | 0.834 (0.1787) | 0.938 (0.0423) | 0.080 (0.0146) | 0.415 (0.1224) |
| FP-nnPU | 0.244 (0.0182) | 0.833 (0.0727) | 0.504 (0.0637) | 0.240 (0.0816) | 0.749 (0.0556) | 0.327 (0.0525) | 0.666 (0.1337) | 0.241 (0.0374) | 0.380 (0.1999) | 0.778 (0.0812) |
| FP-PU | 0.241 (0.0265) | 0.664 (0.1219) | 0.379 (0.0528) | 0.228 (0.0556) | 0.702 (0.0560) | 0.311 (0.0495) | 0.653 (0.1477) | 0.250 (0.0311) | 0.396 (0.1701) | 0.778 (0.0874) |
| DeepDL | 0.170 (0.0199) | 0.764 (0.0754) | 0.598 (0.0481) | 0.092 (0.0112) | 0.590 (0.0233) | 0.195 (0.0389) | 0.681 (0.1329) | 0.530 (0.1553) | 0.102 (0.0196) | 0.612 (0.0686) |
| Ours | **0.501 (0.0232)** | 0.759 (0.0441) | **0.126 (0.0148)** | 0.460 (0.0380) | **0.875 (0.0157)** | **0.513 (0.0451)** | 0.746 (0.0281) | **0.117 (0.0190)** | 0.435 (0.0889) | **0.869 (0.0258)** |

since JTVAE is capable of encoding only the scaffolds present in the training set, in this case the combined set of ZINC and DrugBank approved drugs.

**Table 16:** False-positive rate of toxic compound groups. The best performances and the comparable values (paired t-test p-value < 0.05) are marked in bold.

| | Withdrawn | Hepatotoxic | Cardiotoxic | Carcinogenic |
|---|---|---|---|---|
| **FP-SVM** | 0.98 (0.001) | 0.98 (0.001) | 0.86 (0.006) | 0.98 (0.002) |
| **FP-XGB** | 0.96 (0.003) | 0.96 (0.003) | 0.85 (0.010) | 0.93 (0.010) |
| **FP-SVDD** | 0.95 (0.002) | 0.93 (0.002) | 0.92 (0.003) | 0.99 (0.001) |
| **FP-OCSVM** | 0.69 (0.002) | **0.53 (0.003)** | 0.25 (0.006) | 0.86 (0.001) |
| **FP-DeepSVDD** | 0.81 (0.022) | 0.80 (0.020) | 0.87 (0.032) | 0.56 (0.063) |
| **FP-PU** | 0.95 (0.007) | 0.94 (0.005) | 0.87 (0.021) | 0.85 (0.009) |
| **FP-nnPU** | 0.95 (0.009) | 0.94 (0.007) | 0.87 (0.028) | 0.86 (0.017) |
| **DrugMetric*** | *N/A* | 0.77 (0.073) | 0.76 (0.118) | 0.82 (0.087) |
| **DGCAN** | 0.91 (0.020) | 0.85 (0.023) | 0.88 (0.045) | 0.95 (0.017) |
| **DeepDL** | 0.91 (0.016) | 0.92 (0.018) | 0.85 (0.042) | 0.84 (0.025) |
| **BOUNDR.E** | **0.52 (0.041)** | 0.54 (0.028) | **0.20 (0.019)** | **0.20 (0.043)** |
| **BOUNDR.E_MULT** | 0.51 (0.014) | 0.54 (0.009) | 0.20 (0.009) | 0.19 (0.014) |

*DrugMetric fails to infer scaffolds not present in approved drug and ZINC datasets

### E.4.2 ERROR ANALYSIS ON "PARTIALLY-WITHDRAWN" DRUGS

We conducted an in-depth error analysis on the false-positive withdrawn drugs predicted as "in-drug-boundary" by our model, identifying a trend of predictions involving drugs referred to as "partially-withdrawn"—drugs that are approved in some regions but withdrawn in others, in contrary to "fully-withdrawn" drugs. This category represents complex cases where the criteria for withdrawal may vary.

Our analysis across 10 trials revealed a significantly higher presence of partially-withdrawn drugs in the in-drug-boundary predicted set (61.2%) compared to out-drug-boundary ones (38.8%) with p-value of 7.8E-3 (paired t-test) (Fig. 10). This suggests that our model's predictions reflect real-world complexities in regulatory approval, while maintaining a false positive ratio of 0.52, with 60% of these false positives falling into this partially-withdrawn category.

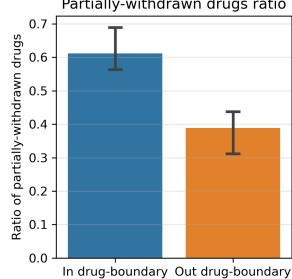

**Figure 10:** Partially-withdrawn drug ratio between in- and out-drug-boundary sets.

**Table 17:** Drug-like compound identification with EM-like boundary optimization on embedding space aligned with alignment method ablations on time-based split scheme. Best performance and comparable values in bold.

| Alignment method | F1 (↑) | IDR (↑) | ICR (↓) | AUROC (↑) | Avg. Precision (↑) |
|---|---|---|---|---|---|
| Ours w/ Original CLIP | 0.727 (0.0365) | 0.670 (0.0605) | 0.018 (0.0066) | 0.801 (0.0506) | 0.755 (0.0481) |
| Ours w/o S,K-mix | 0.466 (0.1705) | 0.745 (0.1058) | 0.270 (0.3446) | 0.818 (0.1825) | 0.420 (0.1995) |
| Ours w/o KS-mix | 0.604 (0.2238) | 0.858 (0.0734) | 0.241 (0.3782) | 0.849 (0.2091) | 0.576 (0.2546) |
| No alignment (only FP) | 0.539 (0.0324) | 0.571 (0.0176) | 0.057 (0.0161) | 0.907 (0.0144) | 0.557 (0.0461) |
| Ours (softened CLIP + S,K,KS-mix) | **0.826 (0.0486)** | **0.781 (0.0326)** | **0.012 (0.0086)** | **0.973 (0.0075)** | **0.877 (0.0419)** |

## E.5 ADDITIONAL ABLATION STUDY RESULTS

### E.5.1 EFFECT OF EM-LIKE OPTIMIZATION

The core advantage of our method lies in its iterative updates to both the decision boundary and the encoder. Unlike other classifiers including MLP, which relies on fixed embeddings, our algorithm dynamically adjusts the feature space and boundary across multiple iterations as following:

1. An initial, coarse boundary is set using the contrastive embeddings.

2. The encoder refines these embeddings based on feedback from the initial boundary, adjusting the representation.

3. A new boundary is established using these refined embeddings.

4. This process repeats, allowing the model to fine-tune both the decision criteria and the feature space.

This iterative refinement can also be seen in Figure 11, where the ratio of out-boundary compounds increases and converges over time with each EM iteration. This progressive refinement demonstrates the limitations of a static MLP approach, reinforcing the necessity of our iterative EM-like strategy for accurate boundary learning.

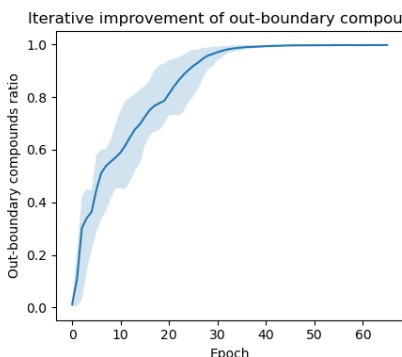

**Figure 11:** Iterative improvement of out-boundary compound ratio. Line plot shows the average over 10 trials, and area between maximum and minimum values are colored.

### E.5.2 EFFECT OF MULTI-MODAL ALIGNMENT WITH SOFTENED CLIP LOSS

Our multi-modal alignment loss encompases four modules; softened-CLIP loss, S and K-mix, and KS-mix. While softened-CLIP loss is designed to integrate prior knowledge as ATC semantic similarity, geodesic mixup-inspired loss terms—S-mix, K-mix, and KS-mix—facilitate the learning of the intermediate space between conflicting representations. Specifically:

- **S-mix & K-mix:** These loss terms focus on intra-space interpolation within the structural (S-mix) and knowledge-based (K-mix) embeddings, respectively. By encouraging the model to interpolate between known data points, it learns a smoother and more continuous embedding space, reducing sensitivity to local conflicts.

- **KS-mix:** This component specifically targets inter-space interpolation, blending structural and biomedical representations. It creates synthetic data points that reflect a balanced compromise between structural and biomedical features, enabling the model to harmonize inconsistencies and achieve a unified representation.

We evaluated the performance of the model by selectively removing each component the final setup (Table 17). The results indicate that each component contributes uniquely to the model's performance. Replacing the softened CLIP loss with the original CLIP loss brought 10 percent point loss in F1 score, highlighting the importance of knowledge integration in our model's accurate performances. Removing both S-mix and K-mix resulted in a drop of 36 percent points in F1 score,

indicating their contribution to aligning embeddings across diverse drug classes and scaffolds in each of structural and knowledge spaces. Additionally, without KS-mix, the model showed a reduction of 22 percent point in F1 score, underscoring the importance of a balanced contribution from both structural and semantic features.

Overall, our results show that the combination of all three strategies yields the best performance, with a synergistic effect that improves both classification accuracy and stability, effectively integrating knowledge and simultaneously resolving conflicts between structural and biomedical spaces.

### E.6 Filtering AI-generated anti-cancer molecules

#### E.6.1 Details on utilized property-based filtering criteria

**PAINS filter**   The PAINS (Pan-Assay Interference Compounds) filter is designed to identify and eliminate molecules that are likely to produce false-positive results in high-throughput screening assays. These compounds often interfere with biological assays through non-specific mechanisms such as covalent binding, redox activity, or fluorescence interference. The PAINS filter operates by detecting specific substructures known to cause assay interference. In our pipeline, each compound is scanned against a comprehensive library of PAINS substructure patterns. Compounds that do not contain any of these substructures are considered clean and retained for further analysis. This filter ensures that the remaining molecules have a reduced likelihood of assay-related artifacts, enhancing the reliability of downstream predictions.

**Lipinksi's Rule of 5**   Lipinski's Rule of Five (Ro5) is a widely accepted guideline to assess the drug-likeness of a molecule based on its physicochemical properties. The rule includes four criteria:

1. Molecular Weight must be less than or equal to 500 Daltons.
2. LogP (Partition Coefficient) must be less than or equal to 5, ensuring favorable lipophilicity.
3. No more than 5 hydrogen bond donors (sum of OH and NH groups).
4. No more than 10 hydrogen bond acceptors (sum of O and N atoms).

Compounds that adhere to all four criteria are considered to have favorable pharmacokinetic properties, such as good oral bioavailability and permeation, and are retained for further consideration. By applying this rule, we effectively filter out molecules that are less likely to succeed in later stages of drug development due to poor absorption or bioavailability.

**Predicted IC50**   Binding affinity prediction is a critical step for assessing the potential biological activity of a compound. We employed XGBoost models to predict IC50 values, which represent the concentration of a compound required to inhibit a biological process by 50%. These models were trained on bioassay datasets from with IC50 values in ChEMBL database, specifically: BCR-ABL (CHEMBL2096618), EFGR (CHEMBL203), and CDK6 (CHEMBL2508) (accessed 16 November 2023).

The input features for these models were Morgan molecular fingerprints, which capture key structural and functional aspects of each compound. Compounds predicted to have an IC50 below 10 μM are classified as "active" and retained. This threshold was selected to balance the need for potent biological activity with the feasibility of further development, ensuring that only promising candidates proceed to subsequent stages of evaluation.

#### E.6.2 Characteristics of in-drug-boundary compounds

In this section, we provide detailed experimental results in investigating the potentials of our model as a complementary data-driven filter in a AI-driven rational drug discovery pipeline. To be specific, our model can serve as an efficient, early-stage filtering tool that can significantly narrow down the

search space in large chemical libraries, thereby easing the computational burden on subsequent analyses.

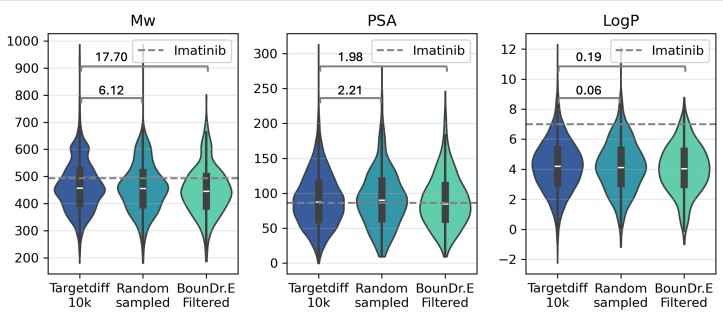

**Figure 12:** Distribution of molecular properties of Targetdiff generated molecules on BCR protein pocket (PDB: 1OPJ) and its filtered sets. BOUNDR.E-filtered set shows more distant distribution of molecular properties from the original 10k molecules.

**Table 18:** Various traditional drug-likeness measures of Targetdiff generated molecules and filtered sets. Most desirable values are in bold. (SAS: Synthetic Accessibility Score; Avg.: Average)

| Target protein | BCR (PDB: 1OPJ) | | | EGFR (PDB: 4HJO) | | | CDK6 (PDB: 5L2T) | | |
|---|---|---|---|---|---|---|---|---|---|
| Groups | SAS (↓) | Avg. QED (↑) | Ro5 ratio (↑) | SAS (↓) | Avg. QED (↑) | Ro5 ratio (↑) | SAS (↓) | Avg. QED (↑) | Ro5 ratio (↑) |
| TargetDiff 10k | 4.956 | 0.425 | 0.474 | 5.562 | 0.410 | 0.521 | 5.378 | 0.384 | 0.507 |
| Random sampled* | 4.958 | 0.426 | 0.475 | 5.586 | 0.409 | 0.514 | **5.353** | 0.382 | 0.508 |
| BounDr.E filtered | **4.930** | **0.433** | **0.532** | **5.477** | **0.413** | **0.546** | 5.523 | **0.392** | **0.532** |

\* Repeated 100 times

We applied our model to filter 10,000 AI-generated compounds from TargetDiff, using three widely-known anti-cancer targets: BCR, EGFR and CDK6, each targeted by cancer drugs imatinib, erlotinib and ribociclib, respectively. After screening with our drug boundary, we retained 300, 374 and 264 in-boundary compounds for each target. For comparison, we randomly sampled the equal amount of molecules (repeated 100 times) and measured key molecular properties of the filtered drugs, including polar surface area (PSA), molecular weight (Mw), and logP.

Figure 12 highlights a significant shift in key drug-like properties in the BOUNDR.E-filtered compounds compared to randomly sampled compounds generated for BCR. Furthermore, Table 18 shows a marked increase in average QED, Ro5-passing ratio and Synthetic Accessibility Score (SAS), implying the sampled compounds are more drug-like whens cross-measured through conventional metrics. In detail, the Wasserstein distance of the three properties from the starting 10k compounds reveal that our filtering strategy significantly alters the distribution of the key molecular properties of filtered compounds (Table 19).

**Table 19:** Properties of filtered Targetdiff-generated molecules and their distributional distance from to the original distribution of 10k generated molecules for three protein targets (BCR, EGFR, CDK6). (W-distance: 1-Wasserstein distance)

| Groups | W-distance from BCR-10k | | | W-distance from EGFR-10k | | | W-distance from CDK6-10k | | |
|---|---|---|---|---|---|---|---|---|---|
| | Mw | PSA | logP | Mw | PSA | logP | Mw | PSA | logP |
| Random sampled | 6.122 | **2.205** | 0.058 | 6.882 | **2.566** | 0.099 | 5.997 | **5.584** | **0.266** |
| BounDr.E filtered | **17.695** | 1.979 | **0.187** | **16.834** | 2.298 | **0.168** | **10.903** | 5.032 | 0.135 |

\* Repeated 100 times

In addition, the Probability Density Function (PDF) of approved drugs, imatinib, erlotinib and ribociclib among the three properties also increased, implying identifying the approved drugs among the filtered molecules is more likely with our filtered set (Table .

**Table 20:** PDF of approved drugs with the distribution of three key molecular properties on different filtered sets, originated from 10k generated molecules for three protein targets (BCR, EGFR, CDK6). (Mw: Molecular weight; PSA: Polar surface area)

| Groups | PDF of imatinib (BCR) | | | PDF of Erlotinib (EGFR) | | | PDF of Ribociclib (CDK6) | | |
|---|---|---|---|---|---|---|---|---|---|
| | Mw | PSA | logP | Mw | PSA | logP | Mw | PSA | logP |
| TargetDiff 10k | 4.00E-03 | 1.02E-03 | 2.26E-01 | 2.88E-03 | 5.10E-03 | 2.14E-01 | 3.64E-03 | 9.49E-03 | 2.06E-01 |
| Random sampled* | **4.02E-03** | 1.03E-03 | 2.27E-01 | 2.84E-03 | 5.05E-03 | 2.14E-01 | **3.68E-03** | **9.57E-03** | 2.06E-01 |
| BounDr.E filtered | 3.94E-03 | 1.05E-03 | **2.32E-01** | **3.09E-03** | **5.32E-03** | **2.26E-01** | 3.49E-03 | 8.87E-01 | **2.18E-01** |

\* Repeated 100 times

The Wasserstein distance and Probability Density Function (PDF) of imatinib properties are measured using gaussian KDE. The properties of the approved drugs are computed with rdkit python package.

These findings demonstrate the practical utility of our model in filtering AI-generated compounds, enabling efficient virtual screening and improving the quality of early-stage candidates.

### E.7 ANTI-CANCER SPECIFIC BOUNDR.E RESULTS

In this section, we provide experimental results on the anti-cancer variant of our model, demonstrating our model's potential real-world impact in targeted drug discovery.

One of the strengths of our one-class boundary approach is its adaptability to domain-specific contexts by relying solely on the input positive labels. To explore this flexibility, we newly designed and conducted a concept study, using anti-cancer drugs. We first filtered our training set to include only drugs classified under the ATC code 'L' (Antineoplastic and immunomodulating agents), which specifically targets the anti-cancer domain. This narrowed training set of 239 drugs allowed our model to learn a more focused boundary representative of the anti-cancer chemical space. We investigated this anti-cancer BounDr.E model with two scenarios:

**Broader boundary for anti-cancer compounds** When filtering the 10k generated compounds with anti-cancer target protein pocket as conditions, the anti-cancer-boundary obtained much higher ratio of drug candidates compared to the general drug boundary, which means the model adequately learned the protein target context of anti-cancer drugs (Table 21).

**Table 21:** Filtering anti-cancer target-based generated molecules with general BounDr.E and anti-cancer-BounDr.E models. Approximately 10k molecules were generated and filtered for BCR, EGFR and CDK6, three well-known anti-cancer targets. Compared to general BounDr.E model, Anti-cancer-BounDr.E model recommends more candidates, according to the generated compounds' context.

| Filtering Method | BCR | EGFR | CDK6 |
|---|---|---|---|
| **Total Generated** | 10,543 (100%) | 12,550 (100%) | 11,496 (100%) |
| **Anti-cancer BounDr.E** | 434 (3.9%) | 434 (3.9%) | 495 (4.9%) |
| **General BounDr.E** | 300 (2.8%) | 374 (3.0%) | 264 (2.3%) |

**Strict boundary in toxic compound filtering** On contrary and interestingly, false-positive ratio on toxic and carcinogenic compounds was significantly reduced when applying the anti-cancer-specific boundary, highlighting the model's ability to filter out irrelevant or potentially harmful compounds more effectively (Table 22) with more compact boundary, while encompassing the contexts of anti-cancer drugs. The results imply that our model's anti-cancer variant, while providing a broader boundary for anti-cancer generated compounds, shows strictness for toxic compounds, tailored for anti-cancer drug discovery.

**Table 22:** Toxic compound filtering comparison with best performances marked in bold. The anti-cancer BounDr.E model displays significant reduction in false positive rate compared to general BounDr.E model.

| | Withdrawn | Hepatotoxic | Cardiotoxic | Carcinogenic |
|---|---|---|---|---|
| **General BounDr.E** | 0.523 (0.0414) | 0.541 (0.0284) | 0.207 (0.0190) | 0.208 (0.0436) |
| **Anti-cancer BounDr.E** | **0.195 (0.0363)** | **0.151 (0.029)** | **0.149 (0.0356)** | **0.148 (0.0321)** |
| **Paired t-test p-val** | 2.30E-09 | 6.80E-12 | 2.20E-04 | 1.00E-03 |

## E.8 STATISTICAL VALIDATION RESULTS

In this section, we provide the statistical validation results for the tables in the main text (Tables $1 \sim 5$), computed with one-sided paired t-test to compare the significance compared to the best performing models.

**Table 23:** Statistical validation for drug-like compound identification performance with time-split setting (Table 1). Mean and standard deviation of 10 fold CV are provided. Best performance and its comparable results (paired t-test $p < 0.05$) are marked in bold.

| | F1 ($\uparrow$) | IDR ($\uparrow$) | ICR ($\downarrow$) | AUROC ($\uparrow$) | Avg. Precision ($\uparrow$) |
|---|---|---|---|---|---|
| **SVM** | 1.0 | 1.0 | 1.0 | 1.0 | 1.0 |
| **XGB** | 1.0 | 1.0 | 1.0 | 1.0 | 1.0 |
| **OCSVM** | 1.0 | 1.0 | 1.0 | 1.0 | 1.0 |
| **DeepSVDD** | 1.0 | 0.9999 | 1.0 | 1.0 | 1.0 |
| **nnPU** | 1.0 | 1.0 | 1.0 | 1.0 | 1.0 |
| **naive PU** | 1.0 | 1.0 | 1.0 | 1.0 | 1.0 |
| **DrugMetric\*** | 1.0 | 1.0 | 1.0 | 1.0 | 1.0 |
| **DGCAN** | 0.9947 | **Best** | 0.9311 | 0.9988 | 0.8841 |
| **DeepDL** | 0.9999 | 0.9905 | 0.9999 | **Best** | 0.4459 |
| **BounDrE** | 0.8596 | 1.0 | 0.7531 | 0.661 | 0.9444 |
| **BounDrE$_{Mult}$** | **Best** | 1.0 | **Best** | 0.07378 | **Best** |

**Table 24:** Statistical validation for cross-dataset evaluation of drug-like compound identification performance on scaffold-split setting, trained on PubChem/ChEMBL and evaluated with ZINC20 compounds (Table 2). One-sided paired t-test p-values of 10 trials compared to the best model are provided. Best and its comparable performances (paired t-test $p < 0.05$) are marked in bold.

| Train set | PubChem + DrugBank | | | ChEMBL + DrugBank | | |
|---|---|---|---|---|---|---|
| | F1 ($\uparrow$) | Average Precision ($\uparrow$) | AUROC ($\uparrow$) | F1 ($\uparrow$) | Average Precision ($\uparrow$) | AUROC ($\uparrow$) |
| **SVM** | 1.0 | 0.9981 | 0.9985 | 1.0 | 0.9204 | 0.9765 |
| **XGB** | 1.0 | 0.9714 | 0.9939 | 1.0 | **Best** | 0.9735 |
| **OCSVM** | 1.0 | 1.0 | 1.0 | 1.0 | 0.9997 | 1.0 |
| **DeepSVDD** | 1.0 | 1.0 | 1.0 | 1.0 | 0.9999 | 1.0 |
| **nnPU** | 1.0 | 1.0 | 0.9999 | 1.0 | 1.0 | 0.9976 |
| **PU** | 1.0 | 1.0 | 1.0 | 1.0 | 1.0 | 0.9957 |
| **DGCAN** | 1.0 | 1.0 | 1.0 | 1.0 | 0.9997 | 0.998 |
| **DeepDL** | 1.0 | 1.0 | 1.0 | 1.0 | 0.9999 | 1.0 |
| **BOUNDR.E** | 0.765 | 0.820 | 0.603 | 0.9488 | 0.8918 | 0.9542 |
| **BOUNDR.E$_{MULT}$** | **Best** | **Best** | **Best** | **Best** | 0.560 | **Best** |

**Table 25:** Statistical validation for false-positive rate of toxic compound groups (Table 3). One-sided paired t-test p-values of 10 trials compared to the best model are provided. Lowest and its comparable results (paired t-test $p < 0.05$) are marked in bold.

|  | Withdrawn | Hepatotoxic | Cardiotoxic | Carcinogenic |
|---|---|---|---|---|
| **XGB** | 1.0 | 1.0 | 1.0 | 1.0 |
| **OCSVM** | 1.0 | **Best** | 1.0 | 1.0 |
| **nnPU** | 1.0 | 1.0 | 1.0 | 1.0 |
| **DrugMetric** | N/A | 0.9616 | 0.9995 | 1.0 |
| **DGCAN** | 1.0 | 1.0 | 1.0 | 1.0 |
| **DeepDL** | 1.0 | 1.0 | 1.0 | 1.0 |
| **BOUNDR.E** | 0.8639 | 0.8571 | 0.7272 | 0.8735 |
| **BOUNDR.E$_{\text{MULT}}$** | **Best** | 0.9875 | **Best** | **Best** |

$^*$DrugMetric fails to infer scaffolds not present in approved drug and ZINC datasets

**Table 26:** Statistical validation for drug-like compound identification with EM-like boundary optimization on embedding space aligned with different alignment methods (Table 4). One-sided paired t-test p-values of 10 trials compared to the best model are provided. Lowest and its comparable results (paired t-test $p < 0.05$) are marked in bold.

| Alignment method | F1 ($\uparrow$) | ICR ($\downarrow$) |
|---|---|---|
| **No Alignment** (only FP) | 1.0 | 0.7489 |
| **Manifold Alignment** | 1.0 | **Best** |
| **CLIP** | 1.0 | 0.4685 |
| **Geodesic Mixup** | 0.9998 | 0.001325 |
| **Ours - softCLIP** | 0.9992 | **8.50E-06** |
| **Ours** | **Best** | 9.86E-08 |

**Table 27:** Statistical validation for drug-like compound identification with different classifiers on knowledge-aligned space (Table 5). Best performance in bold and second best underlined. One-sided paired t-test p-values of 10 trials compared to the best model are provided. Lowest and its comparable results (paired t-test $p < 0.05$) are marked in bold.

| Aligned space | F1 ($\uparrow$) | ICR ($\downarrow$) |
|---|---|---|
| **+ MLP** | 1.0 | 1.0 |
| **+ SVM** | **Best** | 0.9863 |
| **+ XGB** | 1.0 | 1.0 |
| **+ naive PU** | 1.0 | 0.9999 |
| **+ DeepSVDD** | 1.0 | 1.0 |
| **+ Ours $-$ EM** | 1.0 | 0.9978 |
| **+ Ours** | 0.9816 | **Best** |

