# OpenReview forum: "BounDr.E: Predicting Drug-likeness through knowledge alignment and EM-like one-class boundary optimization"
_ICLR.cc/2025/Conference — ICLR 2025 Conference Withdrawn Submission_

### Official Review · Reviewer_vdXE · 2024-10-21

**Soundness:** 3
**Presentation:** 3
**Contribution:** 3
**Rating:** 6
**Confidence:** 3

**Summary:**

In this paper, the authors address two main challenges:

1. The accuracy of identifying drug candidates is still challenging.
2. The distribution of existing approaved drugs is wide-spread, make the one-class classification more challenging.

The authors address the problem by proposing:

1. a novel multi-modal mixup framework for aligning knowledge between knowlwdge graph and structural embedding space.
2. a improved EM-like algorithm to optimize the decision boundary for one-class classification between drug and non-drug compounds.

Through experiments, the proposed method achieves State-Of-The-Art around all metrics. Also, extensive experiments such as zero-shot prediction, embeeding visualization, and distance distribution demonstrates the robustness and effectiveness of the method. Finally, the method is applied to filter generated drug candidates, making it potential applicable to wider field of drug filtering.

**Strengths:**

1. The method improvement is well-designed and very closely related to the existing challenges.
2. The experiment validation from different perspectives are comprehensive and sound.
3. Mathmatical proof is well-written and supportive to the approach enhancement.

**Weaknesses:**

1. Some evaluation metrics are not well-designed, and some conlusion drawn from experiments are not reasonable.
2. Somehow, KS-mix lacks novelty.

**Questions:**

1. Regarding Section4.1, the training data is chosen from DrugBank and ZINC. I can understand the proposed two split methods, but is it possible selecting compounds from other datasets to construct the dataset? It is because in your proposed setting, there should be an intersection between the distribution of drug and compunds. Moreover, the larger the intersection is, the more challenging the task would be. In the paper, only one compound dataset is employed without initial distribution analysis, which makes the validation of model's performance to the real-world dataset questionable. Would you please provide a more detailed introduction to the dataset construction and distribution analysis?

2. Regarding Section4.2, three metrics are proposed, including ICR, IDR, and IDR/ICR. I have two doubts. The first one is the proposed EM improvement would benefit identifying identifying the boundary. But w.r.t. ICR, the performance is lower than existing methods like DeepDL and D-GCAN. Does it mean that the drug decision boundary is relative poor? The second question is that in line 357, "IDR/ICR ratio to report the models’ capability in balancing the trade-off between inclusion of drug-like compounds with exclusion of non-drugs.". But, the detection of drug and compounds is not correlated, which means that is it not a binary classification senario like MCC and F1. Also, the ratio of two metric cannot refect the tradeoff. Instead, the sum would be a better one. Would you please provide a detailed explanation on this metric?

3. In Figure6, there is performance comparsion between F1 score according to Drug/Compound Ratio. We can see that higher ratio leads to higher difficulty. Futhermore, the proposed method achieves SOTA only when the ratio is over 1:50. Thus, a natural question is that what is the corresponding ratio in your training and testing sets.

4. In Figure8, the caption states that "BOUNDR.E-filtered set shows more distinct distribution of molecular propertirs from the original 10k molecules." But, in the plot results, the above conclusion seems not able to be easily drawn. May you provide a detailed analysis?

---

> ### Author Response · Authors · 2024-11-21
> **Response to Reviewer vdXE's comments (1)**
>
> We sincerely thank the reviewers for their insightful comments and constructive feedback, which enabled us to improve our work further. In the following comments, we respond to each comment for your initial assessment. A revised manuscript will be available shortly, incorporating these modifications.
>
> **Weaknesses**
> > **W1.** Some evaluation metrics are not well-designed, and some conlusion drawn from experiments are not reasonable.
>
> We are grateful to the reviewer for pointing out potential weaknesses in the evaluation metrics and the conclusions drawn from our experiments. This feedback is invaluable in refining the clarity and robustness of our work.
>
> 1. We understand the reviewer's concerns about the design of our evaluation metrics. Our choice of metrics aligns with established standards in one-class classification scenarios, focusing on assessing the precision of drug-like compound boundaries without relying on negative examples. To strengthen our validation, we have replaced the controversial metrics to more widely used metrics of AUROC and Average Precision. The details can be found in the responses to Question Q2.
>
> 2. We acknowledge that certain conclusions in the initial manuscript may have appeared overly strong or insufficiently supported by the presented data. We have revisited the conclusions, ensuring they accurately reflect the experimental evidence. The detailed cases of revised conclusions are enlisted in responses to Q3 and Q4.
>
>
> > **W2.** Somehow, KS-mix lacks novelty.
>
> We appreciate the reviewer's feedback regarding the novelty of our KS-mix approach. We acknowledge that the initial presentation may not have sufficiently highlighted the distinct aspects of our contribution.
>
> 1. In our revised manuscript, we have clarified the contributions of our work as below. We hope this improved clarity helps the readers to understand the novelties of our work.
> ```
> Our contributions can be summarized as following:
> 1. Novel formulation of drug-likeness prediction as a one-class classification without reliance on negatives.
> 2. Proposal of EM-like optimization of both the drug-likeness boundary and the embedding space for accurate drug-likeness prediction.
> 3. Knowledge-integrated multi-modal alignment of structure and biomedical knowledge embeddings for defining drug-likeness with machine learning.
> ```
>
>
>
> 2. While our methodology of application of S-mix, K-mix and KS-mix draws inspiration from geodesic Mixup and related approaches, we have **extended these ideas through the application of domain-specific adaptations along with introduction of novel alignment loss, softened CLIP**, in which the harmony of all losses bring the optimal performances, demonstrated through ablation study results. We have additionally clarified the novelty of our alignment losses in the Methods section of our main manuscript by adding:
> ```
> 3.2.1 SOFTENED CLIP LOSS WITH ATC SIMILARITY
>     In this section, we propose a novel knowledge-integration strategy for multi-modal contrastive learning. We soften the CLIP loss (Radford et al., 2021) by incorporating semantic similarity (Jiang & Conrath, 1997) between drugs using Anatomic Therapeutic Chemical (ATC) classification.
> ```
>
> **Questions**
> > **Q1.** Regarding Section4.1, the training data is chosen from DrugBank and ZINC. I can understand the proposed two split methods, but is it possible selecting compounds from other datasets to construct the dataset? It is because in your proposed setting, there should be an intersection between the distribution of drug and compunds. Moreover, the larger the intersection is, the more challenging the task would be. In the paper, only one compound dataset is employed without initial distribution analysis, which makes the validation of model's performance to the real-world dataset questionable. Would you please provide a more detailed introduction to the dataset construction and distribution analysis?
>
> Thank you for pointing out the need for a more comprehensive explanation regarding our dataset construction and distribution analysis. We acknowledge that relying on only DrugBank and ZINC may limit the generalizability of our findings. Recognizing the importance of broader validation, **we have conducted additional experiments using other well-known datasets, specifically ChEMBL and PubChem, to address concerns about dataset diversity**. Specifically, we trained our model on compounds from ChEMBL/PubChem and evaluated its performance using ZINC compounds in the test set, with the drug dataset fixed to DrugBank. This experiment exposed our model to a broader range of chemical diversity, allowing for a more rigorous assessment of its robustness. The newly added results, shown in Table R5-1, demonstrate the high robustness of our model on external compound dataset evaluation, while other baseline models suffer noticeable decline in performances.

---

> > ### Author Response · Authors · 2024-11-21
> > **Response to Reviewer vdXE's comments (2)**
> >
> > **Table R5-1.** Cross-dataset evaluation of drug-like compound identification performance on scaffp;d-split setting,
> > trained on PubChem/ChEMBL and evaluated with ZINC20 compounds. Mean and standard deviation of 10
> > fold CV are provided. Best performances are marked in bold.
> > | Train set     |                 | PubChem + DrugBank    |                 |                 | ChEMBL +   DrugBank   |                 |
> > |---------------|-----------------|-----------------------|-----------------|-----------------|-----------------------|-----------------|
> > |               | **F1**          | **Average Precision** | **AUROC**       | **F1**          | **Average Precision** | **AUROC**       |
> > | FP-SVM        | 0.268 (0.0194) | 0.334 (0.1912)       | 0.795 (0.0759) | 0.371 (0.0519) | **0.494 (0.1982)**       | 0.819 (0.0768) |
> > | FP-XGB        | 0.254 (0.0209) | 0.320 (0.1181)        | 0.773 (0.0741) | 0.358 (0.0589) | 0.469 (0.1839)       | 0.814 (0.0784) |
> > | FP-OCSVM      | 0.179 (0.0582) | 0.366 (0.2717)       | 0.576 (0.1949) | 0.179 (0.0582) | 0.366 (0.2717)       | 0.576 (0.1949) |
> > | FP-SVDD       | 0.151 (0.0033) | 0.055 (0.0019)       | 0.235 (0.0173) | 0.151 (0.0033) | 0.055 (0.0019)       | 0.235 (0.0173) |
> > | FP-DeepSVDD   | 0.147 (0.0294) | 0.080 (0.0146)        | 0.415 (0.1224) | 0.147 (0.0294) | 0.080 (0.0146)        | 0.415 (0.1224) |
> > | FP-nnPU       | 0.244 (0.0182) | 0.240 (0.0816)        | 0.749 (0.0556) | 0.327 (0.0525) | 0.380 (0.1999)        | 0.778 (0.0812) |
> > | FP-PU with NN | 0.241 (0.0265) | 0.228 (0.0556)       | 0.702 (0.056)  | 0.311 (0.0495) | 0.396 (0.1701)       | 0.778 (0.0874) |
> > | DeepDL        | 0.170 (0.0199)  | 0.092 (0.0112)       | 0.590 (0.0233)  | 0.195 (0.0389) | 0.102 (0.0196)       | 0.612 (0.0686) |
> > | D-GCAN        | 0.213 (0.0232) | 0.135 (0.0153)       | 0.685 (0.0436) | 0.314 (0.062)  | 0.211 (0.0601)       | 0.737 (0.1076) |
> > |
> > | BOUNDR.E      | 0.496 (0.0287) | 0.444 (0.0303)       | 0.873 (0.0167) | 0.513 (0.0451) | 0.435 (0.0889)       | 0.869 (0.0258) |
> > | BOUNDR.E_MULT  | **0.501 (0.0232)** | **0.460 (0.038)**         | **0.875 (0.0157)** | **0.546 (0.0406)** | 0.484 (0.0729)       | **0.876 (0.0267)** |
> >
> > We also agree that a detailed analysis of the distribution overlap between drugs and compounds can enhance the understanding of our dataset's challenge. To address this, we have included a comparison of key chemical properties—such as molecular weight, scaffold diversity, and chemical descriptors—in the Appendix, which illustrates the dataset characteristics and overlaps between DrugBank, ZINC along with newly introduced ChEMBL and PubChem datasets. These additions aim to clarify the dataset's composition and support the utility of our model across different real-world data.
> >
> > > **Q2.** Regarding Section4.2, three metrics are proposed, including ICR, IDR, and IDR/ICR. I have two doubts. The first one is the proposed EM improvement would benefit identifying identifying the boundary. But w.r.t. ICR, the performance is lower than existing methods like DeepDL and D-GCAN. Does it mean that the drug decision boundary is relative poor? The second question is that in line 357, "IDR/ICR ratio to report the models’ capability in balancing the trade-off between inclusion of drug-like compounds with exclusion of non-drugs.". But, the detection of drug and compounds is not correlated, which means that is it not a binary classification senario like MCC and F1. Also, the ratio of two metric cannot refect the tradeoff. Instead, the sum would be a better one. Would you please provide a detailed explanation on this metric?
> >
> > Thank you for your insightful observations regarding our evaluation metrics. We appreciate the opportunity to elaborate on the rationale behind the use of ICR, IDR, and the IDR/ICR ratio. ICR (In-boundary Compounds Ratio) is actually equivalent to false positive ratio, where lower ICR indicates better exclusion of non-drugs, directly correlating with higher specificity. Conversely, IDR (In-boundary Drugs Ratio) measures the proportion of actual drugs correctly classified within the boundary, akin to True Positive Ratio (TPR). Hence, our intention in the IDR/ICR ratio was to represent the concept of ‘higher TPR and lower FPR is better’, since this balance is crucial in drug discovery.
> >
> > However, in response to your feedback, we recognize that AUROC may provide a more intuitive evaluation of this trade-off. Therefore, **we have incorporated AUROC as a replacement for the low-intuitive IDR/ICR metric in the main results table**. We have also included the Average Precision (AP) metric to offer further insights into the model's ranking and recommendation quality. These adjustments aim to provide a clearer and more comprehensive understanding of how our model balances the inclusion and exclusion of compounds.
> >
> > While the AUROC metric is added to all of our updated tables, we here provide the updated main performance table with AUROC and AP below in Table R5-2.

---

> > > ### Comment · Reviewer_vdXE · 2024-11-22
> > >
> > > The experimental results on different datasets also seem good to me. The problem is fixed.

---

> > ### Comment · Reviewer_vdXE · 2024-11-22
> >
> > Thank you for your response here. I believe I have known your original motivation, and I have no problem towards these three points.

---

> ### Author Response · Authors · 2024-11-21
> **Response to Reviewer vdXE's comments (3)**
>
> **Table R5-2.**  Revised main performance table with AUROC and Average Precision included. (BOUNDR.E$_\text{MULT}$ refers to an extension of our model with multiple initialization strategy, as explained in the response to W3 of Reviewer jyv4.)
> * DrugMetic fails to provide prediction probability nor ranking.
>
> |              | F1 (↑)          | IDR (↑)         | ICR (↓)         | AUROC (↑)       | Avg. Precision (↑) |
> |--------------|-----------------|-----------------|-----------------|-----------------|--------------------|
> | FP-SVM       | 0.665 (0.0126) | 0.823 (0.0111) | 0.067 (0.0052) | 0.963 (0.0021) | 0.724 (0.0174)    |
> | FP-XGB       | 0.692 (0.0141) | 0.815 (0.0205) | 0.055 (0.0048) | 0.966 (0.0026) | 0.775 (0.0213)    |
> | FP-OCSVM     | 0.09 (0.0025)  | 0.274 (0.000)   | 0.489 (0.0101) | 0.331 (0.003)  | 0.148 (0.0022)    |
> | FP-DeepSVDD  | 0.166 (0.0087) | 0.834 (0.035)  | 0.840 (0.0381) | 0.494 (0.0532) | 0.097 (0.0157)    |
> | FP-nnPU      | 0.608 (0.0239) | 0.789 (0.0367) | 0.083 (0.0081) | 0.944 (0.0049) | 0.706 (0.0261)    |
>  |FP-PU with NN| 0.634 (0.0224) | 0.791 (0.0296) | 0.072 (0.0079) | 0.949 (0.0045) | 0.720 (0.0214)    |
> | DrugMetric*  | 0.170 (0.0319) | 0.767 (0.1271) | 0.760 (0.2028) | N/A             | N/A                |
> | D-GCAN       | 0.669 (0.177)  | **0.942 (0.0337)** | 0.160 (0.2808) | 0.918 (0.1396) | 0.613 (0.1874)    |
> | DeepDL       | 0.740 (0.0584) | 0.888 (0.0546) | 0.054 (0.0225) | **0.979 (0.0114)**  | 0.886 (0.0374)   |
> ||
> | BOUNDR.E     | 0.826 (0.0486) | 0.781 (0.0326) | 0.012 (0.0086) | 0.973 (0.0075) | 0.877 (0.0419)    |
> | BOUNDR.E$_\text{MULT}$| **0.846 (0.0165)** | 0.799 (0.0184) | **0.009 (0.0031)** | 0.978 (0.0029) | **0.908 (0.0096)**    |
> |
>
> Accordingly, we have included further details in the metrics and the inclusion of AP, Rec@k and Prec@k in the Results section as following:
>
>     4.2 DRUG-COMPOUND IDENTIFICATION PERFORMANCES
>         ... IDR, equivalent of True Positive Rate (TPR), reflects how well the boundary captures drug-like compounds, while ICR, representing False Positive Rate (FPR), measures how well non-drug compounds are excluded. We also report the AUROC metric to report the models’ capabilities in balancing the trade-off between TPR and FPR. In addition, we also report Average Precision, Recall@k and Precision@k to further evaluate the quality of recommended compounds.
>
>
> > **Q3.** In Figure6, there is performance comparsion between F1 score according to Drug/Compound Ratio. We can see that higher ratio leads to higher difficulty. Futhermore, the proposed method achieves SOTA only when the ratio is over 1:50. Thus, a natural question is that what is the corresponding ratio in your training and testing sets.
>
> Thank you for raising this important question about the Drug/Compound Ratio (DCR) in relation to the performance trends observed in Figure 6. We acknowledge that a clearer explanation of the DCR settings used in both training and testing is crucial for understanding the model's behavior under varying conditions. The experiment in Figure 6 is intended to highlight the robustness of our model under increasing dataset imbalance, reflecting real-world challenges where the volume of non-drug-like compounds far exceeds that of drug-like ones.
>
> To clarify, our training dataset maintained a fixed DCR to ensure stable learning conditions, while the test set's DCR was varied to simulate more challenging scenarios. This variation was used to demonstrate the model's adaptability and its ability to generalize under realistic imbalances.
>
> We note that the figure reflects performance with the biomedical context-aligned embedding space, demonstrating the advantage of using our proposed context-aware method compared to simple fingerprint-based embeddings (as evidenced in Table 1). The higher DCR, moving from 1:10 to 1:50, presents a more challenging setting, which mirrors the realistic drug discovery scenario where modern generative AI models can create vast numbers of potential compounds, heavily biasing the dataset toward non-drug-like entities.
>
> Our method excels in these imbalanced contexts because of its one-class boundary formulation, which allows it to bound the drug-like chemical space effectively, even when faced with an overwhelmingly large non-drug background. Binary classifiers and PU-learning frameworks, which optimize standard risk minimization objectives, tend to struggle under such heavy bias, leading to diminished performance when the dataset skew becomes severe. The exact ratios for training and testing conditions have been detailed in the revised Appendix for transparency.

---

> > ### Author Response · Authors · 2024-11-21
> > **Response to Reviewer vdXE's comments (4)**
> >
> > > **Q4.** In Figure8, the caption states that "BOUNDR.E-filtered set shows more distinct distribution of molecular propertirs from the original 10k molecules." But, in the plot results, the above conclusion seems not able to be easily drawn. May you provide a detailed analysis?
> >
> > Thank you for pointing out the ambiguity in our original phrasing regarding the distributional changes observed in Figure 8. We agree that the use of the term "distinct" is not appropriate to describe the relationship between the distributions in the figure. To address this, we have revised the terminology to "distant," which better captures the subtle shifts observed in molecular properties post-filtering.
> >
> > Additionally, we have included quantitative values of distances using the Wasserstein distance in the figure to objectively measure the distributional changes between the original and BOUNDR.E-filtered datasets. This metric provides a robust assessment of the filtering impact, offering both a numerical and visual complement to the revised Figure 8. These updates are intended to clarify the model’s influence on dataset composition, as well as the practical implications for drug discovery tasks.

---

> > > ### Comment · Reviewer_vdXE · 2024-11-22
> > >
> > > Thank you for your respnse. The problem is fixed.

---

> > > > ### Comment · Reviewer_vdXE · 2024-11-22
> > > > **I would raise my score to 6.**
> > > >
> > > > Since every experiment seems reasonable and the explanation to the technical problem is also reasonable. So, I decide to raise my score to 6.

---

> > > > > ### Author Response · Authors · 2024-11-23
> > > > > **Appreciation for Your Consideration and Constructive Feedback.**
> > > > >
> > > > > Dear Reviewer,
> > > > >
> > > > > Thank you very much for taking the time to read and consider our rebuttal. We truly appreciate your thoughtful engagement with our work and your willingness to adjust the evaluation. We understand how challenging it can be to shift one's perspective, especially in this open review environment. Your constructive feedback has been invaluable in refining our manuscript.
> > > > >
> > > > > If there are any additional insights or questions you would like to discuss during the discussion phase, we would be more than happy to address them. Your input has already significantly contributed to strengthening our submission, and we welcome any further guidance you might have.
> > > > >
> > > > > Thank you once again for your consideration.

---

> > ### Comment · Reviewer_vdXE · 2024-11-22
> >
> > Thank you for your experimental results. The results seem good, and the revised metrics are also good to me. The problem is fixed.

---

### Official Review · Reviewer_T5wN · 2024-11-03

**Soundness:** 2
**Presentation:** 2
**Contribution:** 2
**Rating:** 3
**Confidence:** 3

**Summary:**

This paper addresses the challenge of accurately identifying drug candidates, an essential task in the drug discovery process. The authors introduce BounDr.E, a novel method designed to construct a compact boundary of drug-likeness within the chemical space. The approach utilizes an Expectation-Maximization (EM)-like iterative optimization process combined with metric learning to refine both the drug-like boundary and the embedding space distribution. By incorporating biomedical context through multi-modal alignment with knowledge graphs, BOUNDR.E achieves significant improvements in drug-likeness prediction, reportedly up to five orders of magnitude in performance metrics. The model's effectiveness is further demonstrated through zero-shot toxic compound filtering and detailed case studies, showcasing its value in large-scale screening of AI-generated compounds.

**Strengths:**

The paper is thorough, offering extensive details on the algorithm, experimental settings, and data used, making it self-contained and easy to follow.

The proposed method builds upon a foundation of established works, such as DREAMwalk and Geodesic Mixup, which contributes to its enhanced performance.

Additionally, the authors have provided theoretical justification for the M-step of the EM-like algorithm, adding further depth to their approach.

The code is provided, ensuring the reproducibility of this work.

**Weaknesses:**

Relying solely on a fully data-driven approach to estimate drug-likeness might be risky and may deviate from a more rational pathway for drug discovery. A more effective strategy for screening drug candidates involves predicting specific properties—such as toxicity and the binding affinity of a drug to specific targets—and leveraging these properties to filter molecule databases, thereby identifying promising drug candidates more reliably. This may indicate the limitation of the proposed framework in real-world practice of drug discovery.

The experiments can be more extensive. One limitation noted is the demonstration, using only one specific example, where the average probability density function (pdf) of Imatinib increased after filtering with the proposed methods. Relying on a single example may not comprehensively illustrate the method’s effectiveness or generalizability.

**Questions:**

See the weaknesses above.

**Details Of Ethics Concerns:**

No ethics concerns.

---

> ### Author Response · Authors · 2024-11-21
> **Response to Reviewer T5wN's comments (1)**
>
> We sincerely thank the reviewers for their insightful comments and constructive feedback, which enabled us to improve our work further. In the following comments, we respond to each comment for your initial assessment. A revised manuscript will be available shortly, incorporating these modifications.
>
> **Weaknesses**
> > Relying solely on a fully data-driven approach to estimate drug-likeness might be risky and may deviate from a more rational pathway for drug discovery. A more effective strategy for screening drug candidates involves predicting specific properties—such as toxicity and the binding affinity of a drug to specific targets—and leveraging these properties to filter molecule databases, thereby identifying promising drug candidates more reliably. This may indicate the limitation of the proposed framework in real-world practice of drug discovery.
>
> We acknowledge the reviewer's concern regarding the reliance on a purely data-driven approach for estimating drug-likeness, especially considering the value of property-specific predictions in traditional drug discovery pipelines. By reviewing your comments, we were able to develop and investigate the potentials of our model as a **complementary data-driven filter in a rational drug discovery pipeline.** To be specific, our model can serve as an efficient, early-stage triaging tool that can significantly narrow down the search space in large chemical libraries, thereby easing the computational burden on subsequent analyses.
>
> To substantiate this point, we performed two additional sets of experiments:
> 1. In this experiment, we evaluated BounDr.E’s filtering capacity against several property-based criteria—predicted drug-target interaction, adherence to the Rule of 5, and PAINS filter—using a dataset of 10,000 AI-generated molecules. BounDr.E produced a filtered set of 300 compounds, the smallest set among the filters, as shown in Table R4-1. The results indicate that BounDr.E effectively reduces the number of molecules that require further, more expensive evaluation, lowering the cost of downstream pipeline.
>
> 2. To simulate a realistic drug discovery scenario, we applied BounDr.E to pre-filter the 10k generated molecules. This pre-filtering was followed by traditional property-based methods. As a result, the combined pipeline, first applying BounDr.E, then sequentially applying IC50, Ro5 and PAINS filters, returned a manageable set of approximately 30 compounds deemed highly likely to succeed in further drug development stages (All filters; Last row of Table R4-1). This outcome illustrates how BounDr.E optimizes the workflow by minimizing the initial candidate pool for downstream experimental validation and simultaneously saving computational resources and time, reducing further screening set from 10k to around 300.
>
> These experiments validate BounDr.E’s role in a hybrid screening approach, highlighting its complementary nature to property-based filtering in practical drug discovery applications. We have replaced the previous case study into this new finding based on your comments, and hope it demonstrates how our model can be leveraged to rational drug discovery pipelines.
>
> > The experiments can be more extensive. One limitation noted is the demonstration, using only one specific example, where the average probability density function (pdf) of Imatinib increased after filtering with the proposed methods. Relying on a single example may not comprehensively illustrate the method’s effectiveness or generalizability.
>
> We acknowledge the shortage in case studies that are insufficient to prove our model’s utility in drug discovery. In response to the reviewer's concerns about the demonstration’s scope, we have **expanded our experiments to include two additional well-known anti-cancer drug targets for erlotinib and riboclicib**, extending from the initial example of imatinib. These targets, along with their corresponding approved drugs, were selected to mimic actual stages of drug discovery pipelines. Detailed results are presented in the Appendix section.
>
> For each target, BounDr.E's filtering capacity was assessed and validated. Similar to the Imatinib case, BounDr.E demonstrated a consistent pattern in enhancing the probability density function (pdf) of known drugs within the filtered set. This consistency across different targets substantiates the method’s reliability in various contexts, demonstrating its ability to generalize beyond a single case study.
>
> Additionally, we tested BounDr.E’s utility in a realistic drug screening scenario, combining it with traditional property-based filters post hoc. In two separate cases, the pipeline returned approximately 20 and 30 candidates for downstream experimental validation. These results not only reinforce BounDr.E’s predictive power but also illustrate its practical value in pre-filtering, producing a focused set of candidates that align with property-specific predictions while reducing computational cost.

---

> > ### Author Response · Authors · 2024-11-21
> > **Response to Reviewer T5wN's comments (2)**
> >
> > **Table R4-1.** The number and the ratio of remaining compounds after applying diverse filters to the 10k Targetdiff-generated molecules for three protein targets. Final row `All filters` refers to the pipeline of first applying BounDr.E, then sequentially applying IC50, Ro5 and PAINS filters.
> >
> > | Filtering Method |    BCR-ABL     |      EGFR      |      CDK6      |
> > |------------------|:--------------:|:--------------:|:--------------:|
> > | Total Generated  |  10,543 (100%) |  12,550 (100%) |  11,496 (100%) |
> > | PAINS filter     | 10,078 (95.7%) | 11,878 (94.6%) | 10,996 (95.6%) |
> > | Ro5              |  4,997 (47.5%) |  6,520 (52.0%) |  5,782 (50.3%) |
> > | Predicted IC50   |  2,786 (26.5%) |   1,018 (8.1%) |  4,734 (41.2%) |
> > | BounDr.E         |     300 (2.8%) |    374 (3.00%) |     264 (2.3%) |
> > | **All filters**      |     **38 (0.36%)** |     **17 (0.15%)** |     **47 (0.40%)** |
> > |
> >
> > Through these additional experiments, we have demonstrated that BounDr.E maintains robust performance across multiple scenarios, supporting its applicability in diverse drug discovery contexts. We hope this broader validation addresses the reviewer’s concerns regarding generalizability and showcases the method’s effectiveness.

---

> ### Comment · Reviewer_T5wN · 2024-11-26
>
> The authors claim that no attempts have been made to align the biomedical knowledge graph embedding space with the molecular structural embedding space for downstream tasks. However, to my knowledge, several studies have already explored unifying molecular structures with knowledge graphs for tasks like property prediction [1,2,3]. Additionally, multi-modal alignment with contrastive learning is not a novel approach.
>
>
>
>
> Drug discovery is fundamentally a multi-objective optimization problem. In its early stages, the goal is not to minimize the number of molecules screened, but to ensure that the selected molecules meet the property requirements for further drug development. The results in Table R4-1 only compare the number of remaining molecules after filtering, which cannot demonstrate the effectiveness of BounDr.E.
>
> As a data-driven virtual screening method, BounDr.E lacks the interpretability for the predicted results compared with property-driven approaches [4].  This creates significant challenges for subsequent drug development; merely being close to known drugs in the "latent space" does not provide meaningful insights for drug discovery on a specific target, as drug targets are highly specific.
>
>
>
> What advantage does BoundDr.E offer when used alongside other property-driven approaches, beyond simply yielding fewer filtered samples? Can BoundDr.E enhance screening outcomes by selecting molecules with better drug-like properties? The presented results (e.g., in Table 5) do not seem convincing enough to demonstrate the superiority of this model.
>
> Defining "drug-like" molecules is inherently challenging, and I question the approach of setting a boundary to filter candidate molecules rather than using distance-based measurements. How do we truly measure "likeness" and what makes one molecule more "drug-like" than another? Property-based methods may address this issue more effectively, providing a clearer rationale for why a given molecule may not qualify as a potential drug. This work also doesn't provide structural analysis (e.g. scaffold, functional motif, non-covelant interactions) of filtered molecules to support the drug-likeness results. As I have mentioned, relying solely on a fully data-driven approach to estimate drug-likeness may be risky and insufficiently convincing. The authors should consider analyzing the results using domain knowledge or comparing them with traditional methods to demonstrate the effectiveness of their approach [5,6].
>
>
>
> To evaluate the properties of filtered molecules, the authors only tested QED, Ro5, and toxicity. However, I believe these metrics alone are insufficient to demonstrate a molecule's drug-likeness. Binding affinity, for example, is a crucial evaluation metric and would provide a more comprehensive assessment of the molecules' potential.
>
> Given the concerns outlined above, I believe the paper does not meet the acceptance standards of ICLR. Therefore, I am maintaining my current score of 3.
>
>
>
> References:
>
> [1] Qing Ye, Chang-Yu Hsieh, Ziyi Yang, Yu Kang, Jiming Chen, Dongsheng Cao, Shibo He, and Tingjun Hou. A unified drug–target interaction prediction framework based on knowledge graph and recommendation system. Nature communications, 12(1):6775, 2021
>
> [2] Fang, Y., Zhang, Q., Yang, H., Zhuang, X., Deng, S., Zhang, W., Qin, M., Chen, Z., Fan, X., & Chen, H. (2022). Molecular Contrastive Learning with Chemical Element Knowledge Graph. Proceedings of the AAAI Conference on Artificial Intelligence, 36(4), 3968-3976.
>
> [3] Yin Fang, Qiang Zhang, Ningyu Zhang, Zhuo Chen, Xiang Zhuang, Xin Shao, Xiaohui Fan, and Huajun Chen. Knowledge graph-enhanced molecular contrastive learning with functional prompt. Nature Machine Intelligence, pp. 1–12, 2023.
> Jiang, Pengcheng, et al. "Bi-level contrastive learning for knowledge-enhanced molecule representations." arXiv preprint arXiv:2306.01631 (2023).
>
>
> [4] Jiménez-Luna, José, Francesca Grisoni, and Gisbert Schneider. "Drug discovery with explainable artificial intelligence." Nature Machine Intelligence 2.10 (2020): 573-584.
>
> [5] Muegge, Ingo. "Selection criteria for drug‐like compounds." Medicinal research reviews 23.3 (2003): 302-321.
>
> [6] Ursu, Oleg, et al. "Understanding drug‐likeness." Wiley Interdisciplinary Reviews: Computational Molecular Science 1.5 (2011): 760-781.

---

> > ### Author Response · Authors · 2024-11-27
> > **Response to reviewer T5wN's comments 2 (1/2)**
> >
> > ### Summary of response to reviewer T5wN
> > The reviewer expresses several concerns regarding our work:
> > 1. the focus on drug-likeness prediction over traditional property-driven methods,
> > 2. the risk of relying solely on a data-driven approach, and
> > 3. the adequacy of metrics used for evaluating drug-likeness over property-based metrics.
> >
> > While the reviewer continuously expresses that our data-driven approach does not meet the standards of the conference, we are confident that our work aligns with the objectives of an AI-centric conference like ICLR, where we believe the emphasis is on **pushing the boundaries of data-driven methodologies.** Since the reviewer strongly favors traditional property-driven methodologies over data-driven approaches, our response is to persuade that drug-likeness prediction, as a multi-objective optimization problem, can be aided by a more holistic and scalable approach, particularly in AI-driven contexts. The scalability, adaptability, and efficiency of our model provide a solid foundation for AI-driven drug discovery, and we believe these aspects are deserving of recognition.
> >
> > 1. Our model’s primary innovation is constructing an approved-drug chemical subspace with biomedical knowledge integration, which ensures alignment with practical drug development pipelines and mitigates risks in later phases including withdrawal, **which traditional property-driven metrics including binding affinity and DTI filters often fail to.**
> >
> > 2. As showcased in the experiment case study (Section 4.7), our model is designed to **complement later-stage property-driven evaluations including binding affinity and scaffold analysis** by efficiently narrowing down candidates, enabling more focused and resource-effective experimental validation to drug-like candidates.
> >
> > We provide the detailed response to the points below.
> >
> > ---
> >
> > > Comment on Drug-Likeness Prediction as a Multi-Objective Optimization Problem
> >
> > We agree with the reviewer that drug-likeness prediction inherently involves multi-objective optimization. However, relying solely on individual properties such as binding affinity for early-stage candidate selection poses risks, particularly in later phases of drug development, where toxicity and off-target effects often emerge. By constructing a boundary that encapsulates the chemical subspace of approved drugs, our model offers a comprehensive filtering mechanism, recommending compounds that align with this constructed space. This approach mitigates the risk of late-stage failure by integrating various drug-likeness dimensions into a unified framework as demonstrated by experiments (Table 3). This holistic filtering process is critical in AI-driven drug discovery, where the volume and diversity of generated compounds necessitate efficient and scalable screening methodologies.
> >
> >
> > > Clarification on the Novelty of Our Multi-Modal Alignment
> >
> > We acknowledge that our manuscript may have overemphasized novelty in multi-modal alignment. However, the core innovation lies in extending this alignment for drug-likeness prediction—a domain distinct from the well studied drug-target interaction and element knowledge integration. While existing methods unify molecular and knowledge graph embeddings for property prediction, our work is unique in integrating these spaces to define a boundary for drug-likeness. This approach is not merely about alignment but about constructing a predictive boundary that effectively handles large-scale, generative AI datasets, a challenge not adequately addressed by traditional methods.
> >
> > We have revised our models' claims in the Related Work sections below, with additional discussion on the cited works in the comment:
> > ```
> >     Recently, contrastive learning has also been actively integrated into the fields of drug-target interaction prediction and element knowledge integration. Despite the significant advancements in multi-modal learning, there has not been an attempt to extend such concepts to align the knowledge graph embedding space with the structural embedding space through multimodal-alignment to construct an approved drug chemical space for drug-likeness prediction.
> > ```
> >
> > > Preference of Property-Driven Methods over Data-driven Approach
> >
> > We respectfully contend that focusing narrowly on property-driven methods for early-stage filtering overlooks the broader challenges posed by generative AI models in drug discovery. **Our approach complements property-based methods by providing a scalable solution that reduces false positives early in the pipeline.** As noted, optimizing isolated properties does not guarantee drug-likeness; many candidates meeting property criteria still fail in preclinical and clinical trials due to unforeseen toxicity or off-target effects. Our boundary model excels in this early-stage filtering of false positives (Tables 1~3), helping identify viable candidates more efficiently, reducing the experimental burden, and improving the success rate downstream.

---

> > > ### Author Response · Authors · 2024-11-27
> > > **Response to reviewer T5wN's comments 2 (2/2)**
> > >
> > > > Rationale for Boundary-Based Classification Over Distance-Based Measures
> > >
> > > The reviewer’s preference for distance-based measurements assumes a linear relationship between molecular proximity and drug-likeness, which is insufficient in the complex, high-dimensional space of molecular embeddings. **Our boundary-based approach is aimed to overcome the arbitrary nature of distance thresholds, offering a clearer, non-parametric threshold that captures the distribution of approved drugs.** By defining this boundary, we provide a more interpretable and robust framework for filtering compounds without the ambiguities of distance metrics, aligning with AI-centric drug discovery’s need for scalable and precise solutions.
> > >
> > >
> > > > Concerns on Choice of Drug-likeness Evaluation Metrics
> > >
> > > While the reviewer suggests metrics like binding affinity for a more comprehensive assessment, such metrics rely on time and labor-intensive experimental data, impractical for the large-scale filtering required in early-stage AI-driven drug discovery. Our selected metrics—QED, Ro5, and toxicity, along with **newly added Syntheic Accessibility Score (SAS) for further cheminformatic demonstration (Appendix E.6)**—provide a broad yet computationally feasible evaluation framework.
> > >
> > >     [Section 4.7]
> > >     Furthermore, the filtered compound list yield a more distant distribution of compounds from the initially generated molecules, showing higher traditional measures of QED, Rule-of-five and Synthetic Accessibility Scores (SAS), ...
> > >
> > > These metrics align with the conference’s AI-centric focus, offering a balance between scalability and practical relevance. Moreover, our model is designed to complement later-stage property-driven evaluations including binding affinity and scaffold analysis by efficiently narrowing down candidates, enabling more focused and resource-effective experimental validation. After initial screening of potential false positives with our model, a property-based filter can then be applied to this higher drug-like set to yield high efficiency and promising drug-like candidates.
> > >
> > > In summary, our work aligns with the objectives of an AI-centric conference like ICLR, where the emphasis is on pushing the boundaries of data-driven methodologies. The scalability, adaptability, and efficiency of our model provide a solid foundation for AI-driven drug discovery, and we believe these aspects are deserving of recognition.
> > >
> > > We believe our work contributes meaningfully to the AI field's ongoing efforts in drug discovery.

---

### Official Review · Reviewer_mQom · 2024-11-04

**Soundness:** 2
**Presentation:** 4
**Contribution:** 2
**Rating:** 3
**Confidence:** 4

**Summary:**

This paper poses the main challenges for the drug-compound identification task: the absence of a clear negative class and the vastness of chemical space. This paper proposes a targeted solution, using an EM-like optimization process to avoid strict decision boundaries. In addition, a contribution is to propose a novel multimodal alignment module to more effectively capture drug-like related features. Experiments have demonstrated the effectiveness of the proposed method.

**Strengths:**

1.  Sound motivation and clear writing. I can clearly understand the problem that this article focuses on. I also agree that this is the main challenges of this task. I can quickly and easily get the technical details (method part) of this article.

2. The author provides detailed introduction to the various technologies and designs used, which is very reader-friendly.

3. Experimental results show obvious advantages from proposed method. Code is provided and is (at least) runnable.

**Weaknesses:**

1. By reading the setting and challenges of the task, I would like to question whether this is a standard PU learning setting. If so, the author could have directly introduced this common scenario (if I am wrong, please correct me). In addition, line106 points out that the existing PU learning methods rely on negative samples, which is confusing. As far as I know, the **training process** of PU learning does not provide negative samples.

2. This point is associated with weakness 1. I do not quite understand the problem setting. This paper claims that most existing methods rely on defined negative samples. So is there any difference in problem setting from BounDr.E and others? To my understand, this task suggest training with positive and unlabeled samples and testing with defined positive and negative ones. If the settings among papers are consistent, you can directly say that something like "The drug-compound identification task are commonly formulated as a PU learning setting. However, existing methods xxxxxx". If not, perhaps you should first introduce the difference.

3. Question about the experiment setup. Drugbank and Zinc datasets are used, but it seems like there is still no defined negative samples. If yes, how do you get them? I personally think that it is not difficult to create some real negative samples for the test set. The simplest way is to predict or query whether a molecule is toxic. We can regard toxic molecules as absolutely non-drug (for training and testing). If your test set does not have absolute negative samples, you should not consider metrics such as F1, but should refer to the ones such as recall in the recommendation system task. If the test set has negative samples, the metric you use is reasonable.

**Questions:**

N.A.

---

> ### Author Response · Authors · 2024-11-21
> **Response to Reviewer mQom's comments (1)**
>
> We sincerely thank the reviewers for their insightful comments and constructive feedback, which enabled us to improve our work further. In the following comments, we respond to each comment for your initial assessment. A revised manuscript will be available shortly, incorporating these modifications.
>
> **Weaknesses**
> > W1. By reading the setting and challenges of the task, I would like to question whether this is a standard PU learning setting. If so, the author could have directly introduced this common scenario (if I am wrong, please correct me). In addition, line106 points out that the existing PU learning methods rely on negative samples, which is confusing. As far as I know, the training process of PU learning does not provide negative samples.
>
> We acknowledge the reviewer’s inspiring comments on the similarities between our formulation of drug-likeness prediction and standard PU learning frameworks. Thanks to the comment, we took the time to clearly define the distinction between our approach and PU learning approaches. In extent, we have refined our explanation in the manuscript, including a table summarizing the key differences between them.
>
> Our problem setting roots on the idea to rescue any non-drugs from the compound libraries by not treating any as ‘negative drugs’. This motivation naturally led us to apply an one-class classification based approach.
>
> On the other hand, PU learning typically assumes that the distribution of unlabeled data, $P_{\text{unlabeled}}$, can be expressed as a mixture model: $P_{\text{unlabeled}} \sim P_\text{positive}$ + $P_\text{negative}$. This leads to training objectives rooted in empirical risk minimization that assume a tractable and bounded space of both positive and negative examples with the dataset as a representative subset of such space. In this context, PU methods often aim to minimize classification error with cross entropy-based loss functions by estimating the contribution of a negative distribution, frequently relying on class prior (ratio of positive/negative in the dataset) estimates.
>
> In contrast, our formulation of the drug-likeness prediction task does not assume a well-defined $P_\text{negative}$. The chemical space is vast, partially explored, and inherently complex, with any sampled "negative" set non-representative of the true distribution of non-drug compounds. Therefore, instead of attempting to estimate a boundary between positive and potential negatives, we propose a one-class classification framework that constructs a drug-likeness boundary to capture the compact space of drug-like compounds directly, optimized based on distance-based metric learning terms.
> **Table R3-1**. Key differences between PU-learning setting and proposed definition of drug-likeness prediction.
> |  | Binary classification | PU-learning | Drug-likeness prediction|
> |-|---|---|-|
> | **Goal** | Decision boundary between positive and negative | Decision boundary between positive and unseen negative | Boundary around positives (here, approved drugs)
> | **Train set composition** | Positive + Negative | Positive + Unlabeled | Drug + Compound
> | **Positive data distribution** |Positives ($P_\text{positive}$)| Positives ($P_\text{positive}$) | $X_\text{drugs}$ as subset of $X_\text{compound}$
> |**Unlabeled data distribution**| - (Only negative data) | $P_\text{positive} + P_\text{negative}$ (unseen)| $X_\text{compound}$
> | **Assumption of unlabeled dataset** | - | Representative of $P_\text{positive}$ and $P_\text{negative}$ | Biased subset of intractable $X_\text{compound}$
> | **Characteristics** | Strong reliance to negative set, lower generalizability | Reliance to unlabeled set, lower generalizability | Low reliance to compound set, higher generalizability
> | **Objective** | Risk minimization with cross-entropy | Risk minimization with class prior and cross-entropy| Metric learning (one-class hypersphere)
> |
>
> Our additional clarification in the manuscript highlights that drug-likeness prediction requires a fundamentally different approach from PU learning, specifically due to the unbounded nature of the chemical space. We hope this clarification resolves any confusion and illuminates why we utilized a one-class classification model grounded in metric learning rather than a conventional PU framework.
>
>
>     INTRODUCTION: Existing methods for drug-likeness prediction tend to fall short. Supervised models (Sun et al., 2022) often generate overly strict decision boundaries by treating unlabeled compounds as hard negatives, while PU learning approaches (Lee et al., 2022) also assume the unlabeled set as a mixture of tractable positive and negative label distribution, which is unpractical in compound space. Both approaches root on risk minimization which enforces their reliance on the negative set. On the other hand, unsupervised models (Li et al., 2024) produce overly broad boundaries that fail to generalize.

---

> ### Author Response · Authors · 2024-11-21
> **Response to Reviewer mQom's comments (2)**
>
> > **W2.** This point is associated with weakness 1. I do not quite understand the problem setting. This paper claims that most existing methods rely on defined negative samples. So is there any difference in problem setting from BounDr.E and others? To my understand, this task suggest training with positive and unlabeled samples and testing with defined positive and negative ones. If the settings among papers are consistent, you can directly say that something like "The drug-compound identification task are commonly formulated as a PU learning setting. However, existing methods xxxxxx". If not, perhaps you should first introduce the difference.
>
> We appreciate the reviewer’s detailed suggestion to clarify the differences in problem settings between BounDr.E and other drug-likeness prediction frameworks. Following this comment and the one above, the manuscript has been revised to provide a clearer comparison.
>
> Conventional methods in drug-likeness prediction mainly employ binary classification and sometimes Positive-Unlabeled (PU) learning frameworks, seeking to classify compounds by minimizing the risk of misclassification between positive (drug-like) and negative (non-drug-like) examples with cross entropy-based objectives. However, these approaches rely on defined negative sets or a representative dataset from $P_\text{negative}$ distribution, which may not be feasible in the vast and partially known chemical space.
>
> To address this limitation, we proposed a novel one-class boundary approach with metric learning objectives, diverging from conventional frameworks. By focusing on capturing the compact space of known drugs, our model does not rely on the need for an explicit negative set, aligning with the challenges of drug-likeness prediction. We hope this clarification solidifies our problem formulation and the rationale behind our methodological choices.
>
>
>
> > $W3.$ Question about the experiment setup. Drugbank and Zinc datasets are used, but it seems like there is still no defined negative samples. If yes, how do you get them? I personally think that it is not difficult to create some real negative samples for the test set. The simplest way is to predict or query whether a molecule is toxic. We can regard toxic molecules as absolutely non-drug (for training and testing). If your test set does not have absolute negative samples, you should not consider metrics such as F1, but should refer to the ones such as recall in the recommendation system task. If the test set has negative samples, the metric you use is reasonable.
>
> Thank you for your insightful feedback regarding the evaluation setup and the absence of clear negative samples. This touches on a key challenge in our formulation of drug-likeness prediction problems.
>
> In the vast chemical space, no definitive negative set exists, leading different studies to use diverse sources like ZINC20 or PubChem as "non-drug" examples for evaluation. Several prior studies have applied diverse "non-drug" sets as negatives, yet resulted in inconsistent performances and have not reached an agreement on a representative dataset. In contrary, we chose not to define any specific compounds as absolute negatives due to the inherent ambiguity of non-drug space, where any compound can be a drug candidate. Considering your comments, we have added new experimental results using PubChem compounds as non-drug training samples, available in the Appendix, to show the robustness of our model under diverse conditions.
>
> Regarding the comment on utilization of toxic molecules as negative samples, while they are indeed non-drug-like, using them as negative sets would shift the focus of the model to toxicity prediction rather than drug-likeness. Meanwhile, the comparison results of accurately filtering the four categories of toxic compounds (withdrawn, cardiotoxic, hepatotoxic, carcinogens) for baseline models are provided in Section 4.3 of the initial submission.
>
> Following the recommendations, to illustrate the effectiveness of our model without relying on arbitrary negative samples, we incorporated metrics commonly used in recommendation systems; average precision, precision@k and recall@k . These metrics measure how well our model identifies drug-like compounds among a broad, potentially drug-laden chemical space. The average precision values are displayed in the main table of the manuscript while precision@k and recall@k metrics are further provided in the Appendix. We provide the revised main table below, which incorporates Average Precision as main metric below:

---

> ### Author Response · Authors · 2024-11-21
> **Response to Reviewer mQom's comments (3)**
>
> **Table R3-2.** Revised main performance table with AUROC and Average Precision included. (BOUNDR.E$_\text{MULT}$ refers to an extension of our model with multiple initialization strategy, as explained in the response to W3 of Reviewer jyv4.)
> * DrugMetic fails to provide prediction probability nor ranking.
>
> |              | F1 (↑)          | IDR (↑)         | ICR (↓)         | AUROC (↑)       | Avg. Precision (↑) |
> |--------------|-----------------|-----------------|-----------------|-----------------|--------------------|
> | FP-SVM       | 0.665 (0.0126) | 0.823 (0.0111) | 0.067 (0.0052) | 0.963 (0.0021) | 0.724 (0.0174)    |
> | FP-XGB       | 0.692 (0.0141) | 0.815 (0.0205) | 0.055 (0.0048) | 0.966 (0.0026) | 0.775 (0.0213)    |
> | FP-OCSVM     | 0.09 (0.0025)  | 0.274 (0.000)   | 0.489 (0.0101) | 0.331 (0.003)  | 0.148 (0.0022)    |
> | FP-DeepSVDD  | 0.166 (0.0087) | 0.834 (0.035)  | 0.840 (0.0381) | 0.494 (0.0532) | 0.097 (0.0157)    |
> | FP-nnPU      | 0.608 (0.0239) | 0.789 (0.0367) | 0.083 (0.0081) | 0.944 (0.0049) | 0.706 (0.0261)    |
>  |FP-PU with NN| 0.634 (0.0224) | 0.791 (0.0296) | 0.072 (0.0079) | 0.949 (0.0045) | 0.720 (0.0214)    |
> | DrugMetric*  | 0.170 (0.0319) | 0.767 (0.1271) | 0.760 (0.2028) | N/A             | N/A                |
> | D-GCAN       | 0.669 (0.177)  | **0.942 (0.0337)** | 0.160 (0.2808) | 0.918 (0.1396) | 0.613 (0.1874)    |
> | DeepDL       | 0.740 (0.0584) | 0.888 (0.0546) | 0.054 (0.0225) | **0.979 (0.0114)**  | 0.886 (0.0374)   |
> ||
> | BOUNDR.E     | 0.826 (0.0486) | 0.781 (0.0326) | 0.012 (0.0086) | 0.973 (0.0075) | 0.877 (0.0419)    |
> | BOUNDR.E$_\text{MULT}$| **0.846 (0.0165)** | 0.799 (0.0184) | **0.009 (0.0031)** | 0.978 (0.0029) | **0.908 (0.0096)**    |
> |
>
> Our manuscript now includes a comprehensive evaluation with both classification and recommendation-based metrics, aiming to position drug-likeness prediction as a task that balances these perspectives. Through considering the comment, we believe this improved dual evaluation framework strengthens our approach's relevance and addresses the reviewer's concerns.

---

> > ### Author Response · Authors · 2024-11-30
> > **Gentle reminder for Reviewer mQom**
> >
> > Dear Reviewer mQom,
> >
> > We thank you for your effort in reviewing our paper and providing us with valuable feedback. We have dedicated all our efforts in thoroughly revising our work based on each of the 3 points you raised.
> >
> > Before the **end of the discussion in three days**, your response on whether we have adequately addressed your concerns would be much appreciated.
> >
> > Thank you for your continued consideration.

---

### Official Review · Reviewer_XeZc · 2024-11-04

**Soundness:** 3
**Presentation:** 3
**Contribution:** 2
**Rating:** 5
**Confidence:** 3

**Summary:**

The authors present a method to predict drug-likeness. Precisely, they a) use a contrastive learning scheme to learn a joint embedding space for molecular structure and biomedical context, and b) learn a decision boundary by an EM algorithm. The authors show that their approach outperforms compared methods on the task of distinguishing DrugBank molecules from ZINC molecules. Also they showed that their method is capable of filtering toxic compounds.

**Strengths:**

- **Clarity**:  The paper is generally well-written. The scope of the work and the research question are clearly presented. The proposed method, including the multi-modal alignment and the EM-algorithm-based decision boundary, is well described.
- **Significance**: The presented method clearly outperforms compared methods. Results are presented with error bars (standard deviation across cv).
- **Relevance**: Drug likeness is an important property to consider in drug discovery projects. Therefore, improved methods to detect drug likeness are desirable.

**Weaknesses:**

- **Little Novelty**:  There is a lot of work wrt aligning two modalities with contrastive learning methods. E.g. the presented idea of softening the CLIP matching has been done before [1].
- **Clarity**: The presented method is provided with enriched information about biomedical context. The authors missed to explain whether baseline method did get a fair chance to operate on this enriched data base as well. Could have performance gains just have arisen because the presented method is provided with more information (biomedical context)?
- **Experiments**: The ablation study is informative. However it would have been interesting to train a MLP (i.e. a universal function approximator) on top of the embeddings from the contrastive learning space to estimate the effectiveness of the EM-based algorithm. What is the intuition why a MLP should not be capable of learning a proper boundary? Is it that not only the boundary but also the encoder is iteratively updated in the EM algorithm? If so, can it be shown that this is the critical point?

[1] CLOOME: contrastive learning unlocks bioimaging databases for queries with chemical structures

**Questions:**

- Could have performance gains just have arisen because the presented method is provided with more information (biomedical context)?
- What is the intuition why a MLP should not be capable of learning a proper boundary? Is it that not only the boundary but also the encoder is iteratively updated in the EM algorithm? If so, can this be somehow shown?
- What if the structural and biomedical representations are naively concatenated and fed into an MLP? Shouldn't this approach be able to get  similar performance values compared to the presented one without the need of the two-stage training procedure?

Comment about hard negatives: Also the proposed method is trained in a way strictly treating non drug-like molecules as negatives while the EM algorithm adjusts the decision boundary. (Compare l72ff - related work: supervised models treat unlabeled compounds as hard negatives.)

---

> ### Author Response · Authors · 2024-11-21
> **Response to Reviewer XeZc's comments (1)**
>
> We sincerely thank the reviewers for their insightful comments and constructive feedback, which enabled us to improve our work further. In the following comments, we respond to each comment for your initial assessment. The revised manuscript will be available shortly, incorporating all the updates.
>
> **Weaknesses**
> > **W1.** Little Novelty: There is a lot of work wrt aligning two modalities with contrastive learning methods. E.g. the presented idea of softening the CLIP matching has been done before [1].\
>
> We believe our presentation of our work’s novelty was short in convincing the readers. We aimed to enhance the clarity of our manuscript in improving the presentation of our work’s novelties in the Introduction by clearly including the contributions of our work:
> ```
> INTRODUCTION
>     ... Our contributions can be summarized as following:
> 1. Novel formulation of drug-likeness prediction as a one-class classification without reliance on negatives.
> 2. Proposal of EM-like optimization of both the drug-likeness boundary and the embedding space for accurate drug-likeness prediction.
> 3. Knowledge-integrated multi-modal alignment of structure and biomedical knowledge embeddings for defining drug-likeness with machine learning.
> ```
> We hope this improved clarity helps the readers to understand the novelties of our work.
>
> Furthermore, we appreciate the reviewer's observation regarding the similarity of our approach to existing work in contrastive learning for modality alignment. While we acknowledge that softening CLIP matching has indeed been explored before, as in CLOOME , we believe our approach offers several key innovations:
>
> 1. Unlike CLOOME, which focuses on bioimaging databases and chemical structures, our work specifically addresses the alignment of drugs' biomedical knowledge with structural embeddings. This domain-specific application for defining drug-likeness with machine learning presents unique challenges and opportunities.
> 2. While CLOOME's loss term is fully data-driven, our modified CLIP loss is augmented with knowledge-based information derived from the Anatomical Therapeutic Chemical (ATC) classification system. This integration of domain knowledge into the loss function is a novel aspect of our approach, compared to fully data-driven approaches including CLIP and CLOOME.
> 3. A core innovation of our work is the application of a one-class classification framework optimized through an EM-like approach. This is fundamentally different from previous attempts that have used binary classifiers or graph neural networks. We believe that while our problem formulation of drug-likeness prediction itself is novel, and applying a one-class boundary to meet the challenges is an unprecedented approach for the essential problem in AI driven drug discovery.
> 4. Our ablation studies (which we will discuss in more detail in response to the next comment) demonstrate that the combination of our multi-modal alignment and EM-like optimization yields performance gains that surpass those achieved by either component alone.
>
> We have included these aspects, especially comparison with the CLOOME model, into our Related Works section to further provide the distinct approach of our knowledge-integrated CLIP.
>
>     2. RELATED WORKS
>     Deep Multi-modal alignment
>         ... In the biochemical domain, CLOOME (Sanchez-Fernandez et al., 2023) has been proposed to modify the CLIP loss through leave-one-out boosting with continuous modern Hopfield networks for chemical and bioassay image alignment.
>         Despite the significant advancements in multi-modal learning, there has not been an attempt to align the knowledge graph embedding space with the structural embedding space through knowledge-integrated contrastive learning for downstream tasks including drug-likeness identification.
>
> We believe these aspects collectively contribute to the novelty and significance of our work in the context of AI-driven drug discovery.

---

> > ### Author Response · Authors · 2024-11-21
> > **Response to Reviewer XeZc's comments (2)**
> >
> > > W2. Clarity: The presented method is provided with enriched information about biomedical context. The authors missed to explain whether baseline method did get a fair chance to operate on this enriched data base as well. Could have performance gains just have arisen because the presented method is provided with more information (biomedical context)?
> >
> > Thank you for your constructive feedback regarding the use of enriched biomedical context in our experiments. We acknowledge that our initial explanation was not clear enough in addressing the comparison with baseline methods. Below, we provide a more detailed clarification:
> >
> > 1. We first want to clarify that state-of-the-art models like DrugMetric, DGCAN, and DeepDL are not designed to incorporate biomedical context into their decision processes. Their architectures are fundamentally constrained to work with structural data alone, such as by utilizing Graph Neural Networks. Adapting these methods to handle multi-modal embeddings would require extensive modifications, effectively resulting in the creation of new methods rather than fair comparisons.
> > 2. The biomedical embeddings utilized in our study are only accessible for a subset of approved drugs (around 2,000 compounds in our dataset). Non-drug compounds, of which there are approximately 100,000, do not have associated biomedical context due to the absence of known biological targets or linked genes. This restriction is a key reason why baseline models, which operate on standard structural data alone, cannot leverage the same biomedical information.
> > 3. To address potential concerns of fairness, we did evaluate limited baseline models with our biomedical context-aligned embeddings, as presented in our ablation study (Table 4). The baseline models, such as SVM and XGBoost, demonstrated performance improvements when trained with the enriched embeddings; for instance, SVM’s MCC improved by up to 20 percentage points. However, these gains were still significantly less pronounced than those achieved by our proposed method of 35 percent points, indicating that our alignment and optimization strategies play a crucial role.
> >
> > We hope this clarifies the distinct contributions of our approach and the reasoning behind our experimental design.
> >
> >
> > > W3. Experiments: The ablation study is informative. However it would have been interesting to train a MLP (i.e. a universal function approximator) on top of the embeddings from the contrastive learning space to estimate the effectiveness of the EM-based algorithm. What is the intuition why a MLP should not be capable of learning a proper boundary? Is it that not only the boundary but also the encoder is iteratively updated in the EM algorithm? If so, can it be shown that this is the critical point?
> >
> > We sincerely thank the reviewer for highlighting the potential benefits of assessing an MLP on top of contrastive embeddings and for raising a critical point about our iterative EM-like approach. We conducted additional experiments using an MLP trained on the same embeddings derived from our contrastive learning space. These results have been added to the Table 5 as shown in Table R2-1.
> >
> > **Table R2-1** Drug-like compound identification with different classifiers on knowledge-aligned space. Best performance in bold.
> > |      |       F1       |      AUROC     |      Average Precision      |
> > |------|:--------------:|:--------------:|:--------------:|
> > | Ours | **0.826 (0.0486)** | **0.973 (0.0075)** | 0.877 (0.0419) |
> > |  MLP | 0.771 (0.0195) | 0.971 (0.0013) | **0.913 (0.0070)** |
> >
> > While the MLP performed reasonably well, it fell short compared to our EM-based approach in F1 and AUROC. This result underscores a fundamental distinction between our method and static classification: the iterative refinement enabled by our EM-like optimization is key. Unlike an MLP, which operates over fixed embeddings, our method involves iterative co-evolution of both the decision boundary and the feature representation. Each EM iteration refines the embedding space, tightening the decision boundary and improving the separation between drug-like and non-drug-like compounds.
> >
> > We realized that the current manuscript lacked sufficient emphasis on this co-evolution. To clarify, we have included a more detailed explanation of this iterative refinement in the revised manuscript, demonstrating how it captures complex, non-linear relationships in the data that a static MLP cannot. We also added an additional figure in the Appendix that shows how the ratio of out-boundary compounds increases and converges across iterations. This empirical evidence solidifies the critical role of the iterative refinement in enhancing predictive accuracy.

---

> > > ### Author Response · Authors · 2024-11-21
> > > **Response to Reviewer XeZc's comments (3)**
> > >
> > > **Questions**
> > > > Q1. Could have performance gains just have arisen because the presented method is provided with more information (biomedical context)?
> > >
> > > We appreciate the reviewer’s concern about the role of biomedical context in driving the performance of our method. We would like to clarify that our performance gains are not merely due to the incorporation of additional biomedical information, but rather arise from the careful integration of this context with our novel EM-like optimization framework.
> > >
> > > Our approach provides two key contributions:
> > > 1. **Biomedical Context Alignment:** We leverage embeddings derived from biomedical literature to inform our initial understanding of drug-like compounds. These embeddings are essential for capturing domain-specific nuances, which purely structural features cannot capture.
> > > 2. **EM-like Optimization:** The iterative EM-like process is pivotal in fine-tuning the drug-likeness boundary. Our ablation studies (Table 4) explicitly show that removing the EM-like iterative refinement results in a notable drop in accuracy, even when biomedical embeddings are retained. This indicates that the EM-like optimization is a significant contributor to our results.
> > >
> > > These results confirm that the EM-like iterative boundary refinement plays a critical role in the observed performance, demonstrating that our method’s success stems from the synergy between biomedical context and iterative optimization, not merely the availability of additional data.
> > >
> > >
> > > > Q2. What is the intuition why a MLP should not be capable of learning a proper boundary? Is it that not only the boundary but also the encoder is iteratively updated in the EM algorithm? If so, can this be somehow shown?
> > >
> > > We appreciate the reviewer's thoughtful question about why an MLP might struggle to achieve the same boundary effectiveness as our EM-like approach. Indeed, the core advantage of our method lies in its iterative updates to both the decision boundary and the encoder. Unlike an MLP, which relies on fixed embeddings, our algorithm dynamically adjusts the feature space and boundary across multiple iterations.
> > >
> > > We realized our initial explanation lacked sufficient clarity regarding this dynamic co-evolution. In the revised manuscript, we have expanded the Discussion section to outline how each iteration works:
> > >
> > > 1. An initial, coarse boundary is set using the contrastive embeddings.
> > > 2. The encoder refines these embeddings based on feedback from the initial boundary, adjusting the representation.
> > > 3. A new boundary is established using these refined embeddings.
> > > 4. This process repeats, allowing the model to fine-tune both the decision criteria and the feature space.
> > >
> > > To further clarify this process, we've added Appendix Figure as replied to **W2**, that shows how the ratio of out-boundary compounds increases and converges over time with each EM iteration. This progressive refinement demonstrates the limitations of a static MLP approach, reinforcing the necessity of our iterative EM-like strategy for accurate boundary learning.
> > >
> > >
> > > > Q3. What if the structural and biomedical representations are naively concatenated and fed into an MLP? Shouldn't this approach be able to get similar performance values compared to the presented one without the need of the two-stage training procedure?
> > >
> > > We appreciate the reviewer’s suggestion to evaluate a simpler baseline approach, such as naively concatenating structural and biomedical embeddings and training an MLP. However, we would like to clarify why this approach is not applicable given the constraints of our dataset and why a dedicated multi-modal alignment is essential.
> > >
> > > In our study, biomedical context embeddings are only available for a small subset of known drugs (approximately 2,000 compounds), while non-drugs lack any biomedical representation. This creates a significant challenge for a naive concatenation approach, as it would result in an inconsistent feature space where the MLP would have to operate with missing data for the majority of samples. The lack of biomedical context for non-drugs would leave the MLP or any other classifiers ill-equipped to distinguish drug-like properties based solely on structural data.
> > >
> > > Thus, the proposed two-stage training procedure to first generate a knowledge-structure aligned space is not a matter of added complexity but a necessary strategy to integrate disparate data sources in a meaningful and effective way, thereby allowing non-drugs to be mapped into the unified space and enabling the accurate prediction of drug-likeness.

---

> ### Comment · Reviewer_XeZc · 2024-11-28
> **Reviwer answer to the authors**
>
> Dear authors,
>
> thank you for your answers and the immense effort you put into considering my thoughts.
>
> **Multi-Modality / Contrastive Learning**
> I still think the related work section could be improved by a more extensive and careful description of how related approaches have been already used in drug discovery. Here, I am pretty much in line with reviewer T5wN. Quite some approaches have been used in drug-discovery related ML already and mechanically I don't see great novelty in the way CL is used in this work (because simply demonstrating CL approaches can be applied to different data source, here drug-likeliness data, is just of low relevance).
>
> **EM-approach to define decision boundary**
> - The error bars in Table R2-1 overlap, suggesting that the EM approach and the MLP perform similarly in terms of predictive performance.
> - The authors' approach (optimized encoder plus EM head) aims to map inputs into an embedding space where molecules are separable by an epsilon-ball based on their drug-likeness. Conversely, MLPs aim to map inputs into an embedding space that is separable by a hyperplane. I don't see a clear advantage of one method over the other or in other words I don't see why one should be more discriminative than the other. If the bottleneck is the fixed embeddings for the MLP approach, isn't this an artificially introduced obstacle? The backbone encoder is fine-tuned for the EM approach but remains fixed for the MLP approach, potentially biasing the results. If the bottleneck is the fixed embeddings for the MLP approach, isn't this an artificially introduced obstacle? The backbone encoder is fine-tuned for the EM approach but remains fixed for the MLP approach, potentially biasing the results.
>
> **One-Class Classification**
> > "We propose a new perspective on the problem of drug-likeness prediction as constructing a compact and adaptive one-class boundary."
>
> This is true, interesting and relevant. However, aren't one-class classification and binary classification essentially equivalent problems, differing only in their formulation?
>
> I appreciate the one-class classification perspective and recognize that the authors follow best practices using CL enriched training. However, I still feel the proposed method lacks substantially novel components compared to existing approaches in the field.
>
> This remains a borderline paper, and I have decided to adhere to my initial rating.

---

> ### Author Response · Authors · 2024-11-29
> **Response to Reviewer XeZc's comments 2**
>
> Dear Reviewer,
>
> Thank you for your thorough review and insightful comments. We deeply appreciate the time and effort you have dedicated to understanding our responses. We have incorporated your feedback and present slightly further clarifications below, which we hope will provide a tipping point for reconsideration.
>
> ### 1. Multi-Modality and Contrastive Learning
>
> We agree that contrastive learning (CL) is not a novel concept in drug discovery. Our novelty lies in extending this with a **modified CLIP loss function, integrating prior knowledge through softened CLIP**, resulting in 10% point improvement over standard CL method (Table 4, page 9).
>
> Additionally, we introduced **geodesic mix-up** to blend chemical structures and biomedical knowledge. Unlike previous approaches that rely on aligning close domains including drug-protein and drug-atomic knowledge, our technique interpolates between distant representation of molecular and clinical level modalities. The geodesic mix-up approach validated as crucial through ablation studies (Table 17, page 27). To our knowledge, this approach is unprecedented in drug discovery.
>
> ### 2. EM Approach to Defining Decision Boundaries
> Firstly, we clarify that the backbone encoder is not fine-tuned for our model. Our model trains only one encoder that generates an embedding space, followed by drug hypersphere identification (center point and radius) on the embedding space, as verified in our repository codes (src/models.py). All models, including ours, receive the identical fixed embedding vectors as input. **The reviewer’s assumption on the bias in results is not valid.**
>
> The fundamental distinction lies in the decision boundaries: MLPs use hyperplanes, while our approach generates a **compact one-class boundary** solely based on drug-like compounds. This compactness is crucial for drug-likeness prediction in scenarios where non-drug compounds vastly outnumber drug compounds, a challenge typically seen in real-world datasets.
>
> To support this claim, we compared MLP and EM models across test sets with varying drug-compound ratios, following the experiment in Figure 6. As the compound proportion increases, the EM approach maintains robustness compared to binary classifiers, aligning with the results in Figure 6. We interpret this result as the compact boundary’s ability to exclude non-drug-like outliers more effectively.
>
> **Table R2-2.** Change of F1 score with the decrease in drug-compound ratio of the test set with different classifiers on knowledge-aligned space.
> | Aligned space | 1:10 | 1:50 | 1:100 |
> |---|---|---|---|
> |+ MLP | 0.771 (0.0195) | 0.410 (0.0273) | 0.255 (0.0164) |
> |+ Ours | 0.826 (0.0486) | 0.691 (0.1019) | 0.558 (0.1146) |
> |+ Ours $_\text{Mult}$ | 0.846 (0.0165) | 0.731 (0.0303) | 0.600 (0.0402) |
>
> Furthermore, Table R2-3 demonstrates our extension Mult, designed to mitigate sensitivity to initial conditions, addressing error bar overlaps by enhancing stability in the EM framework.
>
> **Table R2-3.** Drug-like compound identification with different classifiers on knowledge-aligned space.
> |Aligned space| F1 (↑)          | IDR (↑)         | ICR (↓)         | AUROC (↑)       | Avg. Precsion (↑) |
> |--------------|-----------------|-----------------|-----------------|-----------------|--------------------|
> | + MLP | 0.771 (0.0195) | **0.910 (0.0092)** | 0.046 (0.0053) | 0.971 (0.0013) | **0.913 (0.0070)** |
> | + Ours| 0.826 (0.0486) | 0.781 (0.0326)	| 0.012 (0.0086)	| 0.973 (0.0075)	|0.877 (0.0419)|
> | + Ours $_\text{Mult}$| **0.846 (0.0165)** | 0.799 (0.0184) | **0.009 (0.0031)** | **0.978 (0.0029)** | 0.908 (0.0096)    |
>
>
> ### 3. One-Class vs. Binary Classification
> > Aren't one-class classification and binary classification essentially equivalent problems, differing only in their formulation?
>
> Thank you for raising this critical point. While binary and one-class classification share theoretical similarities, their practical implications diverge significantly, particularly in drug-likeness prediction. Hyperplane-based classifiers like SVMs or MLPs create a linear separation, which can be less effective when the positive class (drugs) is greatly outnumbered by negative class (non-drugs), potentially leading to high false-positive rates. Conversely, one-class classification’s objective is in constructing **compact boundary around positives**, inherently focusing on minimizing false positives by excluding non-drug-like compounds.
>
> Given the vast disparity between the number of drug compounds (~20k) and the theoretical non-drug space (10^60), this compact boundary is critical for practical utility. Our results demonstrate the reduction in false positives, particularly toxic compounds (Table 3) and AI-generated molecules (Table 6), underscoring the practical necessity of this approach in high-imbalance scenarios.
>
> We hope these clarifications highlight the novelty and practical relevance of our work. Thank you once again for your continued valuable feedbacks.

---

### Official Review · Reviewer_jyv4 · 2024-11-11

**Soundness:** 3
**Presentation:** 2
**Contribution:** 2
**Rating:** 5
**Confidence:** 4

**Summary:**

The paper's central contribution lies in its attempt to merge biomedical knowledge with structural information for drug-likeness prediction. However, the approach reveals several concerning technical limitations that significantly impact its potential impact and practical utility.
The authors' adaptation of the CLIP loss for knowledge alignment (Equations 2 and 3) appears superficially innovative, but closer examination reveals concerning oversimplifications. The assumption that ATC classification similarities can be directly translated into embedding space relationships is problematic and insufficiently justified. The authors fail to address the inherent hierarchical nature of ATC classifications and how this hierarchy affects similarity computations. This oversimplification potentially introduces systematic biases in the learned representations. The EM-like optimization process, while mathematically presented with apparent rigor, suffers from serious theoretical gaps. The proof of Theorem 1 relies heavily on assumptions that are neither fully justified nor empirically validated. Most concerningly, the authors claim convergence properties without adequately addressing the potential for local optima or the impact of initialization conditions.

**Strengths:**

1.The paper introduces a novel technical framework integrating biomedical knowledge with molecular structure representation. The use of multi-modal alignment through softened CLIP loss represents an interesting attempt to bridge the gap between chemical structure and biological function.

2.The theoretical foundation, particularly the EM-like boundary optimization process, is mathematically formulated. The authors provide formal proofs for boundary convergence (Theorem 1) and establish conditions for reducing in-boundary non-drugs.

3.The experimental framework appears comprehensive, employing both scaffold-based and time-based splits to evaluate generalization capability. The integration of zero-shot toxic compound filtering demonstrates potential practical utility.

**Weaknesses:**

The paper introduces a novel technical framework integrating biomedical knowledge with molecular structure representation. The use of multi-modal alignment through softened CLIP loss represents an interesting attempt to bridge the gap between chemical structure and biological function. The theoretical foundation, particularly the EM-like boundary optimization process, is mathematically formulated. The authors provide formal proofs for boundary convergence (Theorem 1) and establish conditions for reducing in-boundary non-drugs. The experimental framework appears comprehensive, employing both scaffold-based and time-based splits to evaluate generalization capability. The integration of zero-shot toxic compound filtering demonstrates potential practical utility.

The softened CLIP loss formulation (Equations 2,3) oversimplifies the complex hierarchical nature of ATC classifications. The geodesic mixup strategy lacks proper ablation studies to justify its effectiveness. The boundary optimization process shows concerning sensitivity to initialization conditions, which is inadequately addressed.

The reported "five orders of magnitude improvement" raises serious concerns about potential data leakage or experimental design flaws
Baseline comparisons appear biased, with competing methods using suboptimal default parameters. The scaffold distribution (1.97 molecules per scaffold) indicates extreme sparsity, yet its impact on model generalization is not properly analyzed. Statistical significance testing is notably absent from all experimental results.

Neural architecture choices (2-layer MLP, 512-dimensional hidden layers) lack proper justification. Training protocol shows concerning arbitrariness in hyperparameter selection. The convergence criterion based on in-boundary compound ratio is potentially unstable. Computational complexity analysis is entirely missing.

Zero-shot toxicity prediction results lack proper error analysis and statistical validation. The time-based split evaluation inadequately addresses temporal evolution in drug discovery practices. External validation sets are limited and potentially biased. The relationship between model predictions and actual experimental outcomes is not explored

**Questions:**

1.What theoretical guarantees exist for avoiding local optima in the EM-like optimization process?
2.How does the method handle conflicting or inconsistent information between structural and knowledge-based representations?
3.How could the method be adapted for specific therapeutic area focusing?

---

> ### Author Response · Authors · 2024-11-21
> **Response to Reviewer jyv4's comments (1)**
>
> We sincerely thank the reviewers for their insightful comments and constructive feedback, which enabled us to improve our work further. In the following comments, we respond to each comment for your initial assessment. The revised manuscript will be available shortly, incorporating all the updates.
>
> **Weaknesses**
> > **W1:** The softened CLIP loss formulation (Equations 2,3) oversimplifies the complex hierarchical nature of ATC classifications.
>
> We appreciate the reviewer's insightful observation regarding the complexity of the ATC classification hierarchy. The Anatomical Therapeutic Chemical (ATC) classification is indeed a multi-level ontology, categorizing drugs based on their target system, therapeutic application, and chemical properties. Integrating such a complex hierarchy into our framework is challenging, and we acknowledge the potential limitations of our current softened CLIP loss formulation.
>
> We here provide more explanations on the current choice to simplify the hierarchical structure, which was driven by practical considerations:
> 1. **Practical Feasibility:** Directly integrating the entire ATC tree structure into a neural alignment framework requires complex adjustments to the architecture that may significantly increase computational overhead and complicate the model training process without guaranteeing improved alignment. Our softened CLIP formulation aims to balance between capturing the semantic similarities among drugs while maintaining computational tractability.
> 2. **Capturing Hierarchical Information:** Although our formulation is simplified, it retains the essence of the hierarchical ATC relationships by mapping drugs into a shared embedding space based on their contextual similarities. This shared space still respects the semantic contexts encoded in the ATC classifications, allowing our model to differentiate drugs effectively based on both therapeutic and pharmacological properties.
>
> The usefulness of this abstract yet efficient form of ATC hierarchy is demonstrated in our ablation table (Table R1-1), where our model without ATC similarity shows suboptimal performances compared to the current ATC similarity-integrated version.
>
> **Table R1-2.** Performances of our model compared to model with softened CLIP replaced to original CLIP.
> | Alignment method   | F1 (↑)          | IDR (↑)         | ICR (↓)         | AUROC (↑)       | Avg. Precsion (↑) |
> |--------------|-----------------|-----------------|-----------------|-----------------|--------------------|
> | Ours w/ Softened CLIP | 0.826 (0.0486) | 0.781 (0.0326)	| 0.012 (0.0086)	| 0.973 (0.0075)	|0.877 (0.0419)|
> |    Ours w/ Original CLIP   | 0.727 (0.0365) | 0.670 (0.0605) | 0.018 (0.0066) | 0.801 (0.0506) | 0.755 (0.0481) |
> | Paired t-test p-value | 6.71e-05 | 1.88e-05 | 0.0764 | 3.2e-05 | 0.00526 |
> |
>
> Nonetheless, we recognize the value of a deeper hierarchical integration and consider this a promising area for future work. Exploring more sophisticated neural architectures that can explicitly model tree structured prior knowledge, such as Graph Neural Networks (GNNs), could be a potential extension to better capture the intricacies of ATC. We hope this explanation clarifies our methodological choices and why the current approach, while simplified, remains effective for our target task. We have added the following paragraph in Appendix C.2 section to further clarify the points discussed through this comment:
>
> ```latex
> \subsubsection{C2. Semantic drug similarity calculation with ATC codes}
> A direct modeling of such complex hierarchical structure as prior knowledge in model training is challenging. In order to retain the essence of the hierarchical ATC relationships without complex adjustments to the architecture that may significantly increase computational overhead and complicate the model training process, we utilized the concept of semantic similarities between terms within the hierarchy and integrated them as prior knowledge to our softened CLIP loss.
> ```

---

> > ### Author Response · Authors · 2024-11-21
> > **Response to Reviewer jyv4's comments (2)**
> >
> > > **W2**: The geodesic mixup strategy lacks proper ablation studies to justify its effectiveness.
> >
> > We appreciate the reviewer’s observation regarding the need for a detailed ablation study of the geodesic mixup strategy (composed of S-mix, K-mix, and KS-mix). In response, we have conducted additional ablation experiments (Table R1-2) and included them in the Appendix. Here is a brief summary of the key findings:
> > 1. **Individual Impact of Mixup Components:** We evaluated the performance of the model by selectively removing each component—S-mix, K-mix, and KS-mix—from the final setup. The results indicate that each component contributes uniquely to the model’s performance:
> >     - Removing both S-mix and K-mix resulted in a drop of 36 percent points in F1 score, indicating their contribution to aligning embeddings across diverse drug classes and scaffolds in each of structural and knowledge spaces.
> >     - Without KS-mix, the model showed a reduction of 22 percent point in F1 score, underscoring the importance of a balanced contribution from both structural and semantic features.
> > 2. **Combined Effect:** Our results show that the combination of all three strategies yields the best performance, with a synergistic effect that improves both classification accuracy and stability.
> >
> > We hope this additional evidence addresses the reviewer’s concerns and demonstrates the necessity of each mixup strategy in achieving the optimal alignment of embeddings.
> >
> >
> > **Table R1-2.** Ablation study results on the components of geodesic mixup (S,K-mix, KS-mix).
> > |        |       F1       |       IDR      |       ICR      |      AUROC     |      Avg. Precision      |
> > |--------|:--------------:|:--------------:|:--------------:|:--------------:|:--------------:|
> > |  Ours w/o S,K-mix  | 0.466 (0.1705) | 0.745 (0.1058) | 0.270 (0.3446) | 0.818 (0.1825) | 0.420 (0.1995) |
> > | Ours w/o KS-mix | 0.604 (0.2238) | **0.858 (0.0734)** | 0.241 (0.3782) | 0.849 (0.2091) | 0.576 (0.2546) |
> > |  Ours  | **0.826 (0.0486)** | 0.781 (0.0326) | **0.012 (0.0086)** | **0.973 (0.0075)** | **0.877 (0.0419)** |
> >
> > > **W3**:The boundary optimization process shows concerning sensitivity to initialization conditions, which is inadequately addressed.
> >
> > We appreciate your valuable observation regarding the sensitivity of our EM-like boundary optimization process to initialization conditions. This important point was not discussed enough in the manuscript, which we aim to revise in the updated version. This is indeed a recognized characteristic of EM-based methods, and we have taken additional steps to address this issue in our framework in the review process.
> > 1. **Knowledge-Aligned Embedding to Mitigate Initialization Sensitivity:** A key aspect of our approach lies in the synergy between the knowledge-aligned embedding space and the EM-like boundary optimization. The embedding space, generated through our knowledge-alignment process, creates an informative initial representation that effectively separates drug-like from non-drug-like compounds. This reduces the impact of the early optimization steps, as it places many non-drugs initially out-of-boundary, stabilizing the initial conditions and reducing sensitivity.
> > 2. **Multi-Initialization Strategy:** Although our knowledge-aligned space offers improved initialization point, we further attempted to improve the initialization sensitivity. Inspired by successful strategies in EM-based models, such as the MEME algorithm, **we have newly implemented a multi-initialization approach in our framework**. Specifically, we initialize our boundary optimization process multiple times from different random seeds and retain the best-performing model based on a validation criterion (selected on validation set performance without seeing test data). This approach has proven effective in enhancing performance by mitigating the risk of poor local optima, as demonstrated by the results in Table R1-3.
> >
> > **Table R1-3. Prediction performance of our model with multi initialization strategy with time-split setting.** Overall performance showed improvement in all metrics, with Average Precision demonstrating significant improvement with one-sided paired t-test p-value < 0.05. (Avg.: Average)
> > |              | F1           | IDR         | ICR        | AUROC        | Avg. Precision |
> > |--------------|-----------------|-----------------|-----------------|-----------------|--------------------|
> > | BOUNDR.E     | 0.826 (0.0486) | 0.781 (0.0326)	| 0.012 (0.0086)	| 0.973 (0.0075)	|0.877 (0.0419)|
> > | BOUNDR.E_MULT | **0.846 (0.0165)** | **0.799 (0.0184)** | **0.009 (0.0031)** | **0.978 (0.0029)** | **0.908 (0.0096)**    |
> > | Paired t-test p-value     | 0.124 | 0.085 | 0.246 | 0.069 | **0.034** |
> > |
> >
> > We have elaborated on these points in the newly added “MULTIPLE EM APPROACH FOR AVOIDANCE OF LOCAL OPTIMA” section in the Appendix and the Discussion section to highlight how our framework's unique components contribute to mitigating initialization sensitivity.

---

> > > ### Author Response · Authors · 2024-11-21
> > > **Response to Reviewer jyv4's comments (3)**
> > >
> > > > **W4:** The reported "five orders of magnitude improvement" raises serious concerns about potential data leakage or experimental design flaws.
> > >
> > > We apologize for the ambiguity in the abstract concerning the "five orders of magnitude improvement" in the Abstract, which is reported with the ICR/IDR metric. The sensitivity of this metric, which ranges from a minimum of 0.5 to a maximum of 93, indeed makes it challenging to interpret meaningful improvements in practical terms. We have revised the metric presentation to use the F1 score, a more widely accepted measure of classification performance, instead of ICR/IDR metric.
> > >
> > > We have revised the statement in the Abstract to:
> > > ```
> > > Our model results indicate 10% increase in F1 score over the previous state-of-the-art.
> > > ```
> > > This metric more accurately reflects our model's ability to correctly identify drug-like compounds, providing a clearer and more practical measure of improvement. We hope this change alleviates any concerns regarding metric sensitivity and better represents the significance of our work.
> > >
> > >
> > > > **W5:** Baseline comparisons appear biased, with competing methods using suboptimal default parameters.
> > >
> > > Thank you for pointing out the unclearness in the description of our experiments. To ensure fairness in our comparisons with baseline models, we took deliberate and careful steps to eliminate any potential bias in the evaluation. Here is a detailed breakdown of the measures we implemented:
> > > 1. For comparisons with plain MLP-based architectures, we ensured that both our model and the baselines had identical numbers of layers and parameters. Specifically, each baseline was adjusted to match the total parameter count and architectural capacity of our model, ensuring comparable expressibility.
> > > 2. For machine learning-based baseline models, we conducted limited search across a range of hyperparameters, including number of estimators. This search was performed using cross-validation to ensure that the most effective configurations were applied consistently across all models. The specific ranges and results of this hyperparameter search are now documented in the Appendix to make our tuning process fully transparent.
> > >
> > > We included such details in the Appendix of our paper. We believe these comprehensive efforts eliminate any potential bias and ensure the validity of our results.
> > >
> > > > **W7**: Statistical significance testing is notably absent from all experimental results.
> > >
> > > We appreciate your critical point regarding the need for statistical significance testing to strengthen our experimental results. In response, we have added a comprehensive statistical analysis, which will be detailed in the Appendix. For the sake of comment limitations, we do not provide the whole table here, and hope your understanding untill the revised manuscript is uploaded. Below are the key elements of this added analysis:
> > >
> > > 1. Since we conducted a 10-fold cross-validation for each experiment, for each fold, we calculated the p-value comparing our model to the second-best baseline using a one-sided paired t-test. This ensures that observed differences between SOTA performance are statistically significantly higher rather than random fluctuations due to the dataset.
> > > 2. To clearly convey the statistical significance, we have highlighted in the main tables the best-performing results, along with any models that showed statistically comparable performance (p < 0.05). This visual representation emphasizes where performance differences are meaningful and where they are not.
> > >
> > > These steps provide a rigorous statistical foundation for our reported improvements, addressing your concerns regarding the robustness of our findings.
> > >
> > > > W8: Neural architecture choices (2-layer MLP, 512-dimensional hidden layers) lack proper justification. & W9: Training protocol shows concerning arbitrariness in hyperparameter selection.
> > >
> > > We appreciate your feedback regarding the lacking justification on the choice of neural architecture and hyperparameters.
> > > Our choice of a 2-layer MLP with 512-dimensional hidden layers was guided by empirical experiments conducted during the preliminary phase. The chosen architecture aligns with prior work in drug-likeness prediction and molecular property prediction, where 2-3 layers with 256-1024 dimensions are commonly used due to their balance between expressiveness and computational efficiency. This configuration is particularly effective when the goal is to capture complex yet subtle patterns in molecular data without overfitting. While we prioritized robust empirical evaluation over hyperparameter searches, the selected configuration was validated through a search on a validation set. An updated hyperparameter search space is provided in the Appendix.
> > >
> > > We hope this clarifies the rationale behind our architecture, demonstrating that it has been justified through empirical results.

---

> ### Author Response · Authors · 2024-11-21
> **Response to Reviewer jyv4's comments (4)**
>
> > **W10:** The convergence criterion based on in-boundary compound ratio is potentially unstable.
>
> Thank you for pointing out the potential instability associated with our convergence criterion. We fully recognize that a reliable convergence metric is critical for deep learning models to ensure consistent and meaningful optimization.
>
> In our model, we initially considered using a traditional loss-based convergence criterion, which would directly correspond to the model’s objective of distance minimization. However, due to the nature of our distance metric, convergence using a loss-based criterion proved challenging; it occasionally led to expansions or contractions of the latent space that risked numerical instability (e.g., overflow/underflow issues). Consequently, we adopted the in-boundary compound ratio as the convergence criterion for the following reasons:
>
> 1. **Theoretical Alignment:** In the Appendix A.2 of initial submission, we provided a formal proof demonstrating that optimizing the distance metric inherently results in a decrease in the in-boundary compound ratio. This proof establishes a theoretical link between loss minimization and our chosen convergence criterion, indicating that both approaches are consistent with the model’s objectives.
>
> 2. **Empirical Stability:** We conducted experiments to empirically compare the performance of our model when using the loss-based criterion versus the in-boundary compound ratio (Table R1-4). The results show no significant difference in final model accuracy, with a p-value of 0.737 which is greater than 0.05 based on a two-sided paired t-test, demonstrating that the two methods converge to similar solutions. Furthermore, the average number of training epochs needed for convergence was slightly reduced when using the in-boundary compound ratio, indicating faster stabilization.
>
>     **Table R1-4.** Performances of our model with two different convergence metrics.
>     | Convergence metric| F1 (↑)          | IDR (↑)         | ICR (↓)         | AUROC (↑)       | Avg. Precision (↑) | Train Epochs (↓)|
>     |--------------|-----------------|-----------------|-----------------|---|--|---|
>     | ICR      | 0.826 (0.0486) | 0.7807 (0.0326)	| 0.012 (0.0086)	| 0.973 (0.0075)	|0.877 (0.0419)| **47.7 (4.20)**|
>     | $\mathcal{L}_\text{boundary}$     | 0.833 (0.0463) | 0.806 (0.0236) | 0.014 (0.0098) | 0.973 (0.0071) | 0.885 (0.0463)    | 202.7 (99.20)|
>     |
>     | Paired t-test p-value     | 0.737 | 0.055 | 0.615 | 0.956 | 0.723 |
>     |
>
> To enhance clarity, we have included these empirical comparisons, as well as a detailed theoretical discussion, in the revised Appendix. We hope this additional information adequately addresses your concern regarding the stability and suitability of our convergence criterion.
>
> > **W11**: Computational complexity analysis is entirely missing.
>
> We sincerely appreciate the reviewer’s observation regarding the need for a detailed computational complexity analysis, as it is a fundamental aspect of validating our model’s efficiency and scalability.
>
> 1. **E-step (Boundary Update):** The E-step in our model relies on computing the Euclidean distance from the center, with a time complexity linear in both the number of samples ($N$) and the dimensionality ($D$) of the data, resulting in $O(N \times D)$. This ensures that the boundary update is scalable even for high-dimensional datasets.
> 2. **M-step (Neural Network Optimization):** In the M-step, the primary computational effort involves neural network optimization. If we denote $H$ as the number of layers, $F_h$ as the number of operations in layer $h$, and $N$ as the dataset size, then the complexity for a forward pass is $O(N \cdot \sum_{h=1}^H F_h)$. Given that the backward pass is approximately twice as computationally expensive, the overall complexity for each EM iteration is $O(N \times D) + O(N \cdot \sum_{h=1}^H F_h)$.
>
> These complexities illustrate the model's linear behavior with respect to data size and dimensionality, making it efficient for large-scale drug discovery tasks. To validate these claims empirically, we trained our model with approximately 200 drugs and 2,000 non-drug compounds around 100 epochs using single NVIDIA RTX 3090 GPU, and the total training time was consistently under 5 minutes, demonstrating the alignment between theoretical analysis and practical performance.
>
> We have included this computational complexity breakdown in the revised Appendix to highlight the efficiency benefits of our approach.

---

> > ### Author Response · Authors · 2024-11-23
> > **Response to Reviewer jyv4's comments (5)**
> >
> > > W6: The scaffold distribution (1.97 molecules per scaffold) indicates extreme sparsity, yet its impact on model generalization is not properly analyzed. (We noticed the response to W6 was left out in the last comments, and added them here.)
> >
> > We deeply appreciate the reviewer’s insight regarding the extreme sparsity of the scaffold distribution and its potential impact on model generalization. We fully agree that understanding how our model generalizes to unseen scaffolds is crucial given the limited average scaffold size of the approved drug space.
> >
> > 1. **Scaffold-Split Justification:** To address this concern, we have implemented a scaffold-split evaluation (Appendix E.1), which is specifically designed to challenge the model's ability to generalize when encountering new scaffolds that were not seen during training. In drug discovery, scaffold diversity is a key concern, as new drugs often emerge from novel scaffolds that were previously untested. The scaffold-split evaluation aligns closely with these real-world scenarios, making it a more rigorous and realistic test of generalization than a random split, where similar scaffolds are likely to appear in both training and test sets.
> > 2. **Experimental Findings:** The performance significantly decreases when using scaffold-split compared to randomly splitted setting, indicating that the model's ability to handle unseen scaffolds is inherently more challenging. This underscores the necessity of scaffold-split as a more appropriate evaluation scheme for understanding the impact of scaffold sparsity and further evaluate the models' generalizability. These findings, along with additional statistical analyses, are now detailed in the revised manuscript’s Experiments section and in the Appendix (Table R1-5).
> >
> > **Table R1-5**. Prediction performances of BounDr.E when applied on different split schemes. Our model displays significant decrease in prediction performances when applied with scaffold split, a splitting scheme to evalutate the models' generalizability in the sparse distribution of approved drugs' scaffolds.
> > |                | F1                 | IDR                | ICR                | AUROC                | Avg. Precision     |
> > |----------------|--------------------|--------------------|--------------------|----------------------|--------------------|
> > | Scaffold-based split | 0.655 (0.0209)     | **0.796 (0.0258)** | 0.063 (0.0079)     | **0.938 (0.0049)**   | 0.590 (0.0369)     |
> > | Random split   | **0.689 (0.0142)** | 0.742 (0.0291)     | **0.041 (0.0060)** | **0.942   (0.0037)** | **0.663 (0.0379)** |
> > | paired t-test p-value          | 4.4E-4            | 4.6E-4            | 1.6E-04           | 0.082               | 0.008             |
> > |
> >
> > We have incorporated a detailed discussion in Section 4.1 of the Experiments section to highlight the rationale behind our scaffold-split evaluation. Moreover, we have expanded the Appendix to include a comparative analysis between scaffold and random splits, demonstrating how scaffold sparsity affects model performance.
> >
> >     4. Experiments
> >     4.1 Setup
> >     This evaluation scheme is applied to measure the models’ generalizablilty when an unseen scaffold compound is input, where approved drugs exist extremely sparse in the scaffold space.
> >
> >
> > We believe these changes provide a clearer understanding of how scaffold sparsity impacts our evaluation and substantiate the validity of our chosen evaluation scheme.
> >
> > > W13: The time-based split evaluation inadequately addresses temporal evolution in drug discovery practices.
> >
> > We are grateful to the reviewer for highlighting the need to better justify the time-based split evaluation. We acknowledge that our initial manuscript may not have clearly demonstrated how this split captures the temporal evolution inherent in drug discovery.
> >
> > 1. **Temporal Trends in Drug Development:** The properties of approved drugs have evolved significantly over the past decades, particularly with the rise of new therapeutic modalities and technologies. For example, kinase-targeted drugs and biologics became prominent in the 2000s, leading to an increase in molecular complexity, larger molecular weights, and drugs that often fall outside traditional Rule-of-5 constraints [1]. Additionally, the advancement of drug delivery systems has allowed for a higher range of LogP values (lower solubility) among approved drugs [2].
> > 2. **Empirical Validation of Time-Based Split:** To validate that our time-based split reflects these temporal trends, we have conducted a detailed analysis of drug properties over the periods represented in our dataset. Specifically, we tracked changes in key chemical characteristics (e.g., molecular weight, LogP, polar surface area) across different temporal splits, observing clear shifts that align with known trends in drug development. These findings are now detailed in the revised manuscript, supported by new tables and statistical analyses (see Table R1-6).

---

> > > ### Author Response · Authors · 2024-11-23
> > > **Response to Reviewer jyv4's comments (6)**
> > >
> > > **Table R1-6** Molecular properties of drugs in the train set (approved before 2011) and test set (approved since 2011). Drugs in the test set show significant difference from the train set drugs, according to the temporal evolution of approved drugs.
> > > ||Ro5 |   MW |     LogP |        PSA |
> > > |--------------------:|---------:|-----------:|---------:|-----------:|
> > > | Train (Before 2011) | 3.652739 | 398.120084 | 2.142421 | 100.041105 |
> > > |   Test (Since 2011) | 3.379032 | 540.368339 | 2.937724 | 137.452177 |
> > > |             p-value | 0.000396 |   0.000583 | 0.024349 |   0.033635 |
> > >
> > > We have expanded the Experiments section to include a more thorough explanation of how the time-based split mirrors the real-world evolution of drug properties as below:
> > >
> > >     4. Experiments
> > >     4.1 Setup
> > >     In the time-based split, drugs are partitioned based on their approval year (e.g., drugs approved post-2011 are in the test set), to reflect the temporal evolution of approved drug properties.
> > >
> > > Additionally, the revised Appendix contains new empirical data illustrating how chemical trends correlate with the temporal divisions in our dataset. We hope that these additions clarify the relevance and importance of our time-based split evaluation in capturing the evolution of drug discovery practices.
> > >
> > >
> > > [1] DeGoey, David A., et al. "Beyond the rule of 5: lessons learned from AbbVie’s drugs and compound collection: miniperspective." Journal of medicinal chemistry 61.7 (2017): 2636-2651.\
> > > [2] Vargason, Ava M., Aaron C. Anselmo, and Samir Mitragotri. "The evolution of commercial drug delivery technologies." Nature biomedical engineering 5.9 (2021): 951-967.
> > >
> > >
> > > > W14: External validation sets are limited and potentially biased.
> > >
> > > We sincerely appreciate the reviewer’s concern regarding the limitations and potential biases in our external validation sets. We acknowledge that an appropriate assessment of model generalizability requires external validation to mitigate any inherent biases.
> > >
> > > We newly performed cross-dataset validation using PubChem and ChEMBL. These external repositories are widely recognized for their breadth and diversity in assay-centric compound data. We have carefully examined how these datasets complement our original validation sets. As shown in Table R1-6, our model maintains stable prediction performance across these diverse datasets, demonstrating its ability to generalize effectively beyond the training data.
> > >
> > > **Table R1-6.** Cross-dataset evaluation of drug-like compound identification performance on scaffold-split setting,
> > > trained on PubChem/ChEMBL and evaluated with ZINC20 compounds. Mean and standard deviation of 10
> > > fold CV are provided. Best performances are marked in bold. (AP: Average Precision)
> > > | Train set     |                 | PubChem + DrugBank    |                 |                 | ChEMBL +   DrugBank   |                 |
> > > |---------------|-----------------|-----------------------|-----------------|-----------------|-----------------------|-----------------|
> > > |               | **F1**          | **AP** | **AUROC**       | **F1**          | **AP** | **AUROC**       |
> > > | FP-SVM        | 0.268 (0.0194) | 0.334 (0.1912)       | 0.795 (0.0759) | 0.371 (0.0519) | **0.494 (0.1982)**       | 0.819 (0.0768) |
> > > | FP-XGB        | 0.254 (0.0209) | 0.320 (0.1181)        | 0.773 (0.0741) | 0.358 (0.0589) | 0.469 (0.1839)       | 0.814 (0.0784) |
> > > | FP-OCSVM      | 0.179 (0.0582) | 0.366 (0.2717)       | 0.576 (0.1949) | 0.179 (0.0582) | 0.366 (0.2717)       | 0.576 (0.1949) |
> > > | FP-SVDD       | 0.151 (0.0033) | 0.055 (0.0019)       | 0.235 (0.0173) | 0.151 (0.0033) | 0.055 (0.0019)       | 0.235 (0.0173) |
> > > | FP-DeepSVDD   | 0.147 (0.0294) | 0.080 (0.0146)        | 0.415 (0.1224) | 0.147 (0.0294) | 0.080 (0.0146)        | 0.415 (0.1224) |
> > > | FP-nnPU       | 0.244 (0.0182) | 0.240 (0.0816)        | 0.749 (0.0556) | 0.327 (0.0525) | 0.380 (0.1999)        | 0.778 (0.0812) |
> > > | FP-PU with NN | 0.241 (0.0265) | 0.228 (0.0556)       | 0.702 (0.056)  | 0.311 (0.0495) | 0.396 (0.1701)       | 0.778 (0.0874) |
> > > | DeepDL        | 0.170 (0.0199)  | 0.092 (0.0112)       | 0.590 (0.0233)  | 0.195 (0.0389) | 0.102 (0.0196)       | 0.612 (0.0686) |
> > > | D-GCAN        | 0.213 (0.0232) | 0.135 (0.0153)       | 0.685 (0.0436) | 0.314 (0.062)  | 0.211 (0.0601)       | 0.737 (0.1076) |
> > > |
> > > | BOUNDR.E      | 0.496 (0.0287) | 0.444 (0.0303)       | 0.873 (0.0167) | 0.513 (0.0451) | 0.435 (0.0889)       | 0.869 (0.0258) |
> > > | BOUNDR.E_MULT  | **0.501 (0.0232)** | **0.460 (0.038)**         | **0.875 (0.0157)** | **0.546 (0.0406)** | 0.484 (0.0729)       | **0.876 (0.0267)** |
> > >
> > > We have updated the Experiments section and the Appendix to include a detailed description of the additional validation sets and how they address concerns of bias by measuring the distribution intersection of each datasets. We hope that these adjustments address the reviewer’s concerns and clearly demonstrate the robustness of our model across a broad chemical landscape.

---

> > > > ### Author Response · Authors · 2024-11-23
> > > > **Response to Reviewer jyv4's comments (7)**
> > > >
> > > > > W15:The relationship between model predictions and actual experimental outcomes is not explored
> > > >
> > > > We fully agree with the reviewer’s point that exploring the relationship between model predictions and actual experimental outcomes is crucial for demonstrating the practical impact of our work. We acknowledge that such real-world validation, particularly for filtered drug candidates, is essential to establish confidence in our computational model.
> > > >
> > > > While experimental validation remains the gold standard, computational predictions play a pivotal role in the early stages of drug discovery. By narrowing the candidate space from vast chemical libraries to a more manageable subset of high-potential compounds, our model aims to accelerate the discovery pipeline. This approach allows for a more targeted focus in subsequent experimental validation, which is costly and time-intensive.
> > > >
> > > > We are actively exploring collaborations with drug discovery research groups to validate our filtered drug candidates experimentally. This validation would include a series of in vitro assays to test efficacy and toxicity, followed by detailed pharmacokinetic (PK) and pharmacodynamic (PD) studies. The screened compounds would be assessed for binding affinity, biological activity, and safety profiles to provide a comprehensive proof-of-concept for our data-driven approach.
> > > >
> > > > These discussions have been included in the Discussion section of the revised manuscript.
> > > >
> > > >     Further experimental validation of the screened compounds, including efficacy, toxicity and PK/PD profiles, may provide more    convincing results on the utility data-driven drug filters in drug discovery endeavours.
> > > >
> > > > We hope that this additional context clarifies the future directions of our research and reinforces the value of computational predictions in expediting the drug discovery process.
> > > >
> > > > **Questions**
> > > > > Q1. What theoretical guarantees exist for avoiding local optima in the EM-like optimization process?
> > > >
> > > > Thank you for highlighting such an important point of local optima avoidance in our EM-like optimization process to our notice. We missed out in describing the critical weakness of the EM algorithm and how our approach aims to avoid it.\
> > > > Classical EM algorithms, including K-means clustering and GMMs, typically converge to local optima due to their deterministic and hill-climbing nature of monotonic increase in likelihood. Our method incorporates two distinct features that provide greater flexibility:
> > > > 1. **Stochastic Neural Network Optimization:** Unlike traditional EM, our EM-like optimization leverages a neural network to model the latent space, which is trained using the Adam optimizer—a stochastic gradient-based method. This introduces an element of randomness, providing a non-deterministic trajectory that has the potential to escape shallow local optima, unlike the classical EM hill-climbing behavior.
> > > > 2. **Multi-Initialization and Ensemble Strategy:** As described in the response to the previous reply, we have adopted a multi-initialization approach that resulted in performance improvements by avoiding local optima.
> > > >
> > > > We hope these additions clarify the theoretical and practical robustness of our method in addressing the limitations associated with local optima.
> > > >
> > > > > Q2. How does the method handle conflicting or inconsistent information between structural and knowledge-based representations?
> > > >
> > > > We greatly appreciate the reviewer's insightful comment regarding how our method addresses potential conflicts between structural and knowledge-based representations. We understand that addressing such discrepancies is crucial for robustness.
> > > >
> > > > To mitigate potential inconsistencies, we introduced Geodesic mixup-inspired loss terms—S-mix, K-mix, and KS-mix—which facilitate the learning of the intermediate space between conflicting representations. Specifically:
> > > > - **S,K-mix**: These loss terms focus on intra-space interpolation within the structural (S-mix) and knowledge-based (K-mix) embeddings, respectively. By encouraging the model to interpolate between known data points, it learns a smoother and more continuous embedding space, reducing sensitivity to local conflicts.
> > > >
> > > > **KS-mix**: This component specifically targets inter-space interpolation, blending structural and biomedical representations. It creates synthetic data points that reflect a balanced compromise between structural and biomedical features, enabling the model to harmonize inconsistencies and achieve a unified representation.
> > > >
> > > > In our ablation studies, we found that the inclusion of these mixup techniques significantly improved predictive performance metrics (Table R1-2). As detailed in the response to W2, the model’s alignment improved in terms of both accuracy and embedding coherence, demonstrating the efficacy of our approach in resolving conflicts between structural and biomedical spaces.
> > > >
> > > > We have provided these additional clarifications in the Methods and Appendix Ablation study sections.

---

> > > > > ### Author Response · Authors · 2024-11-24
> > > > > **Response to Reviewer jyv4's comments (8)**
> > > > >
> > > > > Here, we provide responses for all remaining issues.
> > > > >
> > > > > > **W12:** Zero-shot toxicity prediction results lack proper error analysis and statistical validation.
> > > > >
> > > > > Thank you for your valuable feedback regarding the lack of comprehensive validation for our zero-shot toxicity prediction task. We acknowledge the need for more rigorous error analysis and statistical confirmation. In response, we have undertaken two additional measures:
> > > > >
> > > > > 1. **Error Analysis - Case Study with "partially withdrawn" drugs**: We conducted an in-depth error analysis on the compounds predicted as "in-boundary" by our model, identifying a trend of predictions involving drugs referred to as “partially withdrawn”—drugs that are approved in some regions but withdrawn in others, in contrary to “fully withdrawn” drugs. This category represents complex cases where the criteria for withdrawal may vary. Our analysis across 10 trials revealed a **significantly higher presence of partially withdrawn drugs in the in-drug-boundary predicted set (61.2%)** compared to out-drug-boundary ones (38.8%) with p-value of 7.8E-3 (paired t-test). This suggests that our model’s predictions reflect real-world complexities in regulatory approval, while maintaining a false positive ratio of 0.52, with 60% of these false positives falling into this partially withdrawn category. These insights are now included in the Appendix.
> > > > >
> > > > > 2. To provide statistical validation and further support the robustness of our method, we **comprehensively provide the p-values from one-sided paired t-test**, which validated the significance of our model's predictive capabilities. Statistically comparable values (p-value < 0.05) are now clearly marked in the main manuscript as shown in Table R1-7 here, and a comprehensive table of p-values compared to the best model is provided in the Appendix.
> > > > >
> > > > > We hope these additions address your concerns by clarifying the context of our results and providing a statistically grounded assessment of our method’s performance.
> > > > >
> > > > > **Table R1-7.** Zero-shot toxic compound filtering results with statistical validation. The best performances and the comparable values (paired t-test p-value < 0.05) are marked in bold, and second best underlined.
> > > > >
> > > > > |  | Withdrawn | Hepatotoxic | Cardiotoxic | Carcinogenic |
> > > > > |---|---|---|---|---|
> > > > > | Ours | **0.523 (0.0414)** | **0.541 (0.0284)** | **0.207 (0.0190)** | **0.208 (0.0436)** |
> > > > > | DrugMetric | N/A | 0.779 (0.0731) | 0.767 (0.1189) | 0.828 (0.0870) |
> > > > > | DGCAN | 0.913 (0.0203) | 0.859 (0.0237) | 0.884 (0.0452) | 0.953 (0.0177) |
> > > > > | FP-SVM | 0.983 (0.001) | 0.978 (0.001) | 0.862 (0.006) | 0.980 (0.002) |
> > > > > | FP-XGB | 0.969 (0.0033) | 0.968 (0.0033) | 0.859 (0.0100) | 0.935 (0.0100) |
> > > > > | FP-OCSVM | 0.689 (0.0018) | **0.531 (0.0033)** | 0.248 (0.0059) | 0.862 (0.0010) |
> > > > > | FP-SVDD | 0.949 (0.002) | 0.932 (0.002) | 0.921 (0.003) | 0.988 (0.001) |
> > > > > | FP-DeepSVDD | 0.811 (0.022) | 0.802 (0.020) | 0.867 (0.032) | 0.564 (0.063) |
> > > > > | FP-nnPU | 0.946 (0.0089) | 0.939 (0.0067) | 0.865 (0.0279) | 0.860 (0.0172) |
> > > > > | FP-PU | 0.949 (0.0065) | 0.938 (0.0054) |  0.826 (0.0212) | 0.851 (0.0087) |
> > > > > |
> > > > >
> > > > > > **Q3.** How could the method be adapted for specific therapeutic area focusing?
> > > > >
> > > > > We thank the reviewer for raising the important point of how our method could be adapted to focus on specific therapeutic areas. This question directly addresses the model’s versatility and potential real-world impact in targeted drug discovery.
> > > > >
> > > > > One of the strengths of our *one-class boundary approach* is its adaptability to domain-specific contexts by relying solely on the input positive labels. To explore this flexibility, **we newly designed and conducted a concept study, using anti-cancer drugs.** We first filtered our training set to include only drugs classified under the ATC code 'L' (Antineoplastic and immunomodulating agents), which specifically targets the anti-cancer domain. This narrowed training set of 239 drugs allowed our model to learn a more focused boundary representative of the anti-cancer chemical space. We investigated this anti-cancer BounDr.E model with two scenarios:
> > > > >
> > > > > 1. When **filtering the 10k generated compounds with anti-cancer target protein pocket as conditions**, the anti-cancer-boundary obtained much higher ratio of drug candidates compared to the general drug boundary, which means the model adequately learned the protein target context of anti-cancer drugs (Table R1-8).
> > > > > 2. On contrary and interestingly, false-positive ratio on toxic and carcinogenic compounds was significantly reduced when applying the anti-cancer-specific boundary, highlighting the model’s ability to filter out irrelevant or potentially harmful compounds more effectively (Table R1-9) with more compact boundary, while encompassing the contexts of anti-cancer drugs.
> > > > >
> > > > > The results imply that **our model's anti-cancer variant, while providing a broader boundary for anti-cancer generated compounds, shows strictness for toxic compounds, tailored for anti-cancer drug discovery.**

---

> > > > > > ### Author Response · Authors · 2024-11-24
> > > > > > **Response to Reviewer jyv4's comments (9)**
> > > > > >
> > > > > > **Table R1-8.** Filtering anti-cancer target-based generated molecules with general BounDr.E and anti-cancer-BounDr.E models. Approximately 10k molecules were generated and filtered for BCR-ABL, EGFT and CDK6, three well-known anti-cancer targets. Compared to general BounDr.E model, Anti-cancer-BounDr.E model recommends more candidates, according to the generated compounds' context.
> > > > > > | Filtering Method |    BCR-ABL     |      EGFR      |      CDK6      |
> > > > > > |------------------|:--------------:|:--------------:|:--------------:|
> > > > > > | Total Generated|10,543 (100%) |  12,550 (100%) |  11,496 (100%) |
> > > > > > | Anti-cancer BounDr.E|434 (3.9%) |    434 (3.9%) |495 (4.9%) |
> > > > > > | General BounDr.E|300 (2.8%) |    374 (3.0%) |264 (2.3%) |
> > > > > > |
> > > > > >
> > > > > > **Table R1-9.** Toxic compound filtering comparison with best performances marked in bold. The anti-cancer BounDr.E model displays significant reduction in FPR compared to general BounDr.E model. (FPR: False Positive Rate).
> > > > > > |  | Withdrawn | Hepatotoxic | Cardiotoxic | Carcinogenic |
> > > > > > |---|---|---|---|---|
> > > > > > | General BounDr.E | 0.523 (0.0414) | 0.541 (0.0284) | 0.207 (0.0190) | 0.208 (0.0436) |
> > > > > > | Anti-cancer BounDr.E | **0.195 (0.0363)** | **0.151 (0.029)** | **0.149 (0.0356)** | **0.148 (0.0321)** |
> > > > > > | Paired t-test p-val | 2.30E-09 | 6.80E-12 | 2.20E-04 | 1.00E-03 |
> > > > > >
> > > > > > These findings, along with a detailed explanation of the methodology, have been incorporated into the Discussion section and the Appendix of the revised manuscript. We hope this case study adequately demonstrates the model's adaptability to targeted therapeutic contexts.

---

> > > > > > > ### Author Response · Authors · 2024-11-27
> > > > > > > **Response to Reviewer jyv4's comments (10/10)**
> > > > > > >
> > > > > > > Dear Reviewer jyv4,
> > > > > > >
> > > > > > > Thank you all for your comprehensive and constructive reviews. The final updated manuscript has now been uploaded with statistical validation table for all of our performance tables in the manuscript in the last section of the Appendix, and we would greatly appreciate your time in reassessing it. We have dedicated the past 14 days to carefully revising our manuscript, thoroughly addressing each of the 17 points you raised. It is our hope that the revised version reflects our effort to meet the expectations and high standards of this conference.
> > > > > > >
> > > > > > > We are deeply committed to delivering a high-quality contribution, and your continued engagement in this process would be very much appreciated. We welcome any further comments or suggestions you may have during this discussion phase.

---

> > > > > > > > ### Author Response · Authors · 2024-11-30
> > > > > > > > **Gentle reminder for Reviewer jyv4**
> > > > > > > >
> > > > > > > > Dear Reviewer jyv4,
> > > > > > > >
> > > > > > > > We thank you for your effort in reviewing our paper and providing us with valuable feedback. We have dedicated all our efforts in thoroughly revising our work based on each of the 14 Weaknesses and 3 Question points you raised.
> > > > > > > >
> > > > > > > > Before the **end of the discussion in three days**, your response on whether we have adequately addressed your concerns would be much appreciated.
> > > > > > > >
> > > > > > > > Thank you for your continued consideration.

---

### Author Response · Authors · 2024-11-21
**Updates on the results of Submission9796.**

We sincerely thank the reviewers for their insightful comments and constructive feedback, which enabled us to improve our work further. Notably, we have updated the Results with additional experiments inspired from the valuable recommendations to further demonstrate both generalizability and the utility of our model:

1. **Cross-dataset validation:** We conducted a cross-dataset validation experiment, using PubChem or ChEMBL compounds as trainset and ZINC20 compounds as test set. Since the distribution of the non-drug datasets differ, we were able to observe the models' generalizability regarding the non-drug set used.

    **Table G1.** Cross-dataset evaluation of drug-like compound identification performance on scaffold-split setting,
    trained on PubChem/ChEMBL and evaluated with ZINC20 compounds. Mean and standard deviation of 10 fold CV are provided. Best performances are marked in bold.
    | Train set     |                 | PubChem + DrugBank    |                 |                 | ChEMBL +   DrugBank   |                 |
    |---------------|-----------------|-----------------------|-----------------|-----------------|-----------------------|-----------------|
    |               | **F1 (↑)**          | **Average Precision (↑)** | **AUROC (↑)**       | **F1 (↑)**          | **Average Precision (↑)** | **AUROC (↑)**       |
    | FP-SVM        | 0.268 (0.0194) | 0.334 (0.1912)       | 0.795 (0.0759) | 0.371 (0.0519) | **0.494 (0.1982)**       | 0.819 (0.0768) |
    | FP-XGB        | 0.254 (0.0209) | 0.320 (0.1181)        | 0.773 (0.0741) | 0.358 (0.0589) | 0.469 (0.1839)       | 0.814 (0.0784) |
    | FP-OCSVM      | 0.179 (0.0582) | 0.366 (0.2717)       | 0.576 (0.1949) | 0.179 (0.0582) | 0.366 (0.2717)       | 0.576 (0.1949) |
    | FP-SVDD       | 0.151 (0.0033) | 0.055 (0.0019)       | 0.235 (0.0173) | 0.151 (0.0033) | 0.055 (0.0019)       | 0.235 (0.0173) |
    | FP-DeepSVDD   | 0.147 (0.0294) | 0.080 (0.0146)        | 0.415 (0.1224) | 0.147 (0.0294) | 0.080 (0.0146)        | 0.415 (0.1224) |
    | FP-nnPU       | 0.244 (0.0182) | 0.240 (0.0816)        | 0.749 (0.0556) | 0.327 (0.0525) | 0.380 (0.1999)        | 0.778 (0.0812) |
    | FP-PU with NN | 0.241 (0.0265) | 0.228 (0.0556)       | 0.702 (0.056)  | 0.311 (0.0495) | 0.396 (0.1701)       | 0.778 (0.0874) |
    | DeepDL        | 0.170 (0.0199)  | 0.092 (0.0112)       | 0.590 (0.0233)  | 0.195 (0.0389) | 0.102 (0.0196)       | 0.612 (0.0686) |
    | D-GCAN        | 0.213 (0.0232) | 0.135 (0.0153)       | 0.685 (0.0436) | 0.314 (0.062)  | 0.211 (0.0601)       | 0.737 (0.1076) |
    |
    | BOUNDR.E (Ours)      | 0.496 (0.0287) | 0.444 (0.0303)       | 0.873 (0.0167) | 0.513 (0.0451) | 0.435 (0.0889)       | 0.869 (0.0258) |
    | BOUNDR.E$_{MULT}$ (Ours)  | **0.501 (0.0232)** | **0.460 (0.038)**         | **0.875 (0.0157)** | **0.546 (0.0406)** | 0.484 (0.0729)       | **0.876 (0.0267)** |
    |

    In most of the cases, our model achieved the best performances, with notably less decline from the original ZINC20 performance table compared to binary classifiers and PU-learning approaches (Manuscript Table 1).

2. **Rational drug-discovery pipeline case study:** We have performed a more systematic case study to replace our sole BCR-ABL (Imatinib) case, which now includes two more anti-cancer target cases of EGFR (Erlotinib) and CDK6 (Ribociclib). Especially, we have compared our filtering abilities with traditional property-based filters and finally show that when using our model as a complementary initial filter, we were able to minimize the initial candidate pool for downstream experimental validation and simultaneously saving computational resources and time.

    **Table G2.** The number and the ratio of remaining compounds after applying diverse filters to the 10k Targetdiff-generated molecules for three protein targets. Final row `All filters` refers to the pipeline of first applying BounDr.E, then sequentially applying IC50, Ro5 and PAINS filters.

    | Filtering Method |    BCR-ABL     |      EGFR      |      CDK6      |
    |------------------|:--------------:|:--------------:|:--------------:|
    | Total Generated  |  10,543 (100%) |  12,550 (100%) |  11,496 (100%) |
    | PAINS filter     | 10,078 (95.7%) | 11,878 (94.6%) | 10,996 (95.6%) |
    | Ro5              |  4,997 (47.5%) |  6,520 (52.0%) |  5,782 (50.3%) |
    | Predicted IC50   |  2,786 (26.5%) |   1,018 (8.1%) |  4,734 (41.2%) |
    | BounDr.E         |     300 (2.8%) |    374 (3.00%) |     264 (2.3%) |
    | **All filters**      |     **38 (0.36%)** |     **17 (0.15%)** |     **47 (0.40%)** |

We hope the additional results further  highlight the contributions of our work. The revised manuscript will be available shortly, incorporating all the updates.

---

### Author Response · Authors · 2024-11-25
**Notification and invitation to visit our uploaded revised submission**

Dear Reviewers,

We are pleased to share that we have just uploaded a thoroughly revised manuscript, with changes marked in blue. Your comments have been incredibly helpful in guiding the revision process, and we are grateful for the time and effort you have each invested in providing valuable feedback on our submission.

Each of your suggestions was carefully considered, and we have implemented substantial changes to address the key points raised, through either additional experiments or clarifications in the main text. We believe these revisions have significantly enhanced the clarity and quality of the paper.

We warmly invite you to review the revised version and would greatly appreciate any further feedback or suggestions you might have to be discussed in the remaining discussion phase. We would like to also note that the full table of statistical validation for all performance tables are still in route since we have performed numerous benchmark test, and we assure you it will be included as the last section of the Appendix before the deadline arrives.

Thank you again for your contributions to this process.

---

### Note · Authors · 2025-01-24

I have read and agree with the venue's withdrawal policy on behalf of myself and my co-authors.